



# First direct observation of sea salt aerosol production from blowing snow above sea ice

Markus M. Frey[1], Sarah J. Norris[2], Ian M. Brooks[2], Philip S. Anderson[3], Kouichi Nishimura[4], Xin Yang[1], Anna. E. Jones[1], Michelle G. Nerentorp Mastromonaco[5], David H. Jones[1,6], and Eric W. Wolff[7]

[1]Natural Environment Research Council, British Antarctic Survey, Cambridge, UK
[2]University of Leeds, Leeds, UK
[3]Scottish Association for Marine Science, Oban, UK
[4]Nagoya University, Nagoya, Japan
[5]IVL Swedish Environmental Institute, Stockholm, Sweden
[6]now at: Neptec, UK LtD., Oxford, UK
[7]University of Cambridge, Cambridge, UK

Correspondence: M. M. Frey (maey@bas.ac.uk)

Abstract. Two consecutive cruises in the Weddell Sea, Antarctica, in winter 2013 provided the first direct observations of sea salt aerosol (SSA) production from blowing snow above sea ice, thereby validating a model hypothesis to account for winter time SSA maxima in polar regions not explained otherwise. Blowing or drifting snow always lead to increases in SSA during and after storms. Observed aerosol gradients suggest that net production of SSA takes

place near the top of the blowing or drifting snow layer. The observed relative increase of SSA concentrations with wind speed suggests that on average the corresponding aerosol mass flux during storms was equal or larger above sea ice than above the open ocean, demonstrating the importance of the blowing snow source for SSA in winter and early spring. For the first time it is shown that snow on sea ice is depleted in sulphate relative to sodium with respect to sea water. Similar depletion observed in the aerosol suggests that most sea salt originated from snow

on sea ice and not the open ocean or leads, e.g. on average 93% during the 8 June and 12 August 2013 period. A mass budget calculation shows that sublimation of snow even with low salinity ($<1$ psu) can account for observed increases of atmospheric sea salt from blowing snow. Furthermore, snow on sea ice and blowing snow showed no or small depletion of bromide relative to sodium with respect to sea water, whereas aerosol at 29 m was enriched suggesting that SSA from blowing snow is a source of atmospheric reactive bromine, an important ozone sink, with

bromine loss taking place preferentially in the aerosol phase between 2 and 29 m above the sea ice surface. Evaluation of the current model for SSA production from blowing snow showed that the parameterisations used can generally be applied to snow on sea ice. Snow salinity, a sensitive model parameter, depends to a first order on snowpack depth and therefore is higher above first-year than above multi-year sea ice. Shifts in the ratio of FYI and MYI over time are therefore expected to change the seasonal SSA source flux and contribute to the variability of SSA in ice

cores, which both represents an opportunity and a challenge for the quantitative interpretation of the sea salt sea ice proxy. It is expected that similar processes take place above Arctic sea ice.





## 1 Introduction

Atmospheric aerosol represents the largest source of uncertainty in global climate predictions (Boucher et al., 2013), and includes sea salt aerosol (SSA), which is the main background aerosol above the oceans. In general global climate and aerosol models strongly under-predict Aitken and accumulation mode particle concentrations compared to observations (Mann et al., 2014) and do not capture the winter maximum of SSA observed at several locations in the Arctic and Antarctica (e.g. Huang and Jaeglé, 2017). A quantitative understanding of SSA sources is critical since SSA influences radiative forcing and therefore climate both directly by absorbing and scattering sunlight and indirectly by modifying the reflectivity, lifetime, and extent of clouds (O'Dowd et al., 1997; DeMott et al., 2016). Uncertainties in cloud properties explain much of the spread in the modelled climate sensitivity (Flato et al., 2013) and are due in large part to lacking knowledge about aerosol sources. SSA plays also an important role in polar tropospheric ozone and halogen chemistry through the release of active bromine in polar spring contributing to ozone depletion events (ODEs) (e.g. Yang et al., 2010; Kalnajs et al., 2013; Choi et al., 2018). Furthermore, SSA is easily measured in polar ice cores but its use as a quantitative proxy of past sea ice conditions is complicated by uncertainties related to SSA source contributions and processes as well as transport meteorology (Abram et al., 2013; Levine et al., 2014; Rhodes et al., 2017), and more recently in the case of bromide $Br^-$ (e.g. Spolaor et al., 2013) also to post-depositional processing associated with the bromine explosion chemistry of ODEs (Simpson et al., 2005; Pratt et al., 2013).

Globally most SSA originates from the open ocean, where sea spray is produced by wave breaking and bubble bursting generating film and jet drops . However, previous observations in Antarctica provide two-fold evidence of the existence of a significant SSA source associated with sea ice. First, SSA exhibits maxima during winter/spring in the atmosphere (Rankin and Wolff, 2003; Jourdain et al., 2008; Legrand et al., 2017) and in seasonally resolved ice core records (e.g. Frey et al., 2006); in deep ice cores highest values are seen during glacial periods (Wolff et al., 2003). Thus the highest values are observed when sea ice is at its seasonal or long-term maximum. And second, SSA in the lower atmosphere above coastal Antarctica is strongly depleted in sulphate ($SO_4^{2-}$) compared to seawater (Wagenbach et al., 1998) as are also brine and frost flowers in the sea ice nearby (Rankin et al., 2000). Fractionation of $SO_4^{2-}$ and to much lesser extent of sodium ($Na^+$) in sea ice occurs during the cooling of sea ice brine down to its eutectic point due to precipitation of the mineral mirabilite ($Na_2SO_4 \cdot 10 H_2O$) at temperatures below -6.4 °C resulting in ion ratios in liquid and solid phase that are different compared to sea water (Butler et al., 2016). At first it had been thought that observed fractionated SSA originates mostly from highly saline frost flowers, which exhibit a similar sulphate depletion (Rankin et al., 2000). However, recent laboratory (Roscoe et al., 2011; Yang et al., 2017) and model (Huang and Jaeglé, 2017) studies indicate that frost flowers likely play only a minor role in producing





SSA because they do not get easily airborne and do not occur widely enough in space and time to produce large enough SSA quantities.

A more recent hypothesis based on a numerical model suggests that salty blowing snow that undergoes sublimation may be a significant source of SSA with a production per unit area equal or larger than that above the open ocean (Yang et al., 2008). Indeed, SSA winter maxima observed at a number of locations in the polar regions can only be explained if the SSA source from blowing snow based on the parameterisation of Yang et al. (2008) is included in the model (Huang and Jaeglé, 2017). However, the blowing snow hypothesis and mechanism lack validation by direct observation. In particular, the applicability of the scheme used by all model studies to date (Huang and Jaeglé, 2017; Rhodes et al., 2017; Yang et al., 2018) for sea ice is not known, because model parameterisations are based on blowing snow measurements above ice sheets (Budd, 1966; Mann et al., 2000; Nishimura and Nemoto, 2005) and the Canadian prairies (Déry and Yau, 2001, and ref. therein).

In this study we report the first direct observations of sea salt aerosol from within the Antarctic sea ice zone during winter, when the blowing snow source is expected to be most active. We discuss SSA variability during and after blowing snow events supported by a unique set of measurements of physical and chemical properties of atmospheric aerosol, snow particles and the snowpack on sea ice. We then critically evaluate the current model parameterisation of sea salt aerosol production from blowing snow based on the *in situ* observations and discuss chemical fractionation of sulphate and bromide, aerosol size distribution and mass budget of sea salt aerosol above sea ice. A comparison between our observations and a global chemistry transport model is described in a companion study (Yang et al., 2018).

## 2 Methods

Two consecutive Antarctic expeditions were carried out in the Weddell Sea aboard the German icebreaker *RV Polarstern* in 2013, the *Antarctic Winter Ecosystem Climate Study (AWECS)*, ANT-XXIX/6 between 8 June and 12 August (Lemke, 2014), and the spring expedition ANT-XXIX/7 between 14 August and 16 October (Meyer, 2014) (Fig. 1). We report mainly on results from ANT-XXIX/6, which provided a significantly more extensive dataset. Atmospheric measurements and sampling were carried out continuously from the crow's nest of *RV Polarstern*, and on the sea ice during nine 3–104 hr long ice stations (Table 1). Ice station S3 was occupied for 3 hr (Table 1) allowing only for collection of snow samples. Instruments were set up on the sea ice typically at 0.8 to 1.5 km distance from the ship. Sea ice thickness measurements during ANT-XXIX/6 (Arndt and Paul, 2018) showed that ice stations S1-6 in the eastern sector of the Weddell Sea were on first-year sea ice, whereas ice stations S7-S9 near the Antarctic Peninsula were on multi-year sea ice (Fig. 1). This is in agreement with the observation that persistent multi-year sea ice in the Weddell sea occurs only east of the Antarctic peninsula due to a circulating ocean current. The distance to the nearest open water was ∼600–1000 km for ice stations S1-6 and ∼200–400 km for ice stations S7-9 (Fig. 1, Table 1). As *RV Polarstern* travelled south day lengths decreased until the sun remained entirely below the horizon





between 23 June and 7 July providing only a few hours of twilight per day. All times are in UTC. A summary of instrumentation and temporal coverage is given in Table 2 and experimental details for each measurement are described below.

## 2.1 Airborne snow particles

Size resolved number densities of airborne snow particles in the diameter range 36–490 µm were measured at a mean height of ~0.16 m (range 0.07-0.37 m) on the sea ice, and at 29 m from the crow's nest of *RV Polarstern* using an open-path snow particle counter (SPC-95, Niigata Electric Co., Ltd) described previously (Nishimura and Nemoto, 2005; Nishimura et al., 2014). In brief, the SPC is a single slit sensor with a laser diode and measures diameter and number of drifting snow particles by detecting their shadows. The SPC is mounted on a self-steering wind vane, and hence, the sampling area $A$ ($2\times10^{-3}$m $\times$ $25\times10^{-3}$m) and volume ($A\times0.5\times10^{-3}$m) are maintained perpendicular to the prevailing wind direction. Electric pulse signals resulting from snow particles passing through the sampling volume are sent to a transducer and an analysing data logging system. Assuming that the detected particle size is the equivalent diameter $d_p$ of a sphere each signal is classified into one of 64 mean particle diameter classes between 36 and 490 µm. The SPCs were calibrated by the manufacturer in Japan at -10 and -30 °C for particle diameters 137, 229, 314, 399 and 449 µm, respectively. The instrument output at the calibration temperature of -10 °C, close to the ANT-XXIX/6 median of -11.8 °C, showed very good agreement with the expected representative particle diameter. Therefore no temperature corrections were applied. Particle counts from the smallest diameter class ($d_p$=36 µm ) have large uncertainties due to the instrument's detection limit, whereas particle counts from the largest diameter class ($d_p$=490 µm) showed frequent spikes due to detection of precipitating snow. Particle counts from the smallest and largest diameter class were therefore discarded.

Particle counts $h$ measured at a sampling rate $SR$ of 1 Hz are integrated to 1-minute values and divided by sampling area $A$ to obtain particle number flux $F$ in units of m$^{-2}$ minute$^{-1}$. Snow particle number densities for the diameter range 46–478 µm, $N_{46-478}$, are then computed using $N_{46-478} = FU^{-1}$ with horizontal wind speed $U$ and are reported at ambient temperature, pressure and relative humidity in units of m$^{-3}$. Wind speeds measured near the crow's nest at 39 m by the ship's meteorology observatory (see section 2.5) and on the sea ice at 2 m by a sonic anemometer (see section 2.5) were extrapolated to the respective instrument heights. To do this a logarithmic wind profile $U(z)$ is assumed given by $U(z) = k\,u_* \ln(z/z_0)$, with measurement height $z$, the von Karman constant $\kappa$, friction velocity $u_*$ and the surface roughness length of momentum $z_0$ set to $5.6\times10^{-5}$m as measured very consistently above snow at Halley (King and Anderson, 1994). Wind speed at instrument level is then derived using $U(z_2) = U(z_1) \ln(z_2/z_0) / \ln(z_1/z_0)$. It should be borne in mind that the distortion of flow caused by the ship may mean both that speed at 39 m is not representative of flow in the far field at that height, and further, that the turbulent field strength, which governs the gradient of the logarithmic profile, may be a residual from a different, likely lower, height. Thus, we suggest care when interpreting the data, and estimate that the conversion from particle counts



to number density be seen as an estimate suitable for comparison, rather than quantitative with a well behaved uncertainty.

## 2.2 Sea salt aerosol

Size-resolved number densities of SSA sized particles were measured at mean heights of 0.19 m (range 0.13-0.25 m) and 2.16 m (range 1.98-2.2 m) on the sea ice, and at 29 m from the crow's nest of $RV\ Polarstern$ using a Compact Lightweight Aerosol Spectrometer Probe (CLASP) (Hill et al., 2008; Norris et al., 2008). The CLASP is a closed-path optical particle spectrometer, which aspirates sample air at a nominal flow rate $Q$ of $\sim$3 STP-L min$^{-1}$, which is actively controlled by onboard electronics, and recorded to allow subsequent correction of the particle spectra for any minor flow variations. An improved version of the CLASP instrument as used by Norris et al. (2012) was deployed, which measures a 16–channel size spectrum covering particle diameters $d_p$ in the range of 0.36–11.62 µm at ambient humidity at a sampling rate $SR$ of 1 Hz. CLASP pump and scatter cell were calibrated in the lab before and after the cruise. Particle losses to inlet walls are minimized by use of a short inlet of <0.3 m. As an upper limit for particle losses we adopt the estimates by Norris et al. (2012) for a different inlet configuration, which amount to 43% at $d_p = 11.32$ µm, 19% at $d_p = 6.06$ µm and 0.1% at $d_p = 0.44$ µm, respectively. With an estimated cut-off diameter of >11 µm all but the coarsest SSA particles are expected to be detected. Number densities $N_{0.4-12}$ are computed from particle counts $h$ with $N_{0.4-12} = h\,SR\,Q^{-1}$ at IUPAC standard temperature (273.15 K) and pressure (1 bar) and then averaged to 1-minute means. Engine pollution was effectively removed from the crow's nest data prior to averaging by excluding all data when relative wind direction was in the 135–225 ° sector encompassing the ship's engine stack. Due to pump failure of the CLASP unit in the crow's nest usable data at the 29 m level are only available from 8 June to 26 July 2013.

## 2.3 Aerosol chemical composition

Aerosol was collected on filters via continuous low-volume sampling using open-face filter holders protected by a wind shield. The method described below follows the approach from a previous study in coastal Antarctica (Wolff et al., 1998). One sampling unit was deployed on the sea ice for the duration of each ice station, and another one was operated continuously from the crow's nest of $RV\ Polarstern$.

Aerosol filters were polytetrafluoroethylene (PTFE) 1.0–µm pore membrane filters of 37 mm diameter (Zefluor®P5PL037 Pall Lab.) with an aerosol retention of >99.99% as stated by the manufacturer. Filters were pre-mounted prior to the expedition in 3-piece styrene acrylonitrile (SAN) filter holders onto porous cellulose support pads (Pall International Sarl). At the head of each sampling unit, the filter holder was mounted, open-face downward, inside an upturned polyethylene jar intended to prevent blowing snow blocking the filter. On the sea ice the jar was attached to a metal mast at $\sim$2 m above the snowpack and on the crow's nest to a horizontal metal arm at $\sim$29 m above the sea ice surface, and was free to swing in the wind, aiding the removal of snow and rime. An air pump was attached with PFA tubing $\sim$10 m downstream of the sampling head to pull ambient air through the filters. The sea




ice unit had a 12 VDC diaphragm vacuum pump (22 W, N815-KNDC, KNF Neuenberger) powered by a rechargeable lead-acid battery (12-5000X, Sunlyte) with an average flow rate of 9.8 STP-L min$^{-1}$, whereas the crow's nest unit had a 220 ADC diaphragm vacuum pump (420 W, Mod-No. DOA-P725-BN, GAST, Michigan, USA) with a mean flow rate of 8.8 STP-L min$^{-1}$.

Total sample air volume was measured upstream of the pump using a diaphragm gas meter (KG-2-G1.6, BES), which was not temperature compensated. However, gas meter temperature continuously recorded with a TinyTag logger (RS, Gemini) and atmospheric pressure available from the ship's meteorology observatory (see section 2.5) allowed to determine the total air sample volume at IUPAC standard temperature and pressure. Rotameters with glass float (ColeParmer) installed upstream of the gas meter were used to manually check flow rates 1-2 times daily.

Post-season lab tests confirmed accuracy of the crow's nest gas meter, whereas the sea ice gas meter showed a negative bias of 30%, which is accounted for in the computation of final atmospheric concentrations. Filter sampling intervals were on average 6 hr (range 1.5-8.6 hr) at 2 m and 12 hr (range 3.5-31 hr) at 29 m, with more frequent filter changes applied during storms. The resulting sample air volumes were 3.3 (range 0.8-5.1) m$^3$ at 2 m and 6.4 (range 1.8-17.3) m$^3$ at 29 m.

Since isokinetic sampling could not be applied, sea salt aerosol concentrations are expected to be generally underestimated, especially during high wind speeds. The cut-off diameter of the air intake system, which critically controls the observed aerosol load, had not been quantified. However, previously a cut-off diameter of 6 μm was empirically estimated for a similar sampling set up with filter face velocities of 1.1 m s$^{-1}$ (Wagenbach et al., 1998). In this study average filter face velocities were ~0.15 m s$^{-1}$ implying a cut-off diameter smaller than 6 μm. The filter low-bias for
sea salt is found to be on average 40% compared to the CLASP measurements as discussed in section 3.4.4.

Filter units, consisting of filters, support pads and filter holders were cleaned and assembled prior to shipping to Antarctica in a Class 100 clean laboratory in Cambridge (United Kingdom). Filters and support pads were soaked over-night in methanol (ACS >99.9%, Fisher Scientific) in batches of 25, rinsed 4 times with ultra-high purity water (UHP, electric resistivity 18.2 MΩcm), micro-waved for 3 minutes then rinsed once more with UHP. Filters were dried
on their support pads under vacuum in a clean dessiccator. Filter holders were UHP-rinsed, placed for 10 minutes into an ultra-sonic bath, UHP-rinsed 3 times and micro-waved for 5 minutes, then soaked over-night in UHP, followed by 4 UHP rinses and 5 minute microwaving. Filter holders were then dried in the incoming, filtered airflow of the Class 100 laboratory. Filter units were assembled in the Class 100 laboratory, sealed air-tight with plugs and then bagged in 2 layers of polyethylene bags.

In Antarctica, filter units were removed from the bags, mounted onto the sampling head by removing one plug, then opened to give open-face sampling. Field blanks were collected at regular intervals by placing a filter onto the head and leaving it for a few minutes without pumping. After sampling the procedure was reversed, and filter units were shipped in a freezer (-20 °C) back to the United Kingdom and kept frozen until prior to extraction. Filters were transferred in the Class 100 laboratory into 10 ml pre-cleaned polysterene sample vials (Dionex AS-AP Autosampler
vial). Pre-cleaning of vial, lid and septum involved UHP-rinsing followed by 3-minute micro-waving repeated 5 times,





soaking over-night in UHP, repeating the same set of UHP-rinse and microwaving before drying and double-bagging. Filters were then extracted in 8 ml of UHP by repeated shaking and immersion into an ultra-sonic bath for >30 min. Samples were analysed using ion chromatography (IC) as described in the following section. About 7% (10 out of 151) of all filters were visibly grey from exposure to ship exhaust and therefore not analysed in order to protect the

IC columns.

The field blank, mostly due to the filter itself with contributions from field and extraction procedures, was significant for all ions (Table 3). A mean value has been subtracted from all concentrations leading in some cases to negative values. The combination of small air sample volumes and high field blanks led to relatively high limits of detection (LOD) defined here as 2 times the standard deviation of the field blank (Table 3).

2.4   Snow chemical composition

In order to determine the snow chemical composition a total of 24 snow pits was sampled at 9 ice stations during ANT-XXIX/6, between 1–2 snow pits at shorter ice stations and up to 8 at the multi-day ice stations S6 and S8 (Table 1). Snow pit profiles were sampled at 2 cm depth resolution with a custom-built, cylindrical, stainless-steel sampling tool yielding samples of $\sim$60 cm$^3$. Blowing or drifting snow was collected at approximately 0.3, 0.9 and 1.6 m

above the snow surface using modified Mellor gages with a clear thermoplastic body (Schmidt et al., 1984) referred to as rocket traps. All samples were transferred into 50 ml polypropylene tubes with screw-caps (Corning CentriStar), which prior to field deployment had been UHP-rinsed and dried in a Class 100 clean laboratory in Cambridge. One set of snow samples was melted onboard *RV Polarstern* to measure aqueous conductivity using a conductivity meter (SensIon 5, Hach) with a measurement range of 0-200 mS cm$^{-1}$ and a maximum resolution of 0.1 µS cm$^{-1}$ at low

conductivities (0-199.9 µS cm$^{-1}$). The conductivity meter has an automatic non-linear temperature compensation based on a NaCl solution and reference temperature of 25 °C and was calibrated with a standard salt solution (12.880 mS cm$^{-1}$ at 25.1 °C), certified and traceable to NIST (REAGECON Prod. No. CSKC12880, Lot No. CS1288012K1). Conductivity values were converted into practical salinity $S_p$ using the Gibbs-SeaWater (GSW) Oceanographic Toolbox (McDougall and Barker, 2011), which applies the algorithm of the Practical Salinity Scale of 1978 (PSS–

78) (Unesco, 1981, 1983) with an extension to salinities $S_p < 2$ psu (Hill and Woods, 1986). $S_p$ is reported in psu (practical salinity unit), approximately equivalent to the weight of dissolved inorganic matter in grams per kilogram of seawater, and has an accuracy as stated by the manufacturer of ±0.001 psu at low salinities (<1 psu).

The other snow samples were shipped frozen back to Cambridge and only melted prior to analysis in spring 2014 and December 2016. Elevated salinities required dilution of samples with UHP water, typically by a factor 100 for

most snow samples and a factor 10000 for sea ice and frost flower samples. Samples were analysed for major ions using Dionex ICS2000 ion chromatography systems with reagent free eluent generation. Cation analysis was performed with a CS12A separator column with isocratic methylsulfonic acid elution and a 250 µl sample loop. Anion analysis was performed using an AS17 separator column, gradient elution with potassium hydroxide and a 250 µl sample loop. Measurement accuracies were evaluated using European reference materials ERM A408 (simulated rain water)





and CA616 (groundwater) and were all within 5%. Here we only report and discuss concentrations of ions relevant to this study: $Na^+$, $Cl^-$, $SO_4^{2-}$ and $Br^-$.

Chemical fractionation of ion $x$ (=$SO_4^{2-}$, $Br^-$, $Na^+$) with respect to ion $y$ (=$Na^+$ or $Cl^-$) in aerosol and snow compared to sea water is evaluated based on the depletion factor $DF_x = (R_{RSW}\text{-}R_{spl})/R_{RSW}$, with $R$ being the

$x$:$y$ mass ratio in reference sea water (RSW) after Millero et al. (2008) and sample, respectively. Throughout $DF_{SO_4^{2-}(Na^+)}$ and $DF_{Br^-(Na^+)}$ refer to sulphate and bromide depletion with respect to $Na^+$, and $DF_{Na^+(Cl^-)}$ refers to sodium depletion with respect to $Cl^-$. $DF_x$ between 0 and 1 indicates depletion, whereas $DF_x <0$ indicates enrichment. $DF_{SO_4^{2-}}$ values below that of pure mirabilite (= -7.3) are attributed either due to measurement error or sulphate contamination from the ship's engine emissions, and were therefore removed. $DF_{Br^-}$ and $DF_{Na^+}$ smaller

than -3 were also considered outliers and removed. Propagation of the analytical error yields mean uncertainties in $DF_x$ of 0.03–0.04.

## 2.5   Ancillary measurements

General meteorology measurements were taken from *RV Polarstern* onboard sensors described in detail at https://spaces.awi.de/confluence/display/PSdevices/Bordwetterwarte, and include ambient temperature $T_a$ and relative

humidity with respect to water $RH_{aq}$ at 29 m (HMT337, Vaisala, Finland), wind speed $U$ and direction at 39 m (Sonic 2D a, Thies, Germany) and global radiation (Pyranometer CM11, Kipp&Zonen, Netherlands). Still images recorded every 5 minutes by a webcam in the crows nest provided on occasion when light conditions were adequate further qualitative information on the presence of airborne snow particles. During ice stations the three-dimensional wind components $(u, v, w)$ were measured above the sea ice at 25 Hz with an unheated sonic anemometer (Metek USA-1)

that was mounted on a mast at ∼2 m. Processing of raw sonic data in 1-min blocks included temperature cross-wind correction and a double coordinate rotation to force mean $w$ to zero (Kaimal and Finnigan, 1994; Van Dijk et al., 2006), followed by computation of friction velocity $u_*$. $T_a$ and $RH_{aq}$ were also measured above the sea ice on a 2-m mast at approximately 0.6, 1.2 and 2.0 m, respectively, using temperature-humidity probes (HMP45, VAISALA). $RH_{aq}$ was converted to relative humidity with respect to ice $RH_{ice}$. $RH_{ice}$ from the HMP45 was further corrected

after Anderson (1994) to extend the calibrated temperature range to $T_a <$-20 °C. Based on the available data the calibration of $RH_{ice}$ is most accurate for the -40 to -20 °C range, and has greater uncertainty in $RH_{ice}$ in near freezing conditions. Accurate correction of the HMT337 output was not possible since available $RH_{aq}$ values are biased as they had been post-processed by accepting only values up to 105% and setting any values >100% to 100%. In general, measuring $RH_{ice}$ during blizzards is very difficult and its estimated non-systematic error of ∼5% limits

its use for blowing snow calculations. We therefore discuss below only $RH_{ice}$ trends.





## 3 Results and discussion

The presentation and discussion of results in the sections below are organised as follows. An overview of observed meteorology, particle concentrations and aerosol chemistry (section 3.1) is followed by the description of blowing snow events during two time periods (section 3.2). We then evaluate the current model mechanism proposed by Yang

et al. (2008, 2018) for SSA production from blowing snow (section 3.3) and finally discuss chemical composition, aerosol size spectra and air-snow budget of sea salt aerosol above sea ice (section 3.4).

### 3.1 Overview of atmospheric observations

Near-zero or positive ambient temperatures $T_a$ occurred when $RV\,Polarstern$ was in the open ocean at the start and end of ANT-XXIX/6, as well as from the 22 July 2013 onwards, when the ship had moved into the marginal

sea ice zone (MIZ) closer to open water (marker B in Fig. 1,2b). After entry into the sea ice zone on 17 June 2013 (marker A in Fig. 1) $T_a$ decreased to below -20 °C but showed thereafter frequent increases associated with storms (Fig. 1,2a-b). The correlation between $T_a$ and $U_{39m}$ is weak but significant (R = 0.37, p<0.05).

Winter storms occurred frequently with wind speeds ranging between 10 and 20 m s$^{-1}$, occasionally exceeding 20 m s$^{-1}$ (Fig. 2a), and coincided almost always with snowfall inferred from direct observation and presence of

clouds. Bearing in mind that wind direction and origin of air mass depend on the ship's position relative to the path of a low or high pressure system the following is observed: winds were most of the time from W to SSW advecting cold (-15 to -30 °C) air of a low water load with specific humidities $q_v < 3$ g kg$^{-1}$ (Fig. 3). Second most frequent were winds from SSE to SE advecting relatively warm air often between -5 and 0 °C with higher $q_v$ values often between 3–5 g kg$^{-1}$ (Fig. 3) indicating origin from lower latitudes. And third most frequent were winds from

the N coinciding with the highest wind speeds (Fig. 3). The air was (super)saturated with respect to ice most of the time ($RH_{ice}$ >100%), with increased frequency of subsaturation ($RH_{ice}$ <100%) when the winds were from SE (Fig. 3d). Horizontal wind speed at 2 and 39 m correlated well with friction velocity $u_*$ (R=0.89, p<0.05) indicating a well-mixed near-neutral turbulent boundary layer above the sea ice during blowing snow.

Airborne snow particle concentrations at 29 m showed strong variability and a weak but significant correlation

with wind speed (R=0.24, p<0.05) (Fig. 2a,c). During most storms measured snow particle concentrations have contributions from both blowing snow from the sea ice and precipitation from above as further discussed in section 3.3.2. At 29 m mean total number densities $N_{46-478}$ were 8.7×10$^3$ m$^{-3}$ during ANT-XXIX/6 and very similar 7.2×10$^3$ m$^{-3}$ during ANT-XXIX/7. Near the sea ice surface snow particles were measured for a total sampling time of 6 days and showed a mean $N_{46-478}$ of 2.6×10$^5$ m$^{-3}$, on average ~60 times the number density observed during

the same time at 29 m.

Aerosol concentrations at 29 m showed strong variability with many but not all increases associated with storms and blowing snow (Fig. 2d). At 29 m mean total number densities $N_{0.4-12}$ were 2.1×10$^6$ m$^{-3}$ during ANT-XXIX/6 (Fig. 2d). $N_{0.4-12}$ mean values at 2.0 and 0.2 m during ice stations were 1.4×10$^6$ and 1.7×10$^6$ m$^{-3}$, respectively, about



the same as the number densities observed during the same time at 29 m. The median aerosol particle diameters $\overline{d_p}$ at the measurement heights 0.2, 2.0m and 29 m ranged between 0.60 and 0.66 µm showing dominance of sub-micron sized particles in atmospheric aerosol below the instrument particle size cut-off (>11 µm).

Descriptive statistics of the aerosol chemistry are summarised in Table 4. The $Na^+$ concentrations of bulk aerosol showed strong variability with most increases coinciding with storms and aerosol number density $N_{0.4-12}$ peaks (Fig. 2e). $Na^+$ and sea salt concentrations were in general higher at 29 m than at 2 m (Table 4) except during S8 (Fig. 2e). $DF_{SO_4^{2-}}$ values of aerosol at 29 and 2 m varied most of the time between values near 0 (small depletion) and close to 1 (strong depletion), with occasional $DF_{SO_4^{2-}} < 0$ indicating enrichment (Fig. 2e).

Median $DF_{SO_4^{2-}}$ values at 29 m were very similar during ANT-XXIX/6 (=0.34) and ANT-XXIX/7 (=0.30), but larger near the sea ice surface (=0.49), suggesting throughout a significant contribution to the total SSA burden from a fractionated sea ice source (Table 4). Conversely, when in open water at the beginning and end of ANT-XXIX/6 $Na^+$ concentrations of bulk aerosol were relatively large but with small or no sulphate depletion consistent with the open ocean as the main SSA source, except for the very first and last sample (Fig. 2e).

## 3.2 Blowing snow events

Drifting and blowing snow events were frequently observed during ANTXXIX-6 (Fig. 2c), of which seven occurred during the 14 days of total time spent at ice stations. Based on data coverage two 7-10 day long periods were chosen to discuss key features of observed blowing snow and associated SSA increases.

### 3.2.1 Period 23 June to 3 July 2013

The period 23 June to 3 July featured a major storm with 4 wind speed maxima between 15 and 25 m s$^{-1}$ centred around midnight of the 24, 25, 27 and 28 June (Fig. 4a). Temperature $T_a$ increased during that time from -20 °C on 23 June to -3 °C on 26 June before dropping again to -20 °C on 30 June (Fig. 2b).

Wind speed $U_{39}$ increased from <5 m s$^{-1}$ in the late evening of 23 June over a 24 hr period to >20 m s$^{-1}$ (Fig. 4a). First snow particles were detected at 29 m once $U_{39}$ exceeded a threshold of ∼9 m s$^{-1}$ with spectral number densities $N_{46-478}$ reaching values on the order of $10^4$ m$^{-3}$ including large snow particles with $d_p$ >300 µm (Fig. 4b). Near-surface $N_{46-478}$ measured at ice station S2 on 24 June (Table 1) showed very large number densities on the order of $10^6$ m$^{-3}$ across the entire particle size spectrum confirming blowing snow from the sea ice surface as the main particle source (Fig. 4c). A decrease in wind speed $U$ always concurred with a drop in snow particle number densities at 29 m and often also in $RH_{ice}$ (25 June, 26 and 28 June) indicative of air being subsaturated with respect to ice (Fig. 4a-b).

Aerosol at 29 m showed background spectral number densities $N_{0.4-12}$ of $10^5$ m$^{-3}$ for particles with $d_p$ <2 µm during the calm periods on 23 June, 30 June and 2 July ($U_{39m}$ <5 m$^{-1}$), when no blowing snow or precipitation was present (Fig. 4d). During or after the 4 wind speed maxima on 24, 25, 27 and 28 June large increases of aerosol spectral number densities were observed especially of sub-micron sized particles reaching up to $10^7$ m$^{-3}$, often at





$RH_{ice}$ <100% consistent with a source from sublimating snow particles (Fig. 4a,d). Near the surface spectral number densities $N_{0.4-12}$ for particles with $d_p$ <2 μm during the storm on 24 June remained with $10^5$ m$^{-3}$ below those seen at 29 m (Fig. 4e) likely due to scavenging of aerosol by snow particles.

Na$^+$ concentrations in aerosol at 29 m show strong increases from background values of ~0.1 μg m$^{-3}$ to 1.4 μg m$^{-3}$ on 25 June and to 1.1–1.7 μg m$^{-3}$ during 28–29 June coinciding with wind speed maxima and increased number densities of aerosol and snow particles (Fig. 4). Na$^+$ concentrations returned to background values after the storm on 30 June coinciding with low wind speeds $U_{39m}$ <5 m s$^{-1}$, reduction in aerosol number concentrations and absence of any airborne snow particles (Fig. 4). Na$^+$ concentrations also dropped to background values during the storm on 26-27 June, consistent with the absence of blowing snow particles due to wind speed falling below the drift threshold (discussed in section 3.3.1) and a concurrent slight decrease in aerosol concentrations (Fig. 4).

During the calm periods on 24 June, 26–27 June and 29 June $DF_{SO_4^{2-}}$ of aerosol was negative or near 0, while during the storm episodes aerosol became increasingly more depleted in sulphate with $DF_{SO_4^{2-}}$ maxima of 0.53-0.59 coinciding with peaks in aerosol Na$^+$ concentration (Fig. 4f). In order to constrain the origin of the observed aerosol three snow pit profiles were sampled at ice station S2 on 24 June. The mean snow depth was 33 cm, and bulk mean Na$^+$ concentrations were 50, 6 and 17 μg g$^{-1}$ and bulk median $DF_{SO_4^{2-}}$ values were 0.30, 0.24 and 0.48, respectively. Assuming a linear mixing model and that measured snow chemistry at S2 is representative of the regional snowpack the $DF_{SO_4^{2-}}$ in aerosol and snow suggest that up to 80-90% of the sea salt aerosol observed during the storm originates from snow on sea ice. Negative $DF_{SO_4^{2-}}$ in aerosol during calm periods may arise either from contamination by ship's engine exhaust or contributions of non-local nss-SO$_4$$^{2-}$. The local snowpack near ice station S2 was less likely a contributing source as only one snow pit showed some snow layers with $DF_{SO_4^{2-}}$ <0.

### 3.2.2 Period 10 to 16 July 2013

From 10 to 16 July 2013, prior and during ice station S6 (Table 1), three snow drift episodes on 11, 12 and 14 July 2013, and a major blowing snow event from 14 to 16 July 2013 concurrent with strong warming were observed (Fig. 5). Starting on 14 July ambient temperature $T_a$ rapidly increased over a 12-hr period from ~-22 to -1 °C (Fig. 2b).

During the snow drift episodes wind speed at 39 m $U_{39m}$ peaked at 7–10 m s$^{-1}$ (Fig. 5a), snow particle spectral number densities $N_{46-478}$ reached up to $10^6$ m$^{-3}$ near the surface, but remained relatively low at 29 m with $10^2$–$10^3$ m$^{-3}$ with $d_p$ <100 μm except on 11 July (Fig. 5b-c). During the blowing snow event 14-16 July $U_{39m}$ ranged between 15 and 20 m s$^{-1}$ (Fig. 5a). Spectral number densities $N_{46-478}$ reached again up to $10^6$ m$^{-3}$ near the surface but $10^4$ m$^{-3}$ at 29 m for particle diameters $d_p$ 50-200 μm (Fig. 5b-c). The air within 2 m of the surface was supersaturated with respect to ice during drifting or blowing snow, but then became undersaturated towards the end of the snow drift episodes on 11 and 12 July as wind speed and snow particle concentrations decreased (Fig. 5a-c).

Aerosol at 29 and 2 m above the sea ice showed during the calm periods on 10, 12 and 14 July background spectral $N_{0.4-12}$ of $10^4$ to $10^5$ m$^{-3}$ for particle sizes $d_p$ <2 μm (Fig. 5d-e). During the snow drift episodes aerosol number





densities increased significantly especially of sub-micron sized particles at both measurement heights showing more particles near the surface, with spectral $N_{0.4-12}$ of up to $10^7$ m$^{-3}$ for $d_p <2$ µm (Fig. 5d-e). During blowing snow spectral number densities $N_{0.4-12}$ showed similar increases as during drifting snow, however at 29 m concentrations were higher and particles were larger ($d_p >9$ µm) than at 2 m (Fig. 5d-e). Similar to the observations from 24 June this

is consistent with net production of SSA taking place near the top of the layer containing suspended snow particles at $RH_{ice}<100\%$. Within the blowing snow layer SSA net production is suppressed due to saturated conditions and scavenging by snow particles which is efficient at warmer temperatures.

Na$^+$ concentrations in aerosol at 29 m showed strong increases from a background of 0.1 µg m$^{-3}$ to 1.8 µg m$^{-3}$ during the drift episode on 11 July, and to 1.2 µg m$^{-3}$ during the blowing snow event on 14-16 July (Fig. 5f). During

the same times Na$^+$ concentrations in aerosol at 2 m showed only small increases by 0.1 µg m$^{-3}$ (Fig. 5f). $DF_{SO_4^{2-}}$ of aerosol showed large scatter and mostly positive values, with the exception of 2 samples (15-16 July), varying between 0.1 and 1 (Fig. 5f, 6d). Median $DF_{SO_4^{2-}}$ values from 11 to 16 July were 0.61 at 29 m and 0.48 at 2 m confirming that SSA originates from a fractionated sea ice source.

A more detailed view of the blowing snow event from 14 to 16 July reveals trends and phasing of particle number

densities and chemical composition, which are consistent with the hypothesised sequence of processes from the onset of blowing snow to the release of sea salt aerosol due to snow sublimation (Fig. 6). Onset of snow drift at 0.1 m above the snow surface occurred once wind speed $U_{39m}$ exceeded a drift threshold of 12.1 m s$^{-1}$ followed about 1 hr later by detection of blowing snow at 29 m (Fig. 6b). Total snow particle number densities $N_{46-478}$ near the surface and at 29 m decreased again a few hours later despite constant wind speeds of 15 m s$^{-1}$ (Fig. 6b) due to strong warming

during the storm causing an increase in drift threshold wind speed and therefore less uplift of particles, as shown in section 3.3.1.

Two hours after the onset of snow drift total aerosol number densities $N_{0.4-12}$ started to gradually increase at 2 and 29 m to reach peak values first at 29 m after an initial spike, and with a delay of 2 hr also at 2 m coinciding with a decrease in snow particle concentrations (Fig. 6b-c). Despite $RH_{ice} >100\%$ within 2 m of the sea ice surface

during the blowing snow event on 15 July (Fig. 5a) sub-saturation cannot be ruled out because of the large error in $RH_{ice}$ near 0 °C. Thus, the observed anti-correlation between aerosol and snow particle number densities at both measurement heights is consistent with (a) increased aerosol production from snow particle sublimation near the top or above the blowing snow layer, where undersaturated conditions are more likely, and (b) reduced aerosol scavenging by snow particles.

Na$^+$ concentrations in aerosol followed the observed total $N_{0.4-12}$ showing first a maximum at 29 m and then near the surface at 2 m albeit of smaller magnitude (Fig. 6d) suggesting that much of the observed increase of aerosol particles consists of sea salt aerosol. Non-zero $DF_{SO_4^{2-}}$ in aerosol suggest that the sea salt particles originate from a fractionated sea ice source.

In order to further constrain the origin of the aerosol Na$^+$ and $DF_{SO_4^{2-}}$ of the snowpack were measured. Blowing

snow was collected during the storm every two hours over a 12 hr period and snow pit profiles were sampled at various





ice floe locations from 11 to 14 July prior to the storm. During the storm blowing snow $Na^+$ concentrations on average more than doubled from 4.5 to $12\,\mu g\,g^{-1}$ and salinities $S_p$ increased from 0.02 to 0.04 psu, whereas $DF_{SO_4^{2-}}$ remained fairly constant at a median of $\sim 0.54$ (Fig. 6e-f). The large spatial variability in $Na^+$ concentrations and $S_p$ of the snowpack masked any temporal trend, yet the snowpack profiles could be grouped into two types, one showing a very

steep $S_p$ ($Na^+$) decline with snow layer height from 1–10 psu ($10^2\,\mu g\,g^{-1}$) near the bottom to $10^{-3}$ psu ($10^{-1}\,\mu g\,g^{-1}$) near the surface, and the other showing a less steep decline from 30 psu ($10^3\,\mu g\,g^{-1}$) to $10^{-2}$ psu ($10^1\,\mu g\,g^{-1}$) (Fig. 7a-b). $DF_{SO_4^{2-}}$ showed large scatter, with the more saline snow being more depleted in sulphate (Fig. 7c). Negative $DF_{SO_4^{2-}}$ was found at the snow-ice interface and in some snow layers indicating the presence of mirabilite (Fig. 7c).

Comparison shows that the range of $Na^+$ concentrations and $S_p$ values observed in blowing snow can be explained

by mixing between the two snowpack types (Fig. 7a-b). An increase over time of salinity or $Na^+$ concentration in airborne snow is also consistent with snowpack erosion during the storm and exposure to uplift of deeper and more saline snow layers. The $DF_{SO_4^{2-}}$ values of blowing snow were at the top end of the range observed in the more saline snowpack (Fig. 7c). Taking the median $DF_{SO_4^{2-}}$ values in aerosol (=0.48-0.61) and blowing snow (=0.54) it can be concluded that 89-100% of the observed aerosol comes from the local fractionated snow source.

In summary, observations show that blowing or drifting snow leads to increases in aerosol number densities during and after a storm. Spectral $N_{0.4-12}$ increased during individual storms by 2-3 orders of magnitude above background levels for particle sizes $d_p < 2\,\mu m$. Concurrent increases in aerosol $Na^+$ concentration show that the observed new particles consist mainly of sea salt aerosol. Observed aerosol gradients suggest that net production of SSA takes place near the top of the blowing or drifting snow layer. Furthermore, similar sulphate depletion in aerosol, blowing

snow and the local snowpack is strong evidence that the bulk of the observed SSA originated from snow on sea ice and not the open ocean, which is consistent with the independent model results of Yang et al. (2018). Advection of SSA from the open ocean cannot be ruled out during storms with warm and moist air, but plays only a minor role due to the large distance to the nearest open water, in particular at ice stations S1-S6 (Table 1). The scale-length for SSA removal over an ice shelf had been estimated previously to be $\sim 30\%$ per 100 km, thus the reduction of the SSA

burden during advection over 600 to 1000 km would be larger than a factor 6 to 20 (Wagenbach et al., 1998, and refs. therein). Distance to the open ocean together with a temporally very close association of snow particle and aerosol number density dynamics (Fig. 6) further support the above finding that a local SSA sea ice source dominates over advection. Observations show also that snow drift is conceptually not different from a blowing snow event with the same physical processes leading to SSA production.

## 3.3   Evaluation of SSA production scheme from blowing snow

It has been proposed that the production or mass flux of SSA from blowing snow, $Q_{SSA}$, is proportional to the bulk sublimation flux of suspended snow particles, $Q_s$ in units of $kg\,m^{-2}s^{-1}$, and snow salinity, $S_p$ in psu (Yang et al., 2008, 2018). The model scheme however relies on blowing snow measurements above ice sheets (Budd, 1966; Mann et al., 2000; Nishimura and Nemoto, 2005), and parameterisations developed for in-land regions in the high Arctic



(Déry and Yau, 2001, and ref. therein). Below we briefly summarise the model mechanism and then evaluate its applicability to snow on sea ice based on the Weddell Sea observations.

The mass flux of SSA from blowing snow $Q_{SSA}$ in units of $\mathrm{kg\,m^{-2}s^{-1}}$ is computed as

$$Q_{SSA} = \frac{Q_s}{1000} \int\limits_0^\infty \int\limits_0^\infty f(d_p)\, S_p\, \psi(S_p)\, d(d_p)\, d(S_p) \tag{1}$$

where $d_p$ is the snow particle diameter (m), $f(d_p)$ is the particle size distribution of blowing snow and $\psi(S_p)$ the snow salinity probability distribution. The quantities $f(d_p)$ and $\psi(S_p)$ are compared to direct observations from this study, whereas the blowing snow bulk sublimation flux $Q_s$ was not amenable to direct measurement under field conditions. However, the model parameterisation used for $Q_s$ (Déry and Yau, 2001) depends on observable quantities, which are validated against measurements from this study.

### 3.3.1    Blowing snow bulk sublimation flux $Q_s$

The blowing snow bulk sublimation flux $Q_s$ is the local bulk sublimation rate integrated over the entire blowing snow column. $Q_s$ is parameterised after Déry and Yau (1999, 2001),

$$Q_s = K\, A'\, Q_s'\, \frac{q_{bsalt}}{q_{b0}} \tag{2}$$

where $K$ ($= 1.1574 \times 10^{-5}$) is a factor to convert $Q_s'$ into units of $\mathrm{kg\,m^{-2}\,s^{-1}}$, $A'$ is an empirical snow age factor,
$Q_s'$ is a normalised column-integrated sublimation rate ($\mathrm{mm\,day^{-1}, SWE}$), $q_{bsalt}$ is the saltation layer blowing snow mixing ratio ($\mathrm{kg\,kg^{-1}}$) under ambient conditions, and $q_{b0}$ the value it would have when the 10-m threshold wind speed $U_t$, below which no snow drift occurs, has its minimum value $U_{t0}$ ($= 6.975\,\mathrm{m\,s^{-1}}$) based on the empirical model of Li and Pomeroy (1997). The ratio $q_{bsalt}/q_{b0}$ comes from the required introduction of a lower boundary condition for particle number densities in the saltation layer and scales the normalised $Q_s'$ accordingly. Values for $q_{bsalt}$ are
computed with

$$q_{bsalt} = 0.385\,(1 - \frac{U_t}{U_{10m}})^{2.59} u_*^{-1} \tag{3}$$

where $U_{10m}$ is the 10-m wind speed ($\mathrm{m\,s^{-1}}$) and $u_*$ is the friction velocity (Eq.24 in Déry and Yau, 1999). For the 10-m threshold wind speed $U_t$ an empirical model for dry snow was previously derived based on observations in the prairies of Western Canada, which applies to dry snow conditions and is a function of ambient temperature $T_a$ (Li
and Pomeroy, 1997)

$$U_t = U_{t0} + 0.0033\,(T_a + 27.27)^2 \tag{4}$$





This expression is compared to observations above sea ice from this study as follows. The onset of drifting or blowing snow is defined similar to Mann et al. (2000) as the moment when snow drift density $\mu$ right above the snow surface exceeds a threshold value of $\mu_c = 0.005$ (0.001) $\mathrm{kg\,m^{-3}}$. The snow drift density is calculated as $\mu = \frac{4}{3}\pi\rho_{ice}\int_0^\infty N_s(d_p)(\frac{d_p}{2})^3\,d\,d_p$ with the density of ice $\rho_{ice}$ ($= 917\,\mathrm{kg\,m^{-3}}$) and the measured spectral number density $N_s$. $U_{10m}$ is extrapolated from the sonic anemometer measurements at 2 m assuming a logarithmic wind profile and $T_a$ comes from the uppermost sensor on the sea ice at $\sim$2 m. Using the lower $\mu$ threshold most observed $U_t$ values are still within the model range of uncertainty (Fig. 8) and show a mean $U_t$ of 7.1 (range 2.2-9.8) $\mathrm{m\,s^{-1}}$ corresponding to $u_{*t}$ of 0.37 (range 0.08 - 0.58) $\mathrm{m\,s^{-1}}$. For comparison, $u_{*t}$ for blowing snow above an Antarctic ice shelf during winter was observed to range between 0.18 and 0.38 $\mathrm{m\,s^{-1}}$ (Dover, 1993; Mann et al., 2000). Earlier work on land based snowpacks found $u_{*t}$ to range from 0.15 $\mathrm{m\,s^{-1}}$ for loose, fresh, dry snow to 0.4 $\mathrm{m\,s^{-1}}$ for old, wind-hardened snow (Dover, 1993, and references therein). Thus, $u_{*t}$ observed for snow on sea ice are on average at the upper end of previous observations for snow on land or ice shelves. It is concluded that Eq. 4 provides robust estimates of $U_t$ above sea ice for the snowpack conditions encountered during this study exhibiting in general very low salinities (Table 5). However, temperature alone may not be a good predictor of $U_t$ if surface snow on sea ice exposed to wind is very saline with a relatively larger liquid fraction at a given temperature due to freezing point depression leading to increased snow crystal cohesion.

In a next step blowing snow mixing ratios in the saltation layer $q_{bsalt}$ were computed by dividing observed drift density $\mu$ by the density of air and then compared to $q_{bsalt}$ predicted by the empirical parameterisation after Déry and Yau (1999) (Eq. 3). Considered were only observations when $\mu > 0.001\,\mathrm{kg\,m^{-3}}$ and when the SPC was mounted below reported values of the saltation layer depth ($=0.1$ m). It is found that the model overpredicts observed $q_{bsalt}$ on average by at least a factor $\sim$10 (Fig. 9) suggesting that the conditions during this study were likely different to those in the Canadian Prairie for which Eq. 3 was developed. One model uncertainty is the assumption that the saltation layer is saturated with respect to ice, which has not been confirmed yet for the dry and cold polar boundary layer due to the lack of sufficiently accurate humidity measurements.

However, relative changes of $q_{bsalt}$ are well captured by the model, in general showing an increase with wind speed (Fig. 9a,c). Except during the blowing snow event on 14-15 July (ice station S6) $q_{bsalt}$ decreased even though wind speed remained constant at $\sim$12 $\mathrm{m\,s^{-1}}$ (Fig. 9b), as noted above for total $N_{46-478}$ near the snow surface (Fig. 6b). Strong warming during the storm of $\sim$20 K increased snow particle cohesion and therefore threshold wind speed $U_t$ as predicted by the model, thereby reducing the uplift of snow particles (Fig. 9b). A similar case occurred during 26-27 June, when warming increased the theoretical drift threshold and wind speed dropped at the same time below $U_t$ consistent with the absence of blowing snow particles at 29 m (Fig. 4b).

The model bias in $q_{bsalt}$ has only a minor impact on estimates of bulk sublimation rate $Q_s$ (Eq. 2) and therefore also of SSA production $Q_{SSA}$ (Eq. 1) because the calculation uses not absolute values but ratios of actual $q_{bsalt}$ and maximum $q_{b0}$.





### 3.3.2 Snow particle size distribution $f(d_p)$ above sea ice

Previous studies concluded that a 2-parameter gamma probability density function gives a reasonable fit to observed distributions of snow particle diameter $f(d_p)$ (Budd, 1966; Schmidt, 1982; Dover, 1993):

$$f(d_p) = \frac{e^{-\frac{d_p}{\beta}} d_p^{\alpha-1}}{\beta^\alpha \Gamma(\alpha)} \qquad (5)$$

with shape and scale parameter $\alpha$ and $\beta$, and mean particle diameter $\overline{d_p} = \alpha\beta$. Particle size distributions of blowing snow have been found to vary with wind speed and height: at a given height, $\overline{d_p}$ increases with wind speed, whereas at a given wind speed, preferential removal of large snow particles due to gravitational settling leads to a decrease in $\overline{d_p}$ (e.g. Dover, 1993; Mann et al., 2000) and an increase in $\alpha$ with height (Nishimura and Nemoto, 2005). Increasing $\alpha$ is equivalent to a shift to more symmetrical size distributions. Model predictions of the mass flux of SSA from

blowing snow $Q_{SSA}$ depend critically on $f(d_p)$ (Eq. 1) and therefore on $\alpha$ and $\beta$ (Eq. 5).

     In this study we derived $\overline{d_p}$ and $\alpha$ by fitting a 2-parameter gamma distribution to observed particle size spectra. Confidence in retrieved values of $\alpha$ and $\beta$ is affected by two limitations of the SPC measurements. Firstly, SPC size spectra are clipped due to instrument configuration and exclude very small snow particles ($d_p < 46\,\mu\text{m}$). And secondly, SPC data include noise in the particle number count due to natural variability. A Monte Carlo model, which

generates multiple noisy gamma distributions of known $\alpha$ and $\beta$ shows that retrieving these parameters is robust, even when the data are limited to $d_p > \overline{d_p}$, that is capture only the tail of the data. Furthermore, the uncertainty from under-sampling and data noise in $\alpha$ and $\beta$ can be retrieved from the relative scale of the smallest size bin and the residual noise.

     During blowing snow episodes when $U_{10m}$ was above the mean observed threshold wind speed $U_t$ of $7.1\,\text{m s}^{-1}$ $\overline{d_p}$

decreased with height, whereas $\alpha$ increased with height as shown by the respective median values (Table 6) consistent with previous observations. Increase of $\alpha$ with height during blowing snow was found previously at Mizuho station on the Antarctic Plateau where $\alpha$ was ~3 near the surface and increased to values >10 further aloft (Nishimura and Nemoto, 2005). Another study at Halley in coastal Antarctica found no gradient in $\alpha$ within the lower 4 m (Dover, 1993), which was set to 2 following the analysis of King et al. (1996).

However, the observed decrease in mean particle size $\overline{d_p}$ with height was much smaller than that inferred from previous studies. For example, median $\overline{d_p}$ at 29 m was 43% larger than $d_p$ observed between 1.0 and 9.6 m ($=80\,\mu\text{m}$ regardless of wind speed) at Mizuho station (Nishimura and Nemoto, 2005) and ~4 times the $d_p$ predicted by a parameterisation based on the Halley vertical $d_p$ profiles between 0.1 and 4.0 m (King et al., 2001). Further examination of the snow particle size distributions at 29 m reveals in general an unexpected long tail indicating

significant relative contributions from large particles. For example, during the blowing snow event on 14 July $\overline{d_p}$ and $\alpha$ near the surface were $105\,\mu\text{m}$ and 5.2, respectively (Fig. 10b). At 29 m total $N_{46-478}$ was as expected about two orders of magnitude smaller than near the surface. Yet compared to the near-surface values average $\overline{d_p}$ was larger



(131 μm) and the size distribution more skewed ($\alpha$= 3.2) with a long tail showing significant contributions from particles with $d_p$ >200 μm (Fig. 10a).

It is suggested that small or no decrease of $\overline{d_p}$ with height and the long tail in $f(d_p)$ at 29 m can be explained by contributions from falling snow particles since blowing snow frequently coincided with precipitation. In order to analyse $f(d_p)$ of snow precipitation only, a calm 17 h period (3 July 14:30 to 4 July 7:00) period) was chosen during which light snowfall ($N_{46-478}$<800 m$^{-3}$ at both levels) but no snow drift ($U_{10m}$=2.7 m s$^{-1}$) occurred. It is found that the size distribution ($\overline{d_p}$, $\alpha$) of precipitation at 29 m (Fig. 10c) was very similar to that during a blowing snow event (Fig. 10a), and also showed no preferential large size mode as observed at Mizuho station (Nishimura and Nemoto, 2005) preventing a correction of blowing snow particle spectra for snow precipitation based on particle size. The spectral shift to smaller particles near the surface during the snowfall event (Fig. 10d) was likely due to a combination of two processes: sublimation of larger snow particles supported by a mean $RH_{ice}$ of 98.0 % between 0.2 and 2.0 m. And fragmentation of snow particles by more vigorous and smaller eddies near the surface during calm conditions, either through direct break-up of the dentrites or through separation of snow flakes that have have loosely adhered on the decent. Similar size distributions ($\overline{d_p}$, $\alpha$) near the snow surface and at 29 m during blowing snow imply that at 29 m particle sizes lost from the blowing snow size spectrum due to gravitational settling were being replenished by falling snow. Due to the increase in the fraction of precipitating snow with height snow particle size distributions measured at 29 m have additional uncertainty, whereas those observed near the surface are representative of blowing snow.

Plotting $\overline{d_p}$ and $\alpha$ against wind speed allows to evaluate the parameter range characteristic for blowing snow: $\overline{d_p}$ shows the expected increase near the surface but large scatter and little change at 29 m (Fig. 11a-b). Values of $\alpha$ show at both heights some dependence on wind speed (Fig. 11c-d) and cluster between 3 and 6 when $U_{10m} > U_t$. Median, lower and upper quartiles of $\overline{d_p}$ and $\alpha$ for blowing snow, i.e. when windspeed $U_{10m}$ is above the observed mean threshold $U_t$ (=7.1 m s$^{-1}$), are summarised in Table 6 as guidance for model sensitivity studies.

### 3.3.3 Salinity probability distribution $\psi(S_p)$ of snow on sea ice

Correct modelling of SSA production from blowing snow (Eq. 1) requires choosing a snow salinity probability distribution $\psi(S_p)$ representative of the region and season under consideration (Massom et al., 2001). High salinities near the snow-sea ice interface originate from incorporation of frost flowers or upwards migration of brine due to capillary action. Brine is found at the top of sea ice as a result of ion exclusion during the freezing of seawater or of flooding due to negative freeboard. Top snow layers in deeper snowpacks not affected by brine migration receive sea salt aerosol through dry or wet deposition of background aerosol or sea spray.

Vertical profiles of snow salinity $S_p$ measured during this study (Frey, 2017) showed a marked decrease with height above the sea ice surface (Fig. 12a) in agreement with previous sea ice surveys in Antarctica (Massom et al., 2001). High salinities $S_p$ (1–43 psu) were found in basal snow, the sea ice surface and frost flowers, whereas very low salinities (<0.2 psu) were observed in snow layers >20 cm above the snow-ice interface and in blowing snow (range





0.01–0.12 psu) (Fig. 12a). The inferred mean migration distance of ∼20 cm above sea ice that brine can affect the snowpack falls within the range of those reported by previous studies, up to 17 cm in the Arctic (Domine et al., 2004), and 30 cm in the Weddell sea (Massom et al., 2001).

From the above follows that to a first order the distance of individual snow layers to the sea ice surface determines
their salinity, and therefore snowpack depth determines to a first order mean volume-integrated salinity and salinity probability distributions $\psi(S_p)$ in snow on sea ice. Average snowpack depth based on snow pits from this study was 21 cm above first-year sea ice (FYI) and 50 cm above multi-year sea ice (MYI) (Table 5), in the range of previous observations in the Weddell sea sector, e.g. mean snowpack depth during winter 1992 was 14 cm (Massom et al., 2001, and references therein). As expected the mean salinity of the shallow FYI snowpack (=1.4 psu) was larger
than that of the deep MYI snowpack (0.82 psu) (Table 5), and corresponding salinity probability distributions $\psi(S_p)$ for snow on FYI are shifted to higher salinities when compared to $\psi(S_p)$ above MYI (Fig. 12b). Secondary factors, which contribute to higher salinities of snow on FYI in comparison to snow on MYI are flooding of thin FYI with sea water due to negative freeboard and brine rejection and drainage, which decreases MYI salinity over time.

Previous modelling of SSA production from blowing snow based $\psi(S_p)$ on much higher mean snow salinities $S_p$
than observed during this study: 8.5 psu from observations in the Indian ocean sector in August 1995 (Yang et al., 2008) and half that value in a later study (Levine et al., 2014). Bearing in mind that only surface snow, which often has lower density and crystal cohesion, is likely to get airborne, it is sensible to include only the top snow layers in the calculation of average snow salinity. For example, in this study the mean $S_p$ of snow layers within the top 10 cm of the snow surface was 0.31 psu (Table 5), a factor 28 smaller than that used for $\psi(S_p)$ by Yang et al. (2008). More
recent studies updated the model scheme accordingly using the very low salinities found here in surface and blowing snow (Rhodes et al., 2017; Yang et al., 2018). It should be noted that snow on sea ice even at the lower end of the $S_p$ spectrum contains amounts of sea salt large enough to produce significant SSA especially in the sub-micron range (e.g. Yang et al., 2018), and is on average significantly saltier than snow in inner Antarctica. For example, mean ice core $Cl^-$ or $Na^+$ concentrations on the West Antarctic ice sheet are ∼100 ng g$^{-1}$ or less (Frey et al., 2006),
equivalent to $S_p$ of ∼$10^{-3}$ psu, thus up to 4 orders of magnitude smaller than in snow on sea ice (Fig. 12a).

In summary, model parameterisations for SSA production from blowing snow require a snow salinity probability distribution $\psi(S_p)$, which takes into account regional and seasonal variabilities. Snow depth on sea ice, retrievable from satellite measurements, may be a good predictor of mean snow salinity as more field measurements become available.

3.4   The sea ice source of sea salt aerosol

Below we discuss chemical fractionation in snow and aerosol, aerosol spectra and the air-snow sea salt budget characteristic for sea ice.





### 3.4.1   Chemical fractionation of $\mathbf{SO_4^{2-}}$

In non-summer months when biogenic non-ss $SO_4^{2-}$ is at a minimum Antarctic aerosol was observed to be depleted in $SO_4^{2-}$ (with respect to $Na^+$) and in $Na^+$ (with respect to $Cl^-$) (Wagenbach et al., 1998; Legrand et al., 2017). Similar fractionation in brine and frost flowers in the sea ice nearby attributed to precipitation of the mineral mirabilite

$(Na_2SO_4 \cdot 10H_2O)$ suggested that the observed SSA must originate from the sea ice and not from the open ocean (Wagenbach et al., 1998; Rankin et al., 2000).

Here we show for the first time that snow on sea ice is depleted in sulphate (Fig. 13a). $DF_{SO_4^{2-}}$ of most snow samples was positive, showing large scatter and no particular dependence on snowpack depth or sea ice age (Fig. 13a). Negative $DF_{SO_4^{2-}}$ suggests presence of mirabilite precipitated during freezing. The median volume-integrated $DF_{SO_4^{2-}}$ in snow

ranged between 0.24 and 0.35 (Table 5) matching very well the sulphate depletion found in aerosol during the period the snow was sampled (ANT6) with median $DF_{SO_4^{2-}}$ ranging between 0.29 and 0.48 (Table 4). The similarity of sulphate depletion in both snow and aerosol strongly suggests that snow sea ice in the Weddell Sea is the dominant source of regional sea salt aerosol.

The sulphate depletion observed in aerosol and snow (Figs. 2f, 13a) is consistent with mirabilite precipitation. Mean

temperatures during all ice stations except S3 were below the -6.4 °C threshold of mirabilite precipitation(Table 1). And a recent lab study found that mirabilite precipitation results in $DF_{SO_4^{2-}}$ of 0.93 in sea water brine cooled down to -20.6 °C (Butler et al., 2016). Conversely, $DF_{SO_4^{2-}} < 0$ in snow or aerosol suggests contributions from non-sea salt $SO_4^{2-}$ or crystallized mirabilite, which has a $DF_{SO_4^{2-}}$ of -7.3. The latter is supported by the observation that most $DF_{SO_4^{2-}} < 0$ occur in or near the sea ice surface, where most of the partitioning between brine and mineral is

expected to have occurred during freezing (Fig. 13a).

If the partitioning of crystallised $Na_2SO_4 \cdot 10H_2O$ between brine and the snow-ice matrix is the dominating $SO_4^{2-}$ fractionation process then $Na^+$ should be depleted as well in the brine. Following the analysis of Wagenbach et al. (1998) we take $Cl^-$ as the reference species for bulk sea water and find from a mass balance calculation that the $Cl^-$ to $Na^+$ mass ratio would then be linearly related to the $SO_4^{2-}$ to $Na^+$ ratio with a slope of -0.98 and an intercept of

2.04. From that a theoretical relationship between $DF_{Na^+}$ (with respect to $Cl^-$) and $DF_{SO_4^{2-}}$ (with respect to $Na^+$) is derived and compared to observations (Fig. 14). The observed fractionation is largely in agreement with the model prediction. Snow on sea ice follows closely the theoretical mirabilite fractionation line, whereas aerosol shows large scatter and a tendency to apparent $Na^+$ enrichment (Fig. 14). The latter was observed previously and attributed to acid-induced chloride loss, which had occurred from aerosol in the atmosphere or as a sampling artefact from sea

salt already accumulated on the filter surface (Wagenbach et al., 1998; Legrand et al., 2017). It is noted that if all sulphate is removed by mirabilite precipitation ($DF_{SO_4^{2-}} = 1$) then sodium depletion reaches a theoretical maximum of ~12% ($DF_{Na^+} = 1.1204$) (Fig. 14).

The median $DF_{SO_4^{2-}}$ values of bulk aerosol observed during this study were smaller than previous winter observations at coastal Antarctic sites, where the sea-ice surface was shown to be the dominant source of sea-salt aerosol.





Wagenbach et al. (1998) reported sulphate depletion in aerosol corresponding to $DF_{SO_4^{2-}}$ of 0.72 at Halley (75°S, 26°W), of 0.76 at Neumayer (70°S, 85°W), and of 0.62 at Dumont d'Urville (66°S, 140°E). The main cause of smaller $DF_{SO_4^{2-}}$ observed in aerosol above the Weddell Sea in winter 2013 appears to be the snow source being less depleted in sulphate during that particular year. Thus, caution is warranted when sea ice SSA fractions of the total aerosol burden at a specific location are estimated based on sulphate depletion (e.g. Legrand et al., 2017), since $DF_{SO_4^{2-}}$ of pure sea ice emissions vary temporally and spatially thereby introducing additional uncertainty if not accounted for.

### 3.4.2  Chemical fractionation of $Br^-$

The importance of sea salt as a source of atmospheric bromine species in the mid to high southern latitudes is now well established, and SSA from blowing snow is expected to release bromine (Yang et al., 2008, 2010) driving ozone depletion events observed during or after snow storms (Jones et al., 2009). One of a number of processes identified to cause halogen release from aerosol or ice surfaces involves the reaction of HOBr with halides ($Br^-$ and $Cl^-$) on acidic ice surfaces forming BrCl and $Br_2$ that are subsequently photolyzed to form reactive halogen atoms (Abbatt et al., 2012, and references therein). This is a multiphase process which, depending upon the substrate, may or may not proceed faster under acidic conditions (Abbatt et al., 2012), and associated uncertainties are currently still significantly limiting the modelling of impacts on tropospheric halogen chemistry (Sander et al., 2003; Long et al., 2014). Multi-phase bromine chemistry models use observed sea salt aerosol depletion of bromide relative to sodium (or chloride) with respect to seawater composition, i.e. bromide depletion factors $DF_{Br^-}$, to estimate the bromine flux from sea-salt aerosol (e.g. Yang et al., 2008; Long et al., 2014). However, to date observations of bromide aerosol depletion are only available from north of 55°S (Sander et al., 2003, and references therein), with the exception of two recent pioneering studies in Antarctica (Legrand et al., 2016; Hara et al., 2018). Below we discuss bromine chemistry observations above the Weddell Sea during this study.

Bulk aerosol concentrations of bromide ranged between <1 and 19 ng m$^{-3}$ (Fig. 15a), showing occasional large increases to levels seen elsewhere in coastal Antarctica only during summer (Legrand et al., 2016). However, median bulk aerosol concentrations of $Br^-$ (Table 4) were similar to coastal observations at Dumont d'Urville during winter (Legrand et al., 2016). It should be borne in mind that the bromide aerosol concentrations were often below the estimated LOD due to the short filter exposure times employed (Fig. 15a). Median bromide concentrations in snow ranged between 70 and 180 ng m$^{-3}$ (Table 5).

It is found that aerosol at 29 m was on average strongly depleted in bromide relative to sodium with respect to seawater, whereas aerosol near the sea ice surface at 2 m showed mostly strong enrichment (Table 4, Fig. 15b). Median $DF_{Br^-}$ in the snowpack was 0.00–0.06 suggesting overall no or small depletion in bromide (Table 5). However, examination of individual snow layers as well as blowing snow samples shows large scatter with both positive and negative $DF_{Br^-}$, suggesting both bromide depletion and enrichment (Fig. 13b). The sea ice surface below the snow was with the exception of one sample always enriched in bromide (Fig. 13b). During the storm on 14-15 July (see





section 3.2.2) it is found that the median $DF_{Br^-}$ in blowing snow collected on 15 July at $<1\,\mathrm{m}$ was close to zero but 0.39 in aerosol at $29\,\mathrm{m}$ sampled over the same time interval.

These observations are in agreement with previous studies in coastal Antarctica: Legrand et al. (2016) were the first to report bromide depletion in aerosol at Dumont D'Urville. And Hara et al. (2018) observed at Syowa
Station bromide depletion in aerosol but not in blowing snow, which was enriched, and concluded that heterogeneous chemistry driving bromine activation occurs on sea salt aerosol and not on blowing snow. Lieb-Lappen and Obbard (2015) observed bromide depletion in blowing snow but only at $>5\,\mathrm{m}$ above the surface.

From this study it is concluded that air-ice recycling of bromine occurs on surface snow, airborne snow and aerosol particles. However, the vertical gradient in $DF_{Br^-\,(Na^+)}$ observed in aerosol suggests that the bulk of the net bromine
release must take place in the aerosol phase between 2 and $29\,\mathrm{m}$ above the sea ice surface. The bromine net release from large snow particles is expected to be reduced compared to aerosol due a combination of shorter atmospheric lifetime and smaller specific surface area limiting diffusional outgassing. The bromine release from SSA produced by blowing snow may be more efficient because it has a large fraction of sub-micron sized particles (see section 3.4.3), resides at the well ventilated top of the blowing snow layer, and may also be more acidic. Thus sea salt aerosol from
blowing snow provides an additional bromine reservoir, which is readily depleted.

Models assume a dependence of $DF_{Br^-}$ on season and aerosol diameter (Yang et al., 2008; Sander et al., 2003). $DF_{Br^-}$ of aerosol showed a weak positive trend from winter into spring largely due to less frequent negative values during August and September (Fig. 15b, Table 4). Increasing $DF_{Br^-}$ is consistent with enhanced bromine loss from aerosol as incident radiation increases concurrent with activation of the reactive bromine (BrO$_x$) cycle (Fig. 15b).
This becomes more evident considering monthly median $DF_{Br^-}$ values of aerosol at $29\,\mathrm{m}$, which increased from June (0.25), July (0.44), August (0.54) into September (0.59). Lower values but a similar trend had been observed at Dumont d'Urville, where $DF_{Br^-}$ in bulk aerosol increased gradually from a minimum in June (0.04), intermediate values in July to Sep (0.22-0.39) to a maximum in October (0.42) (Legrand et al., 2016). $DF_{Br^-}$ values from this study would be even higher had they been referenced to a sea water ratio adjusted for the maximum possible removal
of sodium due to precipitation of mirabilite as done by Legrand et al. (2016). Assuming that observed $DF_{Br^-}$ is representative of SSA with a median particle diameters $d_p$ of $\sim$0.60-0.66 $\mu$m, it is found that observed bromide depletion is larger than that reported by Sander et al. (2003) with a maximum of 0.4 for a $d_p$ of 1.3 $\mu$m, dropping to 0.1 at $10\,\mu$m.

### 3.4.3 Aerosol size distribution

Average aerosol number and volume distributions observed in the Weddell sea show that during calm conditions ($U_{10m}$ $3\pm1\,\mathrm{m\,s^{-1}}$) concentrations across most of the size spectrum were smaller above sea ice than above the open ocean (Fig. 16). The wind speed chosen for calm conditions is well below the mean snowdrift threshold $U_t$ of $7.1\,\mathrm{m\,s^{-1}}$ observed during this study and at the lower end of the range when breaking of waves commences (O'Dowd et al., 1997). A low sea salt aerosol background above compact sea ice during calm conditions is consistent with the absence





of any active local sources and the long distance to the nearest open ocean. The average aerosol volume distribution shows two modes, one at 1-2 µm and the other at $> 7$µm (Fig. 16b), indicating that most of the aerosol particle mass resides in the super-micron range as expected for sea salt aerosol.

During stormy conditions ($U_{10m}$ $10\pm1\,\mathrm{m\,s^{-1}}$) average aerosol number and volume concentrations above the sea ice increased significantly for all particle diameters $d_p$ below 10 µm, and slightly more so at smaller particle sizes ($d_p < 2$ µm), albeit remained below those above the open ocean observed over two days during this study (Fig. 16). However, the relative increase of aerosol number and volume concentrations during storms above background levels over the sea ice was equal or larger than above the open ocean. This implies that corresponding aerosol number and mass fluxes during storms were also equal or larger above sea ice than above the open ocean consistent with Yang et al. (2008) and underlining the importance of a sea ice source for atmospheric sea salt aerosol in winter and early spring (Fig. 16).

The main observational constraint for the origin of the observed sea salt aerosol comes from its chemical source signature as discussed above. In addition, the relatively larger increase of small particle number densities provides further support for the blowing snow SSA source based on physical arguments: snow particles of low salinity as observed during this study will generate very small sea salt particles after complete loss of water ice by sublimation. Taking the dry particle diameter $d_{dry} = d_p(S_p/1000\,\rho_{ice}/\rho_{\mathrm{NaCl}})^{1/3}$, where $\rho_{ice}$ and $\rho_{\mathrm{NaCl}}$ are densities of ice and NaCl, respectively, we would expect based on observed snow salinity and snow particle size (Table 5, 6) potential $d_{dry}$ on the order of $10^{-3}$µm, or larger if not all water is lost. Instruments with a lower size detection limit than available in this study are needed to further investigate very small particle formation from blowing snow.

It is important to note that the use of sea salt measured in ice cores as a proxy of past sea ice conditions requires that average emissions above sea ice exceed those above an equivalent area of open ocean and consequently lead to a comparatively larger burden of atmospheric SSA. Observations from this study support the former requirement, but not the latter. The low SSA background concentrations observed above sea ice may have been due to a combination of low storm frequency and low snow salinities in the area. Indeed, model calculations suggest that size distribution and associated flux of SSA originating from blowing snow are very sensitive to snow salinity $S_p$, e.g. decreasing $S_p$ from 0.92 to 0.06 psu causes SSA spectral number densities to decrease by about one order of magnitude (Yang et al., 2018). Further atmospheric modelling is needed to address this issue and will be subject of future research.

### 3.4.4 The sea salt atmosphere-snow mass budget

Assessing the sea salt mass distribution between atmosphere and snow during calm and blowing snow conditions provides further evidence for the importance of snow on sea ice as a SSA source (Fig. 17). To do this atmospheric sea salt concentrations were estimated in two ways for the time period when $RV\,Polarstern$ was well within the FYI zone (18 June to 21 July 2013; Fig. 1): one by multiplying the Na$^+$ concentration measured on aerosol filters by 3.262 based on the Na$^+$ mass fraction in reference seawater (Millero et al., 2008). And the other by converting the observed spectral particle number densities $N_{0.4-12}$ into total particle mass concentration assuming spherical




particles with the density of NaCl ($= 2160\,\mathrm{kg\,m^{-3}}$). Comparison shows that the median sea salt concentrations derived from $N_{0.4-12}$ during filter sampling intervals are on average 40% larger than filter-based values (data not shown). A low bias of the filter samples is expected because the smaller cut-off diameter ($<6\,\mu\mathrm{m}$) compared to the optical particle counter ($>11\,\mu\mathrm{m}$) limits capture of coarse sea salt aerosol, where much of the particle mass is located

(Fig. 16b).

During storms median atmospheric sea salt concentrations from both estimates showed a significant increase above background values (Fig. 17a). For example, median sea salt concentration based on $N_{0.4-12}$ increased by a factor 3, from 390 to $1215\,\mathrm{ng\,m^{-3}}$ when comparing calm ($3\pm1\,\mathrm{m\,s^{-1}}$) with windy ($10\pm1\,\mathrm{m\,s^{-1}}$) conditions (Fig. 17a).

A potential atmospheric sea salt concentration if surface snow released its sea salt content by blowing snow

sublimation is estimated as follows: taking a typical column total blowing snow sublimation rate of $0.1\,\mathrm{mm\,day^{-1}}$ observed at Halley during winter (King et al., 2001) and assuming a mean storm duration of 1 day on average the sea salt within the top 0.1 mm of snow can be released. Total sea salt mass observed in the top 0.1 mm of surface snow was converted into atmospheric concentration assuming a snow density $\rho_{snow}$ of $300\,\mathrm{kg\,m^{-3}}$ and complete mixing into a winter atmospheric boundary layer with an estimated mean depth of 100 m (Fig. 17). Further assuming that

the atmospheric concentration measured at 29 m is constant throughout the 100 m atmospheric column it is found that the potential snow reservoir is a factor 10 larger in comparison to the total atmospheric burden and therefore easily accounts for the sea salt aerosol increase observed during blowing snow (Fig. 17a).

Sulphate depletion in aerosol and snow provides a means to estimate source contributions from the open ocean and the blowing snow source above sea ice. During the 8 June and 12 August 2013 (ANT6) median $DF_{\mathrm{SO_4^{-2}}}$ of the

top 10 cm of snowpack was 0.27 and that of aerosol observed at 29 m was 0.29 (Table 5,4). Assuming linear mixing between two end members, i.e. the open ocean and the snow source, and assuming they are constant in space and time, one finds that on average 93% of the aerosol observed originates from snow on sea ice. Similarly for the the time period of the sea salt mass budget estimate the overlap in the ranges of $DF_{SO_4^{2-}}$ (with respect to $\mathrm{Na^+}$) observed in aerosol and snow show that most of the aerosol observed during storms had its origin in the regional snow on sea

ice (Fig. 17b).

4  Conclusions

Two consecutive sea ice cruises in the Weddell sea, Antarctica, during winter/spring 2013 provided the first direct observations of sea salt aerosol production from blowing snow above sea ice, thereby validating a model hypothesis to account for winter time sea salt maxima in polar regions not explained otherwise. Blowing or drifting snow always

30  lead to increases in aerosol during and after storms consisting mostly of sea salt inferred from the concurrent increase in atmospheric sodium concentrations. Observed aerosol gradients suggest that net production of SSA takes place near the top of the atmospheric layer containing suspended snow particles at $RH_{ice}<100\%$. Within the blowing





snow layer SSA net production is suppressed due to saturated conditions and scavenging by snow particles which is efficient at the warmer temperatures encountered during storms.

During storms average number and volume spectra of SSA over the sea ice increased significantly above background concentrations but remained below those over the open ocean. However, the observed relative increase of SSA
concentrations with wind speed suggests that on average the corresponding aerosol mass flux during storms was equal or larger above sea ice than above the open ocean, demonstrating the importance of the blowing snow source for atmospheric sea salt aerosol in winter and early spring. Lower SSA concentrations above sea ice relative to the open ocean may have been due to a combination of low storm frequency and low snow salinities in the area sampled. Upscaling of the SSA source flux of Weddell sea ice and atmospheric modelling similar to the study of Levine et al.
(2014) are needed to address this issue as well as implications for the sea ice proxy interpretation.

The main evidence for the sea ice origin of the sea salt aerosol observed comes from its chemical fractionation: for the first time we show that snow on sea ice is depleted in sulphate relative to sodium with respect to sea water. Similar depletion observed in the aerosol above the sea ice suggests that most sea salt originated from snow on sea ice with possibly minor contributions from frost flowers, and not the open ocean or leads, e.g. on average 93% during
the 8 June and 12 August 2013 period. A temporally very close association of snow particle and aerosol number density dynamics together with the far distance to the open ocean further support that the local blowing snow source of SSA dominates over advection. A mass budget calculation shows that even snow with low salinity ($<1$ psu) can account for observed increases of sea salt from blowing snow.

It is found that SSA originating from blowing snow is an important source of reactive bromine to the atmosphere,
which then can contribute to ozone depletion events. On average snow on sea ice and blowing snow showed no or small depletion of bromide relative to sodium with respect to sea water, whereas aerosol at 29 m was enriched suggesting that bromine loss takes place preferentially in the aerosol phase between 2 and 29 m above the sea ice surface.

Evaluation of the current model mechanism for sea salt aerosol production from blowing snow (Yang et al., 2008,
2018) showed that the model parameterisations used can generally be applied to snow on sea ice:

- estimates of drift threshold wind speed based on the empirical model of (Li and Pomeroy, 1997) that depends on temperature only agreed well with observations above the low-salinity snow encountered during this study. However, temperature alone may not be a good predictor if the surface snow is very saline, implying a relatively larger liquid fraction at a given temperature and therefore increased snow crystal cohesion and drift threshold wind speed. Further
measurements above a variety of sea ice surfaces will be needed to address this issue.

- modelled saltation layer snow mixing ratios exceed absolute values by a factor of 10, but the relative changes match the observations, and so the impact on calculating the SSA mass flux from blowing snow $Q_{SSA}$ is minor

- retrieval of mean particle diameter $\overline{d_p}$ and shape parameter $\alpha$ from fitting a 2-parameter gamma distribution to observed snow particle size spectra is robust, even when data exclude very small snow particles ($d_p < 46\,\mu$m); a
recommend range of $\overline{d_p}$ and $\alpha$ values is given for blowing snow episodes. Relative contributions from precipitating




snow to total suspended snow particles become more important a few tens of meters above the sea ice surface and need to be accounted for when interpreting snow particle size spectra. Future work needs to fill the observational gap ($d_p$ 12-46 µm)between very small snow particles and aerosol due to the importance for the formation of sub-micron aerosol and associated climate impacts.

- to a first order it is the distance to the sea ice surface, i.e. snowpack depth, that determines the salinity probability distribution of snow on sea ice. FYI can therefore be distinguished from MYI based on snow salinity, because snow on FYI is in general more shallow than on MYI. Secondary factors potentially increasing the difference in salinity between FYI and MYI are more frequent flooding of FYI with seawater due to negative freeboard and MYI desalination due to brine drainage. Snow depth retrieved from satellites may allow estimating snow salinity in
the absence of ground-based measurements.

    Snow salinity was shown to be a sensitive model parameter (Yang et al., 2018), which implies that SSA production above FYI should be larger than above MYI. Therefore, shifts in the ratio of FYI and MYI over time are expected to contribute to the variability of SSA in ice cores, which both represents an opportunity and a challenge for the quantitative interpretation of the sea salt sea ice proxy. It is expected that similar processes take place in the Arctic
regions.

Data availability. Data are available upon request from M.M. Frey (maey@bas.ac.uk)

Author contributions. MF, EW, AJ and PA designed the field experiments. MF, DJ and MNM carried out the field measurements. SN and IB contributed to CLASP, KN and PA to SPC instruments and data interpretation. MF prepared the manuscript and led the data interpretation with contributions from all co-authors.

Competing interests. The authors declare that they have no conflict of interest.

Acknowledgements. We are grateful to the Alfred Wegener Institute for Polar and Marine Research to enable our participation in expedition ANT-XXIX/6 and ANT-XXIX/7, cruise leader P. Lemke, captain and crew of RV Polarstern, and participating scientists for providing scientific and logistical support. We gratefully acknowledge financial support from the Natural Environment Research Council (UK) through the BLOWSEA project (NE/J023051/1 and NE/J020303/1). We thank
E. Ludlow and R. Tuckwell for support of the IC analysis at BAS, J. Klepacki, A. Webb (BAS) and S. Rodwell (SAMS) for invaluable engineering support, K.C. Leonard for helpful discussions and lending the rocket traps, T. Phillips for generating the Antarctic sea ice map, and LL & MM for continued inspiration.





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



Table 1. Overview of duration $\Delta t$, mean position, distance to the nearest open water $\Delta x$ and meteorology (ambient temperature $T_a$, wind speed $U_{39m}$ and wind direction at 39 m) for ice stations S1-S9 during ANT-XXIX/6 (8 June to 12 August 2013).

| Station ID[a] | begin – end (UTC) | $\Delta t$ (hr) | lat / lon (°) | $\Delta x$[b] (km) | $T_a$ (°C) | $U_{39m}$ (m s$^{-1}$) | wdir (°) |
|---|---|---|---|---|---|---|---|
| S1 (PS81/0493-2) | 21.06.2013 14:23 – 22.06.2013 00:29 | 10 | -66.445 / 0.116 | 607 | -20.6 | 8.2 | 207 |
| S2 (PS81/0496-1) | 24.06.2013 08:23 – 24.06.2013 18:18 | 10 | -67.470 / -0.033 | 721 | -12.6 | 12.5 | 51 |
| S3 (PS81/0497-1) | 26.06.2013 11:20 – 26.06.2013 14:20 | 3 | -68.049 / -0.340 | 786 | -3.4 | 13.2 | 19 |
| S4 (PS81/0500-5) | 03.07.2013 08:32 – 05.07.2013 20:52 | 60 | -67.909 / -6.762 | 993 | -23.2 | 5.2 | 200 |
| S5 (PS81/0503-2) | 08.07.2013 11:50 – 08.07.2013 23:39 | 12 | -67.192 / -13.225 | 913 | -14.0 | 3.8 | 251 |
| S6 (PS81/0506-1) | 11.07.2013 09:38 – 15.07.2013 12:41 | 99 | -67.203 / -23.165 | 803 | -16.8 | 6.3 | 167 |
| S7 (PS81/0515-1) | 26.07.2013 11:30 – 27.07.2013 00:00 | 13 | -63.415 / -51.246 | 381 | -19.6 | 13.9 | 246 |
| S8 (PS81/0517-2) | 29.07.2013 12:35 – 02.08.2013 20:24 | 104 | -63.400 / -51.170 | 379 | -25.2 | 6.8 | 208 |
| S9 (PS81/0518-3) | 04.08.2013 13:44 – 05.08.2013 20:18 | 31 | -62.918 / -53.265 | 214 | -7.8 | 9.1 | 273 |

[a]station ID used in this paper (Figs. 1,2); in brackets *RV Polarstern* nomenclature

[b] horizontal distance to the nearest open water on the ice station longitude based on sea ice extent on 15-June for S1-S3, on 15-July for S4-S7, and 15-Aug for S8 (Fig. 1).





Table 2. Overview of observed parameters, instruments used and reported data resolution $\Delta t$ during ANT-XXIX/6 (ANT6) from 8 June to 12 August 2013, and ANT-XXIX/7 (ANT7) from 14 August to 16 October 2013.

| Parameter | Instrument | $\Delta t$ | ANT6 | ANT7 |
|---|---|---|---|---|
| 1. Crow's nest (height 29 m) | continuous | | | |
| snow particle number density, $N_{46-478}$ ($d_p$ 46-478 µm) | Snow Particle Counter[a] | 1 min | ✓ | ✓ |
| aerosol number density, $N_{0.4-12}$ ($d_p$ 0.36-11.62 µm) | Compact Light-Weight Aerosol Spectrometer[b] | 1 min | ✓ | |
| bulk aerosol chemistry ($Na^+, Cl^-, SO_4^{2-}, Br^-$) | Low-Volume Filter Samples | 3–48 h | ✓ | ✓ |
| meteorology ($T, p, RH, U, wdir$) | *RV Polarstern* Observatory[c] | 1 min | ✓ | ✓ |
| 2. Sea ice (height 0.07-2.0 m) | during ice stations | | | |
| snow particle number density, $N_{46-478}$ ($d_p$ 46-478 µm) | Snow Particle Counter[a] | 1 min | ✓ | |
| aerosol number density, $N_{0.4-12}$ ($d_p$ 0.36-11.62 µm) | Compact Light-Weight Aerosol Spectrometer[b] | 1 min | ✓ | |
| bulk aerosol chemistry ($Na^+, Cl^-, SO_4^{2-}, Br^-$) | Low-Volume Filter Samples | 3–48 h | ✓ | |
| snow chemistry ($Na^+, Cl^-, SO_4^{2-}, Br^-, S_p$) | Ion Chromatography, Salinometer[d] | hr–days | ✓ | |
| 3-D wind field ($u, v, w$) | Sonic Anemometer[e] | 1 min | ✓ | |
| $T, RH$ | Temperature/ Humidity Probe[f] | 1 min | ✓ | |

[a]SPC-95, Niigata Electric Co., Ltd. (Nishimura et al., 2014) [b]CLASP (Hill et al., 2008; Norris et al., 2008) [c]details at https://spaces.awi.de/confluence/display/PSdevices/Bordwetterwarte [d]SensIon 5, Hach [e]METEK USA-1 [f]VAISALA HMP45





Table 3. Overview of mean aerosol filter procedure blanks and resulting limit of detection (LOD): 1. ultra-high purity (UHP) water, 2. filter extraction, 3. cleaned and unused filter and 4. field blank (see text). Field blank values are used to correct the raw filter ion concentration.

| Parameter | $Na^+$ | $Cl^-$ | $SO_4^{2-}$ | $Br^-$ | $N^{a}$ |
|---|---|---|---|---|---|
| A) Aerosol filter procedure blank | | | | | |
| 1. UHP (electric resistivity $18.2\,M\Omega cm$) ($ng\,g^{-1}$) | 0.4 | 4.0 | 4.5 | 0.0 | 85 |
| 2. extraction ($ng\,g^{-1}$) | 3.6 | 4.8 | 8.6 | 3.4 | 14 |
| 3. filter ($ng\,g^{-1}$) | 2.3 | 9.5 | 8.5 | 0.0 | 8 |
| 4. field ($ng\,g^{-1}$) | 6.0 | 16.8 | 9.1 | 0.5 | 10 |
| ratio field blank : sample mean (%) | 3.1 | 4.9 | 20.3 | 34.3 | 183 |
| B) Limit of detection (LOD) | | | | | |
| LOD ($ng\,g^{-1}$)[b] | 13.4 | 27.9 | 17.1 | 1.0 | 10 |
| LOD ($ng\,m^{-3}$)[c] | 22.6 | 47.2 | 29.0 | 1.7 | 141 |
| LOD ($ng\,m^{-3}$)[d] | 35.4 | 73.9 | 45.4 | 2.6 | 43 |

[a]sample size [b]defined as $2\times1$-$\sigma$ of the field blank [c]based on crow's nest mean air sample STP-volume ($6.4\,m^3$) and interval ($12\,hr$) [d]based on sea ice mean air sample STP-volume ($3.3\,m^3$) and interval ($5.6\,hr$)





Table 4. Descriptive statistics of the aerosol chemistry during ANT-XXIX/6 (ANT6) and ANT-XXIX/7 (ANT7) with mean and median values weighted by the filter sampling interval. Ion and sea salt concentrations are in units of $\mathrm{ng\,m^{-3}}$. See section 2.4 for definition of depletion factors $DF$.

| Parameter | ANT6 at 2 m | | | at 29 m | | | ANT7 at 29 m | | |
|---|---|---|---|---|---|---|---|---|---|
| | mean | median | $N^{\mathrm{a}}$ | mean | median | $N^{\mathrm{a}}$ | mean | median | $N^{\mathrm{a}}$ |
| sea-salt[b] | 707 | 336 | 43 | 1253 | 639 | 106 | 559 | 425 | 28 |
| $\mathrm{Na^+}$ | 217 | 103 | 43 | 384 | 196 | 106 | 171 | 130 | 28 |
| $\mathrm{Cl^-}$ | 379 | 179 | 43 | 656 | 302 | 106 | 311 | 232 | 27 |
| $\mathrm{SO_4^{2-}}$ | 28 | 19 | 38 | 75 | 45 | 84 | 33 | 23 | 28 |
| $\mathrm{Br^-}$ | 2.0 | 1.9 | 42 | 1.5 | 0.7 | 98 | 0.5 | 0.5 | 23 |
| $DF_{\mathrm{SO_4^{2-}}}$ | 0.29 | 0.48 | 38 | 0.07 | 0.29 | 74 | 0.12 | 0.21 | 27 |
| $DF_{\mathrm{Na^+}}$ | -0.08 | -0.03 | 43 | -0.09 | -0.03 | 99 | -0.02 | -0.01 | 27 |
| $DF_{\mathrm{Br^-}}$ | -0.50 | -0.41 | 25 | 0.08 | 0.37 | 88 | 0.27 | 0.51 | 22 |

[a]sample size [b]sea salt concentration is derived by multiplying the $\mathrm{Na^+}$ concentration by 3.262 based on the $\mathrm{Na^+}$ mass fraction in reference seawater after Millero et al. (2008)



Table 5. Descriptive statistics of the volume-integrated snow chemistry during ANT-XXIX/6 on first-year sea ice (FYI) at ice stations S1-S6, on multi-year sea ice (MYI) at ice stations S7-9, and for snow layers within 10 cm of the snow surface (TOP10). Ion and sea salt concentrations are in units of $\mu g\,g^{-1}$. See section 2.4 for definition of depletion factors $DF$.

| Parameter | FYI | | | MYI | | | TOP10 | | |
|---|---|---|---|---|---|---|---|---|---|
| | mean | median | $N^{a}$ | mean | median | $N^{a}$ | mean | median | $N^{a}$ |
| snow depth (cm) | 20.9 | 19.0 | 17 | 50.0 | 33.0 | 7 | - | - | - |
| $S_p$ (psu) | 1.40 | 0.11 | 110 | 0.82 | 0.02 | 104 | 0.31 | 0.06 | 96 |
| sea salt[b] | 1163 | 82 | 87 | 584 | 21 | 96 | 246 | 57 | 81 |
| $Na^+$ | 357 | 25 | 87 | 179 | 6 | 96 | 76 | 17 | 81 |
| $Cl^-$ | 676 | 48 | 88 | 302 | 13 | 99 | 143 | 35 | 82 |
| $SO_4^{2-}$ | 61 | 5.89 | 88 | 30 | 0.87 | 99 | 17 | 2.68 | 82 |
| $Br^-$ | 4.25 | 0.18 | 86 | 1.74 | 0.07 | 91 | 1.01 | 0.12 | 79 |
| $DF_{SO_4^{2-}}$ | 0.19 | 0.24 | 86 | 0.33 | 0.35 | 94 | 0.27 | 0.27 | 80 |
| $DF_{Na^+}$ | 0.02 | 0.06 | 87 | 0.04 | 0.08 | 91 | 0.00 | 0.06 | 80 |
| $DF_{Br^-}$ | -0.12 | 0.05 | 81 | 0.01 | 0.00 | 80 | -0.01 | 0.05 | 72 |

[a]sample size [b]sea salt concentration is derived by multiplying the $Na^+$ concentration by 3.262 based on the $Na^+$ mass fraction in reference seawater after Millero et al. (2008)





Table 6. Summary of snow particle size distribution properties during blowing snow, i.e. when $U_{10m} > U_t$ (=7.1 m s$^{-1}$ as observed during this study): median, lower ($Q_{0.25}$) and upper quartile ($Q_{0.75}$) of particle mean diameter $\overline{d_p}$ (=$\alpha\beta$) in µm and shape parameter $\alpha$ derived from fitting a 2-parameter gamma probability density function (Eq. 5) to observations; $\Delta t$ is total sampling time .

| parameter | median | $Q_{0.25}$ | $Q_{0.75}$ | $\Delta t$ (hr) |
|---|---|---|---|---|
| $\overline{d_p}$ at <0.2 m | 129 | 117 | 152 | 47 |
| $\overline{d_p}$ at 29 m[a] | 115 | 95 | 139 | 47 |
| $\overline{d_p}$ at 29 m | 115 | 93 | 147 | 867 |
| $\alpha$ at <0.2 m | 4.8 | 4.0 | 11.2 | 47 |
| $\alpha$ at 29 m[a] | 8.0 | 4.1 | 11.8 | 47 |
| $\alpha$ at 29 m | 6.4 | 4.2 | 11.0 | 867 |

[a]including only data when also measurements at <0.2 m are available during blowing snow ($U_{10m} > U_t$).




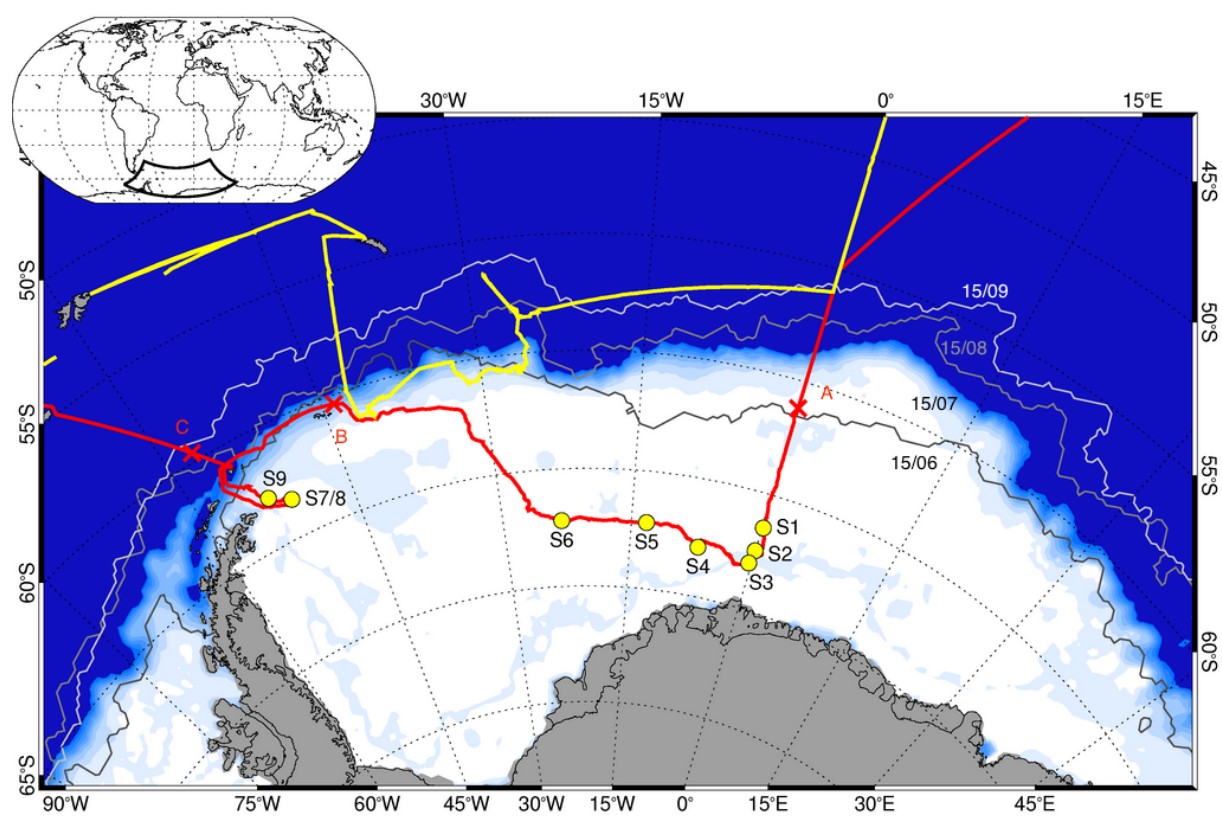

Figure 1. Cruise tracks of *RV Polarstern* in the Weddell Sea for the winter expedition ANT-XXIX/6 from 8 June to 12 August 2013 (red line) and the spring expedition ANT-XXIX/7 from 14 August to 16 October 2013 (yellow line). Symbols indicate the location of ice stations S1–9 (Table 1). Crosses show ship positions when entering the sea ice on 17 June (A), reaching the marginal sea ice zone MIZ on 22 July (B) and returning to the open ocean on 9 August (C). Sea ice concentrations on 15 July 2013 are shown as shaded area, and sea ice extent on 15 June, 15 August and 15 September 2013, respectively, as grey solid lines, all based on Nimbus-7 satellite microwave radiometer measurements (Comiso, 2018).



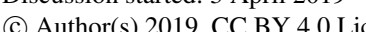


Figure 2. Overview of atmospheric observations in the Weddell Sea from 8 June to 12 August 2013 (ANT-XXIX/6): (a) horizontal wind speed $U$ at 39 m; time periods at ice stations S1-9 (Table 1) are highlighted in red. (b) ambient temperature $T_a$ and relative humidity with respect to ice $RH_{ice}$ at 29 m; red symbols refer to $RV\,Polarstern$ positions shown in Fig. 1, indicating when the ship entered the sea ice on 17 June (A), reached the marginal sea ice zone (MIZ) on 22 July (B) and returned to the open ocean on 9 August (C). (c) total number densities $N_{46-478}$ of airborne snow particles at 29 m with grey shaded areas indicating periods with observed drifting or blowing snow. (d) total number densities $N_{0.4-12}$ of aerosol at 29 m, (e) aerosol Na$^+$ concentrations and (f) sulphate depletion factor $DF_{SO_4^{2-}}$, both at 29 m and 2 m, respectively.



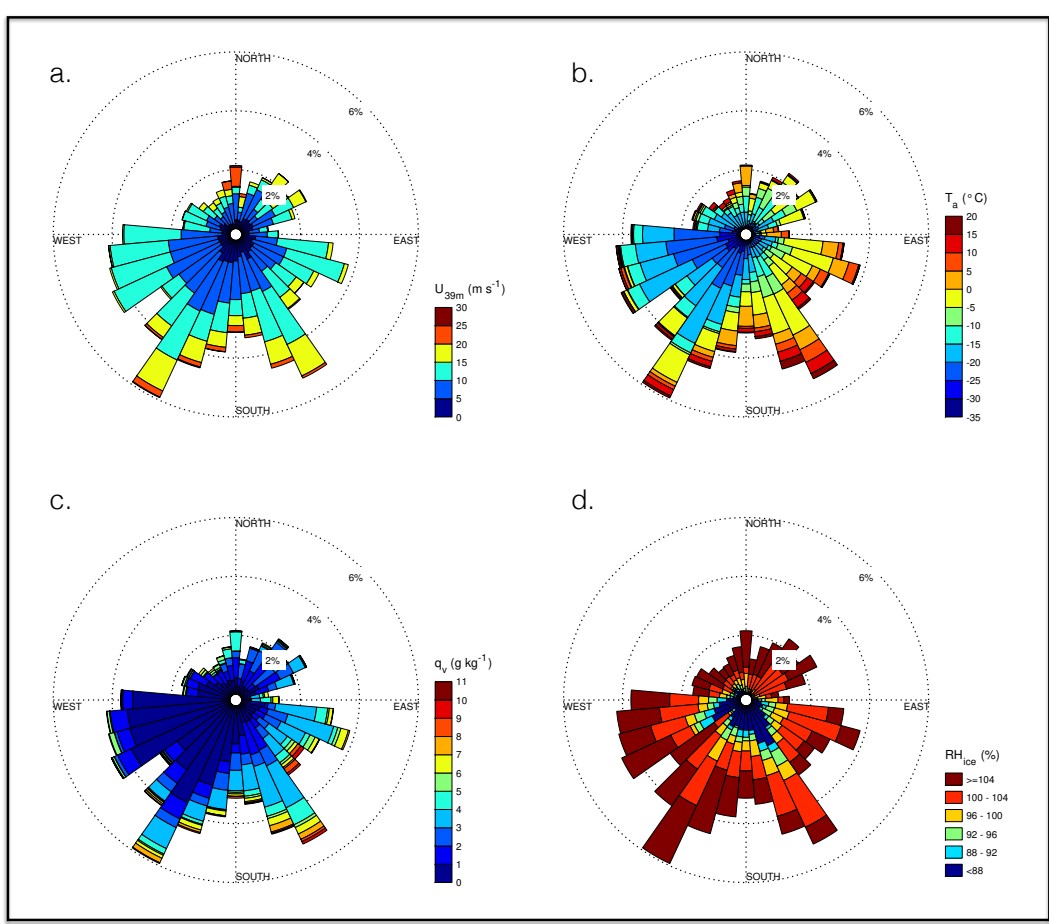

Figure 3. Wind rose plots for the position of *RV Polarstern* during ANT-XXIX/6 (8 June–12 August 2013): (a) horizontal wind speed $U$ at 39 m, (b) ambient temperature $T_a$, (c) specific humidity of air $q_v$, and (d) relative humidity with respect to ice $RH_{ice}$ (b–d at 29 m).




Figure 4. Blowing snow events during the 23 June – 2 July 2013 period: (a) wind speed $U$ at 39 m and and $RH_{ice}$ at 29 m, (b) snow particle size distribution $dN/\log d_p$ (46 μm< $d_p$ <478 μm) at 29 m and (c) at 0.1 m; (d) aerosol size distribution $dN/\log d_p$ (0.36 μm< $d_p$ <11.62 μm) at 29 m and (e) at 2 m; dark blue shading indicates zero particle counts and white background that no data are available. (f) aerosol $Na^+$ concentrations (vertical bars) and sulphate depletion factor $DF_{SO_4^{2-}}$ (open symbols) at 29 m and 2 m, respectively.



Figure 5. Blowing snow events during the 10–16 July 2013 period: (a) wind speed $U$ at 39 m and $RH_{ice}$ at 1.2 and 2.0 m (blue and cyan solid lines, repectively); vertical red lines delimit the period shown in more detail in Fig. 6. (b) snow particle size distribution $dN/\log d_p$ (46 µm$< d_p <$478 µm) at 29 m and (c) at 0.1 m; (d) aerosol size distribution $dN/\log d_p$ (0.36 µm$< d_p <$11.62 µm) at 29 m and (e) at 2 m; dark blue shading indicates zero particle counts and white background that no data are available; (f) aerosol $Na^+$ concentrations (vertical bars) and sulphate depletion factor $DF_{SO_4^{2-}}$ (open symbols) at 29 m and 2 m, respectively.




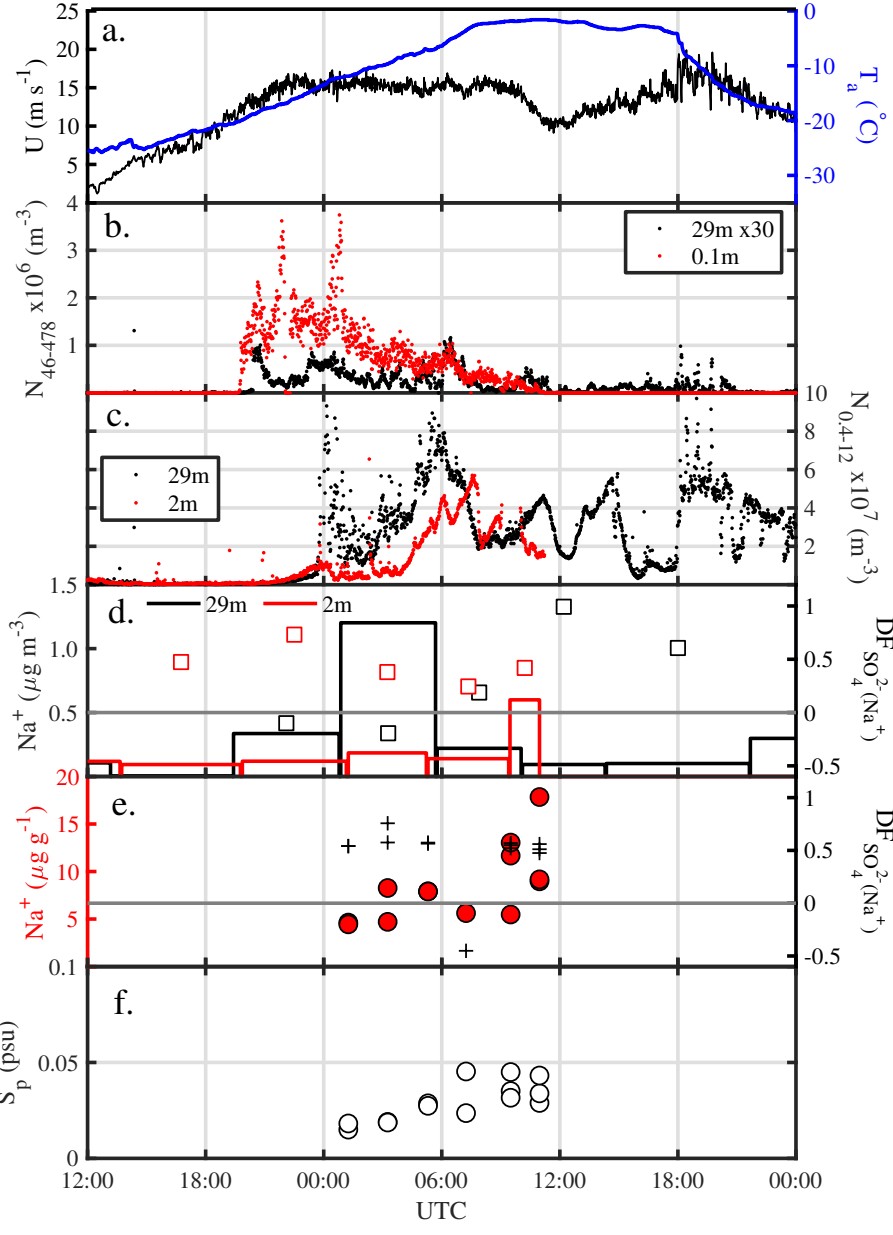

Figure 6. Details of the blowing snow event from 14 to 15 July 2013 (period marked by vertical red lines in Fig. 5a): (a) ambient temperature $T_a$ at 29 m and wind speed $U$ at 39 m, (b) total number densities $N_{46-478}$ of airborne snow particles at 29 and 0.1 m, (c) total number densities $N_{0.4-12}$ of aerosol at 29 and 2 m, (d) aerosol $Na^+$ concentrations (vertical bars) and $DF_{SO_4^{2-}}$ (open symbols) at 29 m and 2 m, respectively; (e) $Na^+$ concentrations, sulphate depletion factor $DF_{SO_4^{2-}}$ and (f) salinity $S_p$ in blowing snow collected 1-17 cm above the surface.





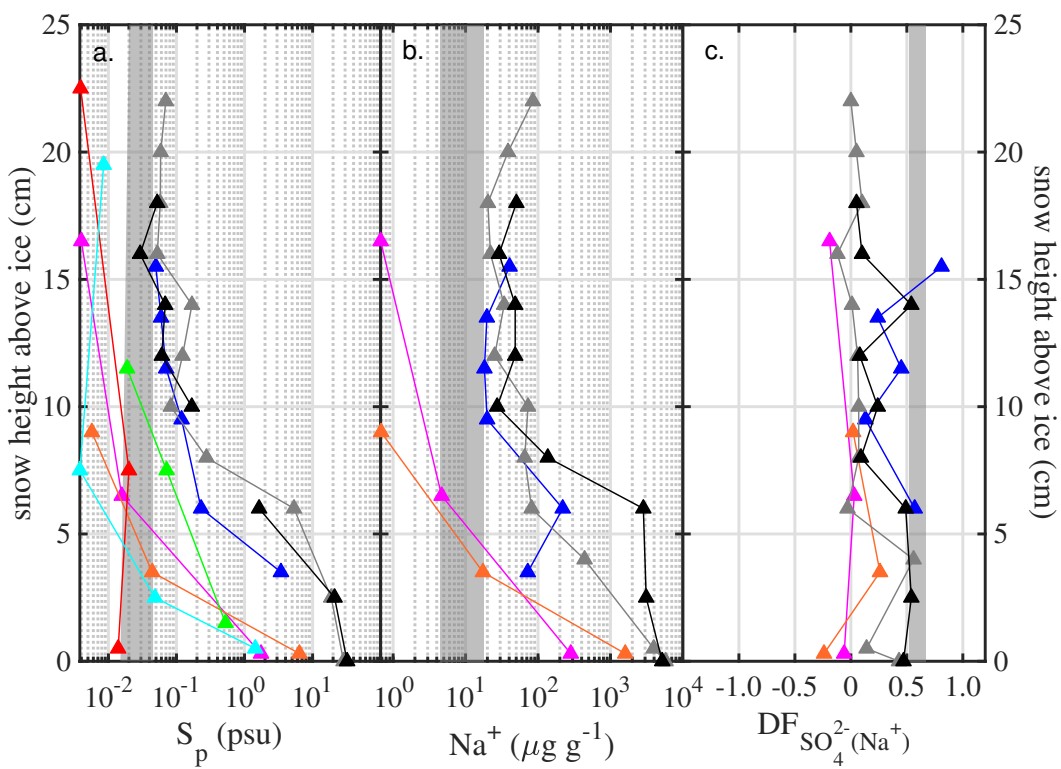

Figure 7. Vertical snowpack profiles sampled at various locations on the ice floe of ice station S6 during the 11–14 July 2013 period: (a) salinity $S_p$, (b) $Na^+$ concentrations and (c) sulphate depletion factor $DF_{SO_4^{2-}}$ with respect to $Na^+$. Shaded areas illustrate the range of the respective parameter measured in blowing snow on 15 July 2013 (Fig. 6e-f).





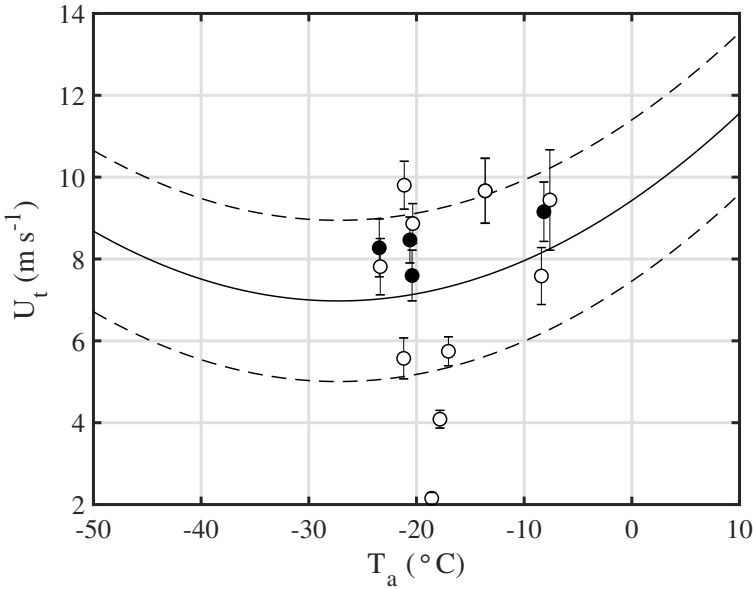

Figure 8. The threshold wind speed of blowing snow $U_t$ at $10\,\mathrm{m}$ as a function of ambient temperature $T_a$ at $2\,\mathrm{m}$, above the sea ice in the Weddell Sea. Plotted are 10-minute means of observations centred on the time, when the snow drift density $\mu$ right above the snow surface exceeds a threshold of $0.005\ \mathrm{kg\,m^{-3}}$ (closed symbols) and $0.001\ \mathrm{kg\,m^{-3}}$ (open symbols), respectively; errobars indicate $\pm 1\sigma$. Shown for comparison are predictions by the parameterisation of Li and Pomeroy (1997) (Eq. 4) and their stated uncertainties as solid and dashed lines, respectively.





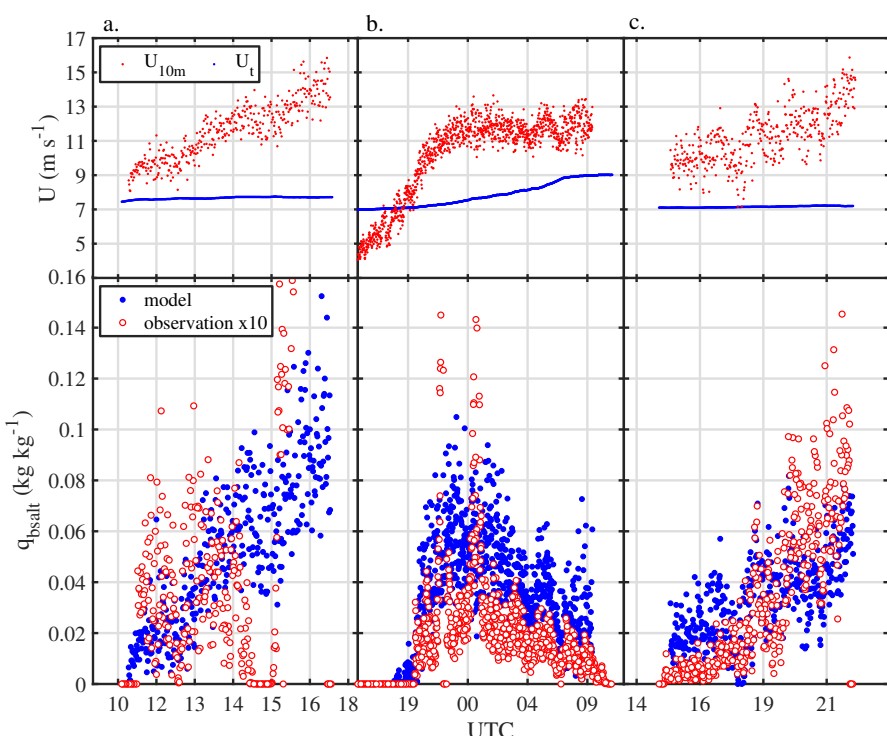

Figure 9. Snow mixing ratios in the saltation layer, $q_{bsalt}$, during (a) 24 June (ice station S2), (b) 14-15 July (ice station S6) and (c) 26 July (ice station S7), where the snow particle counter was mounted below the reported value of the saltation layer depth ($=0.1$ m). Observations multiplied by ten are compared to predictions by the parameterisation of Déry and Yau (1999) (Eq. 3). The top panels show observed wind speed $U_{10m}$ and threshold wind speed $U_t$ computed with the empirical model of Li and Pomeroy (1997) (Eq. 4), both at 10 m.




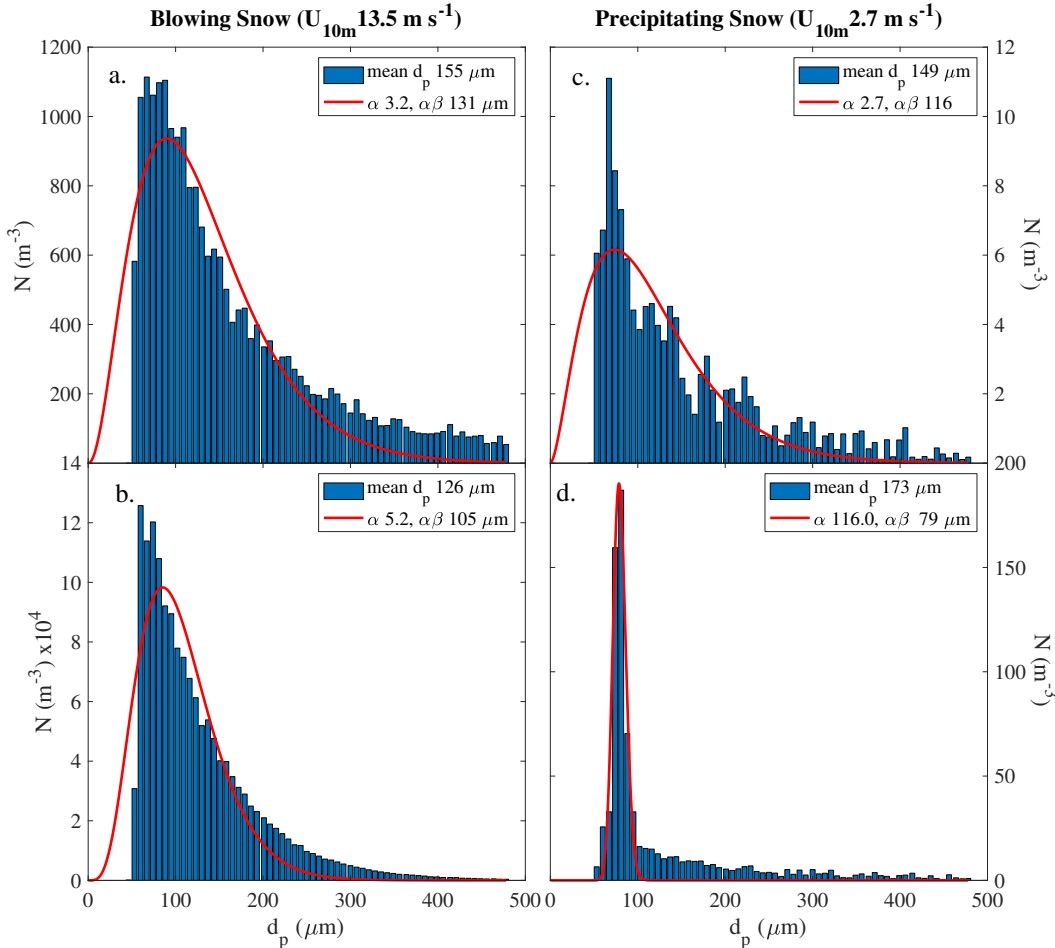

Figure 10. Average snow particle size spectra $f(d_p)$ measured at 29 m and <0.2 m during blowing snow on 14 July 23:15-23:25 (a,b) and during light snowfall coinciding with very low winds on 3 July 14:30 - 4 July 7:00 (c,d). Solid lines indicate the fit obtained with a 2-parameter gamma probability density function (Eq. 5). Note that mean particle diameters $d_p$ computed from observations show a positive bias compared to $d_p$ ($=\alpha\beta$) retrieved from the fitting procedure, since the SPC instrument does not detect small particles ($d_p$<46 μm).




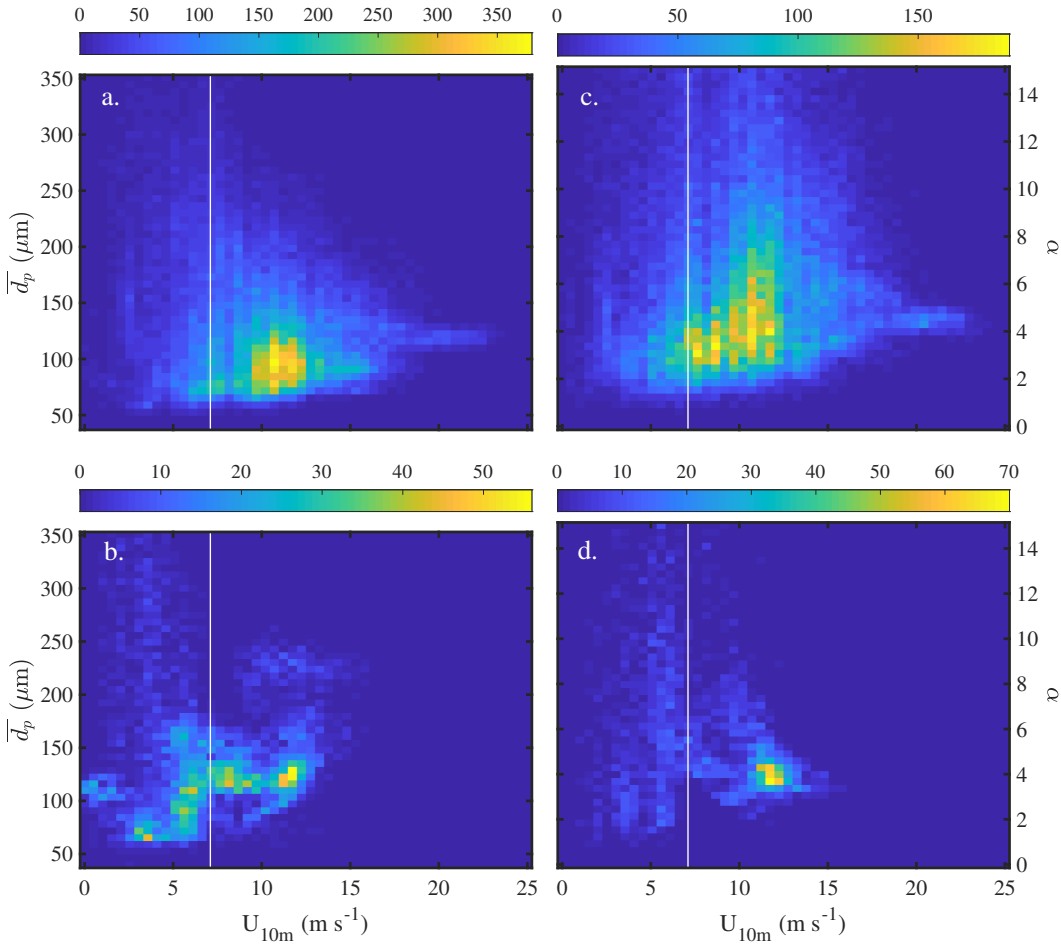

Figure 11. Properties of blowing snow particle size distributions above sea ice as a function of the 10-m wind speed $U_{10m}$ retrieved from fitting a 2-parameter gamma probability density function (Eq. 5) to observations. Mean snow particle diameters $\overline{d_p}$ ($=\alpha\beta$) (a) at 29 m during ANT-XXIX/6 and 7, and (b) at <0.2 m during ANT-XXIX/6. The shape parameter $\alpha$ (c) at 29 m and (d) at <0.2 m. The vertical line represents the observed mean snowdrift threshold wind speed $U_t$ of $7.1\,\mathrm{m\,s}^{-1}$ corresponding to $u_{*t}$ of $0.37\,\mathrm{m\,s}^{-1}$. Data were binned into 50 bins in x and y direction, with the colour indicating the sample density in each bin.




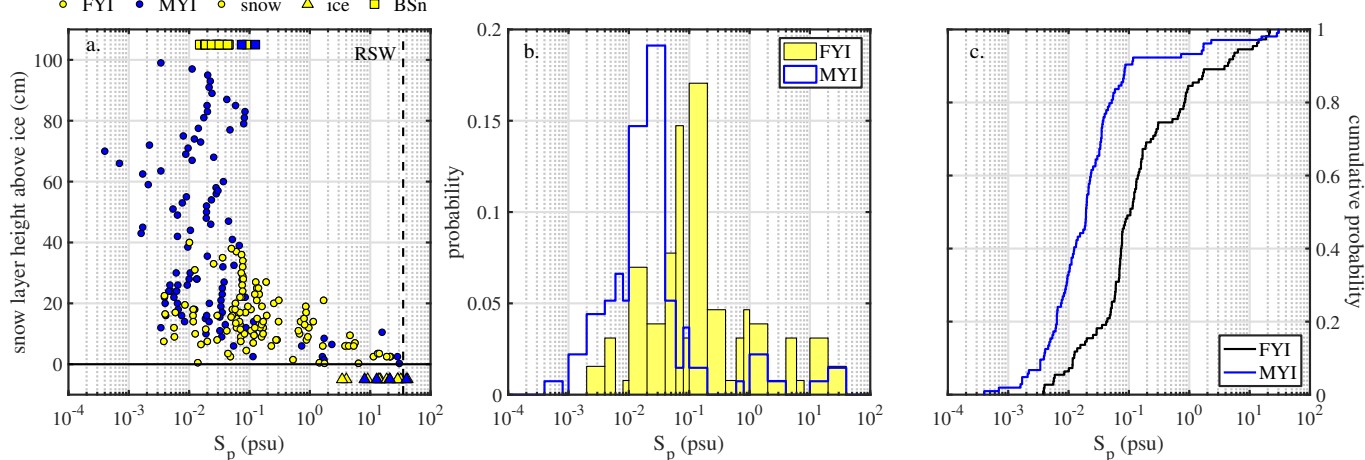

Figure 12. Panel (a) shows salinity $S_p$ of snow on first-year sea ice (FYI, yellow symbols) at ice stations S1-6 and multi-year sea ice (MYI, blue symbols) at ice stations S7-9 in the Weddell Sea during austral winter 2013 as a function of snow layer height above the sea ice surface. For comparison $S_p$ of the sea ice surface (triangles) and blowing snow at 1-17 cm above the snowpack (squares) are shown as well. The vertical dashed line indicates $S_p$ (=35.165 psu) of reference sea water (RSW). Panel (b) shows salinity $S_p$ probability distributions for shallow snowpacks (mean depth 21 cm) above first-year sea ice (FYI) at ice stations S1-6 and for deep snowpacks (mean depth 50 cm) above multi-year sea ice (MYI) at ice stations S7-S9 in the Weddell Sea during austral winter 2013. Panel (c) shows respective cumulative probabilities of $S_p$. The distribution statistics are summarised in Table 5.





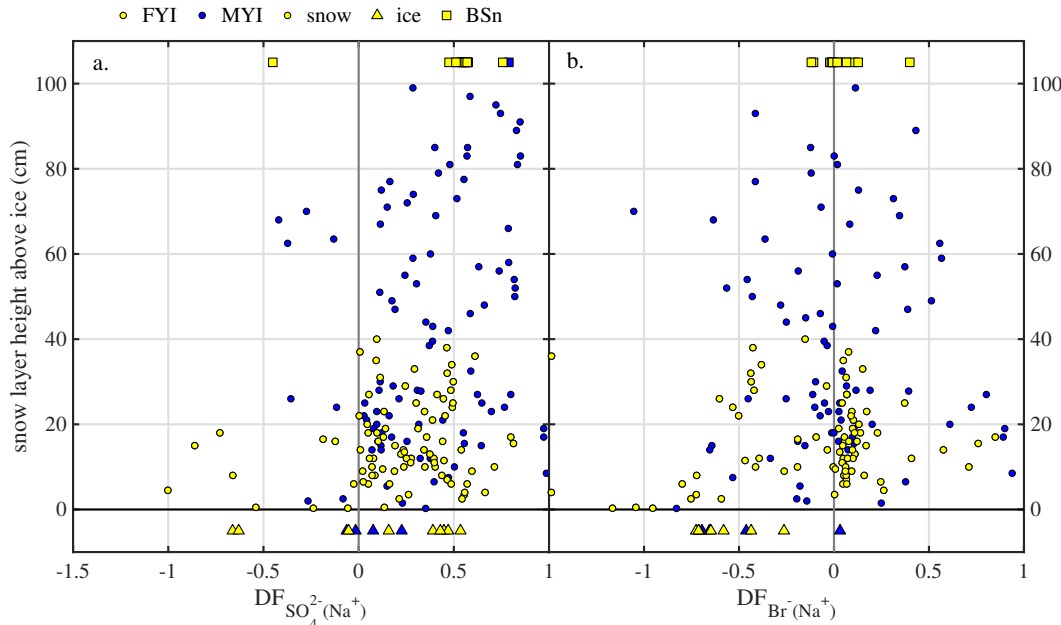

Figure 13. Chemical fractionation of snow on sea ice in the Weddell Sea during austral winter 2013. Panel (a) shows sulphate depletion factors $DF_{SO_4^{2-}}$ (with respect to $Na^+$) of snow on first-year sea ice (FYI, yellow symbols) at ice stations S1-6 and multi-year sea ice (MYI, blue symbols) at ice stations S7-9 in the Weddell Sea as a function of snow layer height above the sea ice surface. For comparison $DF_{SO_4^{2-}}$ of the sea ice surface (triangles) and blowing snow at 1-17 cm above the snowpack (squares) are shown as well. Panel (b) shows the same but for bromide depletion factors $DF_{Br^-}$ (with respect to $Na^+$). The sample statistics are summarised in Table 5.




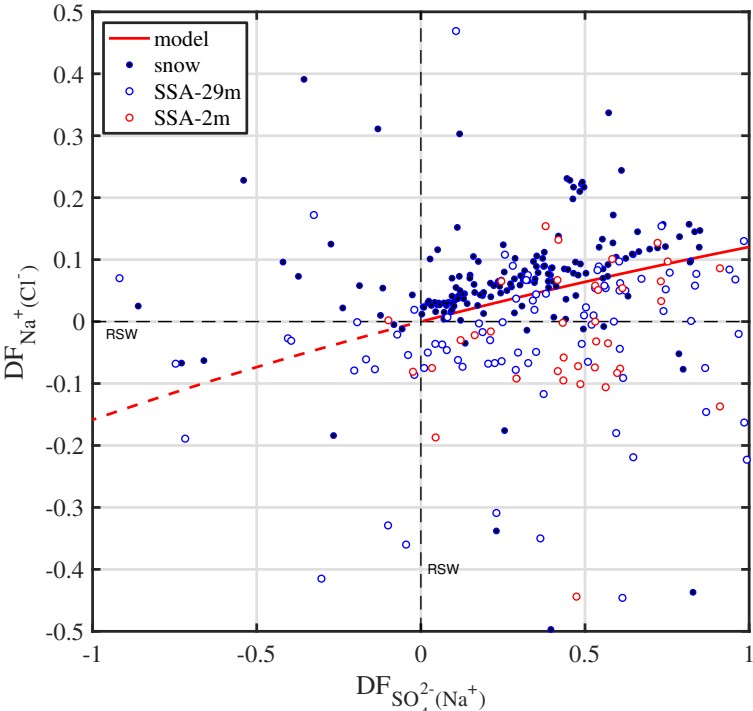

Figure 14. Theoretical relationship between depletion factors of sodium $DF_{Na^+}$ (with respect to $Cl^-$) and of sulphate $DF_{SO_4^{2-}}$ (with respect to $Na^+$) in freezing seawater if mirabilite ($Na_2SO_4 \cdot 10 H_2O$) is progressively precipitated and instantaneously removed (red line). When all sulphate is removed ($DF_{SO_4^{2-}} = 1$) sodium depletion reaches its theoretical maximum ($DF_{Na^+} = 1.1204$). See section 2.4 for definition of depletion factors $DF$, which are zero in reference seawater (RSW) (black dashed lines). Data refer to all observations from this study in snow (solid symbols), and aerosol at 2 m and 29 m above the sea ice surface (open symbols).





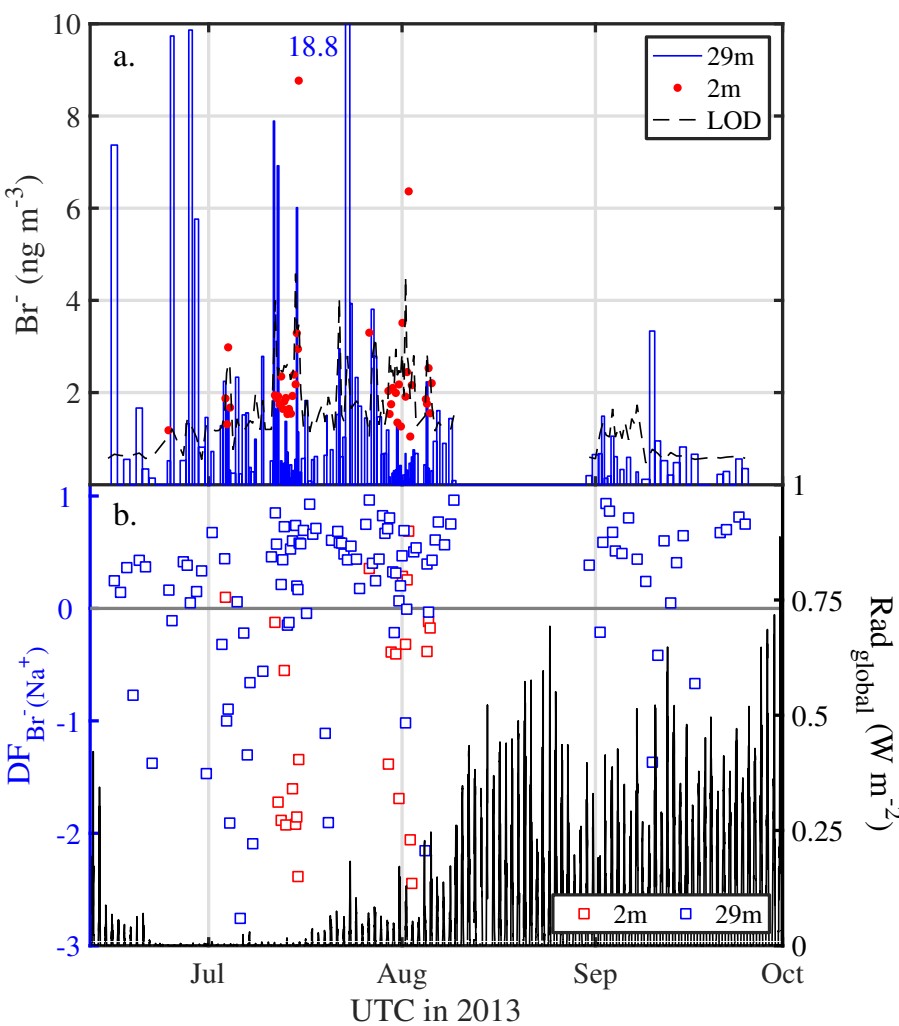

Figure 15. Aerosol bromine chemistry above sea ice observed in the Weddell Sea during austral winter/spring 2013. (a) aerosol Br$^-$ concentrations at 29 and 2 m with the dashed line indicating the theoretical LOD (section 2.3); (b) depletion factors of bromide $DF_{Br^-}$ (with respect to Na$^+$) in aerosol at 29 and 2 m and global radiation to indicate light conditions.





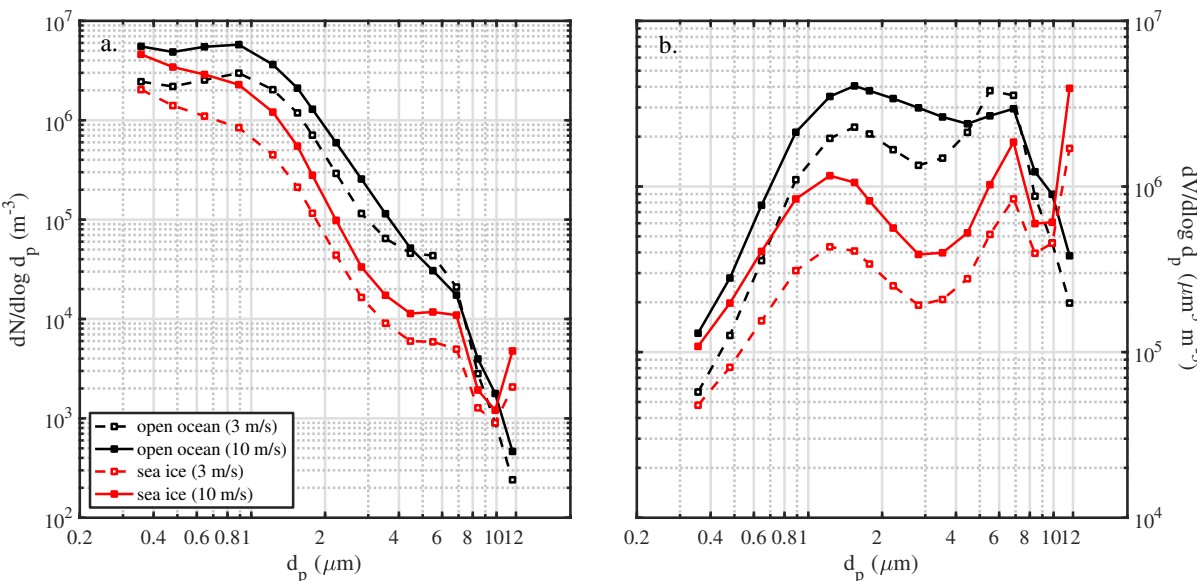

Figure 16. Average number distributions (panel a) and volume distributions (panel b) of aerosol above the open ocean (13 to 16 June 2013) and sea ice in the Weddell Sea (18 June to 21 July 2013) during calm and windy conditions. Data used are observations from $29\,\mathrm{m}$ above the sea surface at ambient $RH$; numbers in brackets indicate the wind speed $U_{10m}$ $\pm 1\,\mathrm{m\,s^{-1}}$. Note that standard deviations of the mean values are always smaller than half the symbol size.



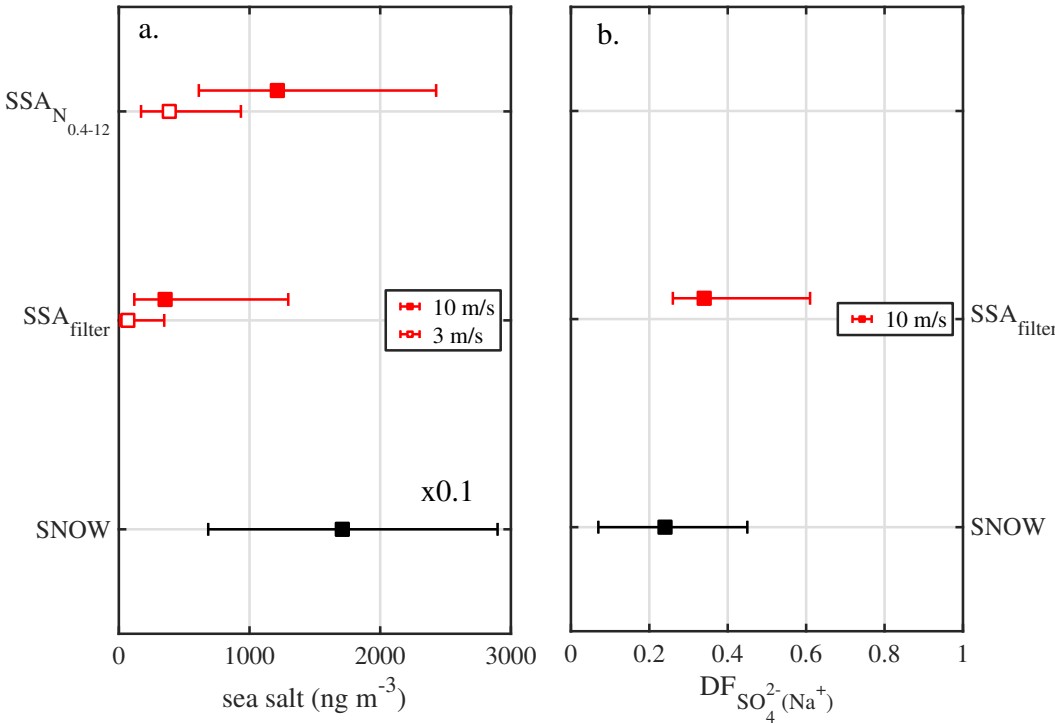

Figure 17. The sea salt mass budget and sulphate depletion in snow and atmosphere above first year sea ice (18 June to 21 July 2013): (a) median atmospheric sea salt concentrations derived from particle number densities $N_{0.4-12}$ and filter $Na^+$ concentrations at 29 m during calm ($3\pm1\,\mathrm{m\,s^{-1}}$) and windy ($10\pm1\,\mathrm{m\,s^{-1}}$) conditions. For comparison, total sea salt mass observed in the top 0.1 mm of surface snow was converted into atmospheric concentration assuming a snow density $\rho_{snow}$ of $300\,\mathrm{kg\,m^{-3}}$ and mixing into a 100 m deep atmospheric boundary layer (for comparison with atmospheric observations multiplied by 0.1). Panel (b) shows for the same time period median sulphate depletion factor $DF_{SO_4^{2-}}$ (with respect to $Na^+$) in aerosol and surface snow. Symbols and errorbars represent median and lower and upper quartiles, respectively.