# Peer review of "First direct observation of sea salt aerosol production from blowing snow above sea ice"

_Atmospheric Chemistry and Physics, 2019_

## Short Comment (SC1) · 3 Jun 2019

This manuscript describes an interesting set of measurements and detailed analysis confirming the blowing snow as a significant source for sea salt aerosol in the vicinity of sea ice in coastal Antarctica. We agree that this is an important result with significant implications for polar tropospheric aerosol loadings and heterogeneous halogen chemistry. However, it would be helpful to both the authors and readers of this article to refer to prior work also published in ACP showing similar results from measurements taken on sea ice in the Ross Sea. Giordano et al., 2018 also clearly identifies blowing snow on sea ice as a significant source of chlorine rich sea salt aerosol from online Aerosol Mass Spectrometer measurements of aerosol composition, optical measurements of blowing snow and interstitial aerosol concentrations and offline measurements of sur-

face and blowing snow composition. The consistency between the results from observations using different techniques and on opposite sides of the Antarctic continent further indicates the importance of this mechanism to the overall Antarctic aerosol budget.

Lars Kalnajs and Peter DeCarlo

Reference:

Giordano, M. R., Kalnajs, L. E., Goetz, J. D., Avery, A. M., Katz, E., May, N. W., Leemon, A., Mattson, C., Pratt, K. A., and DeCarlo, P. F.: The importance of blowing snow to halogen-containing aerosol in coastal Antarctica: influence of source region versus wind speed, Atmos. Chem. Phys., 18, 16689-16711, https://doi.org/10.5194/acp-18-16689-2018, 201

---

## Referee Comment (RC1) · Anonymous Referee #1 · 12 Jun 2019

Source assignment of proxies is a basic prerequisite for interpreting climate archives in terms of past climate as well as climate change. Concerning polar ice cores, ionic impurities originate primarily from aerosol deposition. Amongst them, interpretation of sea salt aerosol deposition archived in ice cores is especially challenging and controversial because the contribution of two different and competing sources - viz. open water versus sea ice - is up for debate. In addition it became apparent that sea salt aerosol production over ice-covered oceans may contribute significantly to the global sea salt aerosol budget. The manuscript at hand addresses this pivotal subject and provides thorough and direct observational evidence of sea salt aerosol production from blowing snow above sea ice. The important conclusions drawn are based on comprehensive state of the art ship-borne aerosol and snow measurements during

winter / early springtime in the Weddell Sea region. Although the main conclusions are primarily restricted to the chosen site, there are certainly strong implications for climate research in the Southern Ocean realm and climate related interpretation of sea salt profiles from ice cores in general.

The authors have accomplished a clear, well-organized and concise paper. The methodology is sound and assumptions are identified clearly and conscientiously. From my point of view, all parts, including figures, are essential. The manuscript certainly addresses the scientific scope of ACP and I recommend a final publication after some minor revisions I specified below.

1. Abstract, page 1, line 13 and Conclusions, page 24, line19: The authors state that bromine enrichment was typical at 29 m height, but from Chapter 3.4.2 and Fig. 15, bromine depletion is evident. Please clarify.

2. Chapter 2.3, Aerosol chemical composition: Could you assess the impact of pollution on chemical aerosol composition? Was the bulk aerosol sampling contamination controlled?

3. Chapter 3.2: Impact of snow precipitation on blowing/drifting snow: Did you access the regular weather reports from the ships meteorological office in this case?

4. Pages 12/13 and Fig.7: Regarding the salinity (Sp) of blowing snow, corresponding Sp-values of the uppermost surface snow layer are decisive. Did you take samples from surface snow; say < 1 cm deepness below surface? Figure 7: The reader cannot get an idea about the salinity of the surface snow layer from this graph. It would be informative as well to specify the total depth of the snow layer shown here, not just the snow height above sea ice.

5. Chapter 3.3.2, Snow particle size distribution: Is it possible to rate the impact of the ships profile on the local wind field and eventually on the measured snow particle size distribution? 6. Chapter 3.4.1, page 19, lines 28-30: As for Antarctic winter, acid induced Cl- loss is rather extraordinary because production of acidic sulphur compounds usually cease at the end of summer / fall. Are there any indications for alternative HNO3 induced Cl- loss in your data?

7. Chapter 3.4.2, Chemical fractionation of Br-, lines 28-33 and Fig. 15: There is strong bromine depletion during polar night in July when global radiation was about zero (Fig. 15b). This peculiarity deserves some discussion.

8. Figure 5 and page 12, lines 8-9: By the way: During late afternoon of the 11 July, there is an outstanding Na+ peak associated with corresponding sulphate depletion, while the wind speed seemed just close to the threshold value (well below 10 m/s throughout the whole day). Any ideas?

---

## Referee Comment (RC2) · Anonymous Referee #2 · 13 Jun 2019

Frey et al present an observational study of sea salt aerosol (SSA) production from blowing snow above sea ice, through measurements during winter 2013 in the Weddell Sea, Antarctica. Since the modeling hypothesis presented by Yang et al (2008, GRL), the mechanism of SSA production from blowing snow has been implemented in numerous modeling studies, unfortunately without observational evidence of the mechanism itself. This work provides a detailed study of the proposed mechanism through measurements of size distributions and inorganic chemical composition of aerosols and blowing snow, and comparisons to modeled parameters of blowing snow SSA production. Given the prevalence of the use of the blowing snow SSA production parametrization, this is a very valuable study. My comments mainly focus on clarification of the manuscript and assessment of statistical significance throughout. Given the significant

length and many figures and tables, the authors are encouraged to consider moving some material to a supplementary information file if appropriate.

One overarching and major comment that needs to be addressed throughout the manuscript is for uncertainties (or standard deviations) to be listed with average values. This is important for assessing data variability, as well as for assessment of statistical significance. Indeed, statistical tests of significance should be applied to inform whether 'trends' and 'differences' are indeed statistically significant, which would greatly strengthen the findings presented in the manuscript. This is important because trends sometimes seem to be overstated in the text when compared to large scatter shown in the figures. Routine statements of statistical significance would significantly strengthen the conclusions throughout.

I highly recommend reorganizing the manuscript to improve readability. Section 3.2 relies significantly on depletion factors. Therefore, I recommend reorganizing to move Sections 3.4.1-3.4.3 to be before Section 3.2. Also, the current Section 3.4.3 would be best after Section 3.3.

Major Comments: Page 1, Line 21 & Page 25, Lines 14-15: These sentences state generally that 'similar processes take place in the Arctic', yet no supporting discussion is provided. Since the current work focuses on the specific conditions of the Antarctic work and no data are provided to evaluate this statement, these sentences should be removed.

Page 1, Lines 2-3 and Page 3, Lines 5-7: The statement "validating a model hypothesis to account for winter time SSA maxima in polar regions not explained otherwise" generalizes beyond the Antarctic, which is not appropriate, and it also not consider other factors, such as lower boundary layer height and lead-based SSA production. This statement should be rephrased to focus on validating wintertime SSA production from blowing snow (which is excellent), as a comprehensive discussion of wintertime SSA maxima causes in both the Arctic and Antarctic is not presented in this work. Further,

the work of Huang and Jaegle (2017) did not consider the observed influence of lead-based SSA production in the Arctic (May et al. 2016, JGR). I suggest focusing on the Antarctic, as this is the strength of this work.

Figure 1; Page 3, Lines 30-33; Page 13, Lines 22-23: Please provide a legend for sea ice concentration. It appears that stations S2, S3, and S9 were in areas of reduced sea ice concentration. While there is significant evidence for blowing snow SSA production based on chemical analyses, a discussion of the distance to open leads, in addition to open water (Page 3, Line 32), needs to be included, since there is measurement evidence of wind-dependent lead-based SSA production (e.g., Nilsson et al. 2001, JGR).

Page 7, Lines 3-5: Please clarify whether these time periods of ship exhaust influence were also removed from the aerosol size distribution data, as they should be.

Page 7, Line 9 and Table 3: LODs are normally defined as 3*sigma, rather than 2*sigma. What is the authors justification here? Also, LODs should be reported with one significant figure (too many shown in Table 3, which can be misleading).

Tables 4-5: Data below the LOD should be labeled as such, as exact values below LOQs are not meaningful.

Page 8, Lines 3-5: Instead of reporting depletion factors, I highly encourage the authors to consider reporting "enrichment factors" (e.g. Krvanek et al. 2012, Atmos. Environ.), which are more intuitive to understand in my opinion (i.e. enrichments are >1, depletion corresponds to <1).

Page 8, Lines 8-11: I am quite concerned that data were selectively removed from the datasets presented. I can understand if certain samples are not used for externally identified reasons, but if, for example, sulfate concentration is removed for a given sample, I'm concerned about continuing to use other ions from that sample, as appears to have been done based on the numbers shown in Tables 4 and 5. I worry that

the presented datasets are skewed based on the removal of these datapoints. What fraction of the time did ship emissions impact the dataset? It needs to be clarified what fraction of the data were removed. This data treatment is very important for later statements about the distribution of depletion factors (e.g., statements on Page 10, Lines 7-9).

Page 9, Lines 28-30; Page 10, Lines 1-3: Please reference where these data are presented, or please add them as supplementary information.

Section 3.4.2 and associated text in Conclusions: The authors should be mindful that only aerosol and snow bromine were measured and that no measurements of reactive bromine are presented. Therefore, the strength of the implications for reactive bromine production should be weakened to account for this uncertainty and other factors that contribution to reactive bromine production and abundance.

Page 21, Lines 9-10: Depletion factors examine the degree of depletion, but they do not provide information on the mass present. Therefore, the data here cannot assess contribution to the fraction of net bromine release, as currently presented, especially without reactive bromine measurements.

Page 19, Lines 22-25: This analysis is only valid if you assume there is no precipitation of NaCl.2H2O. Please verify that based on temperature, and perhaps take out the very low temperature points.

Page 19, Lines 127-28: Does this also mean that the aerosols collected were a mixture of sea salt emitted from the ocean and sublimation of blowing snow?

Page 22, Lines 32-33: A conversion factor is used to calculate [SSA] based on Na+ and using seawater composition, but this seems to undermine and not take into account the sulfate-depletion observed.

Page 21, Lines 30-31 and elsewhere: Is this U10m and the associated data in Fig 16 an average, or threshold? It isn't clear how the data were binned. Please clarify calm

and stormy conditions. Does calm represents U10m<5 m/s? How about stormy?

Page 22, Lines 16-19: It seems "not all water is lost" could represent a large uncertainty of blowing snow sublimation. This is important for reactions that depend on the surface area of aerosols. It could be highlighted in the abstract or conclusion. Also, please justify how to get $10^{-3}$ um. Using snow salinity of 0.06 psu from Table 5, median snow particle of 100 um from Table 6, yields d(dry) of 1 um.

Page 23, Lines 1-3: Please show this comparison and data in a supplementary file.

Page 25, Lines 5-10: This is not a new finding and has been presented in other work. Therefore, either these sentences should be removed here or other work should be referenced to further support these findings.

Data Availability: Since the current work is expect to be very valuable for informing future modeling work and other studies, I highly encourage the authors to put these data in a public archive.

Figure 7: Please add a legend to give meaning to the colors presented. Also, it is stated throughout the manuscript that the surface snow is typically significantly sulfate depleted (justifying the sea ice source for sulfate-depleted aerosol), but here the surface is more often near 0. Please clarify.

The highly relevant work of Giordano et al. (2018, ACP) "The importance of blowing snow to halogen-containing aerosol in coastal Antarctica: influence of source region versus wind speed" should be considered in this manuscript.

Minor/Technical Comments: Throughout the manuscript, watch for 'paragraphs' that are only 1-2 sentences, as this disrupts the flow and limits discussion. Consider reorganization to prevent this.

Page 1, Line 9: Please state the size of the sulphate-depleted aerosol.

Page 1, Line 13: Based on the data presented later, 'enriched' is likely a typo and

should be 'depleted' here with respect to aerosol at 29 m.

Page 2, Line 20: Provide a reference to a SSA review here.

Page 4, Lines 27-28: I think it is dividing kappa instead of multiplying. Please check. Also, please provide the value for the von Karman constant in parentheses.

Page 5, Lines 12-14: Please provide a greater description of the inlet. Also, please clarify whether the data presented where corrected for these particle loss estimates ('we adopt' is confusing phrasing).

Page 5, Lines 22 and 27: Please clarify the size range of aerosol collected.

Page 9, Line 8: I assume the authors are discussion temperature in degrees Celsius, but this needs to be stated.

Page 9, Line 14: Where is the timing of the snowfall presented/shown?

Page 9, Line 22-23: Please provide a reference that connects the friction velocity with the boundary layer conditions. Also, reference where these data are shown, or add to a SI.

Page 10, Line 16: Please clarify "two 7-10 day long periods". I'd suggest wording such as "two periods, one lasting 7 days and another 10 days", or similar.

Page 11, Lines 3-4: Please provide concentrations in parentheses for context.

Page 13, Lines 16-17: The direct comparison of $N_{0.4-12}$ to $d_p < 2$ um here is confusing since these are different size ranges.

Page 14, Line 15: Please define SWE (snow water equivalent?) and the 'saltation layer' (what height?).

Page 15, Line 3: What does "(0.001)" correspond to here? Please clarify.

Page 15, Lines 6 and 11: Please clarify that $U_t$ and $u*_t$ are calculated, not observed.

Page 15, Line 15: Please show how u*t values were calculated.

Page 15, Line 32: Please define what you mean by 'minor' here. Please quantify.

Page 17, Line 13: Please delete "have" typo.

Page 17, Line 3: Didn't mean dp increase?

Page 19, Line 32: Do you mean 0.1204 here?

Page 20, Lines 8-10: The wording "well established" should be removed, as the Yang et al papers are models based on a hypothesis rather than measurement based and this associated uncertainty should be noted.

Page 20, Line 27: Data in Table 5 are presented in ug g-1. Please fix or clarify.

Page 21, Line 11: Change "due a" to "due to a".

Page 21, Line 14: No data are presented examining the acidity of the surface snowpack.

Page 23, Line 22: Delete extra "the".

Page 23, Lines 29-30: Remove "always" and replace with "often" to more appropriately reflect the data shown.

Page 25, Line 27: "LL & MM"?

Figure 16: The variations in these distributions (e.g. standard deviations) should be shown.

Figure 17: This figure is difficult to understand currently.

---

## Author Comment (AC1) · 9 Oct 2019

**Replies to Referee Comments (RC1, RC2) and Short Comment (SC1)**

We thank both referees No.1 and No.2 for their detailed and insightful comments which helped to further improve the manuscript. Below we address all referee comments as well as the contributed short comment (in italics). Revised text, keyed to the ACPD online version, is shown in blue, and is included in the final manuscript we are submitting to ACP. Updated figures and tables as well as material for a supplement are attached at the end of our reply.

**RC1:** *Source assignment of proxies is a basic prerequisite for interpreting climate archives in terms of past climate as well as climate change. Concerning polar ice cores, ionic impurities originate primarily from aerosol deposition. Amongst them, interpretation of sea salt aerosol deposition archived in ice cores is especially challenging and contro- versial because the contribution of two different and competing sources - viz. open water versus sea ice - is up for debate. In addition it became apparent that sea salt aerosol production over ice-covered oceans may contribute significantly to the global sea salt aerosol budget. The manuscript at hand addresses this pivotal subject and provides thorough and direct observational evidence of sea salt aerosol production from blowing snow above sea ice. The important conclusions drawn are based on comprehensive state of the art ship-borne aerosol and snow measurements during winter / early springtime in the Weddell Sea region. Although the main conclusions are primarily restricted to the chosen site, there are certainly strong implications for climate research in the Southern Ocean realm and climate related interpretation of sea salt profiles from ice cores in general. The authors have accomplished a clear, well-organised and concise paper. The methodology is sound and assumptions are identified clearly and conscientiously. From my point of view, all parts, including figures, are essential. The manuscript certainly addresses the scientific scope of ACP and I recommend a final publication after some minor revisions I specified below.*

**Reply:** We thank the reviewer for the positive assessment of this work.

**RC1:** *1. Abstract, page 1, line 13 and Conclusions, page 24, line19: The authors state that bromine enrichment was typical at 29 m height, but from Chapter 3.4.2 and Fig. 15, bromine depletion is evident. Please clarify.*

**Reply:** In both cases it should read "depleted", which is now corrected.

**RC1:** *2. Chapter 2.3, Aerosol chemical composition: Could you assess the impact of pollution on chemical aerosol composition? Was the bulk aerosol sampling contamination controlled?*

**Reply:** The setup of this study did not include any contamination control of the bulk aerosol filter sampling such as pump control based on wind speed and direction. As described aerosol number concentrations at the crows nest showed significant spikes, when air came from the direction of the ship stack, whereas no evidence of pollution was detected in aerosol number concentrations observed on the sea ice. We clarified text in section 2.2 including also the fraction of CLASP data filtered out:

Raw aerosol number concentrations at the crow's nest showed significant spikes, when air came from the direction of the ship's engine stack, whereas no evidence of pollution was detected in the observations on the sea ice. Pollution spikes were effectively filtered out prior to averaging by excluding all data when relative wind direction was in the 135–225° sector encompassing the ship's engine stack. A total of 21% of the available 1-second data was removed from the crow's nest data.

We now include an assessment of the potential impact of pollution on bulk aerosol chemistry from filters and added the following text to section 2.3:

In order to assess the impact of potential pollution on bulk aerosol chemistry from filters collected at the crow's nest we calculated for each filter sample the fraction of the total filter run time during which relative wind direction was within the 135–225° sector encompassing the ship's engine stack. Considering all filters sampled from June to September 2013 (N=141) the fraction of total filter run time with winds from the polluted sector was on average 9.5%. Polluted time fraction and atmospheric concentrations of $Na^+$, $Cl^-$, $SO_4^{2-}$ and $Br^-$ did not show any correlation ($R^2 < 0.05$), suggesting that the impact of pollution on the respective ion concentrations is small. A weak, but significant negative correlation was found between polluted time fraction and depletion factors $DF_{SO_4^{2-}}$ ($R^2 = 0.19$, p<0.01)

and DF$_{Br^-}$ (R$^2$=0.13, p<0.01) suggesting that enrichment in sulphate (and bromide) may be more likely during polluted conditions. The bulk aerosol chemistry observations on the sea ice showed no evidence of pollution. Thus, in the case of sulfate we cannot rule out that some of the sulfate enrichment in atmospheric aerosol observed at the crow's nest may be due to ship exhaust rather than presence of mirabilite. It follows that estimates of sea ice contributions to total SSA derived from depletion factors discussed in section 3.4.4 have to be considered as lower bounds of true values.

**RC1:** *3. Chapter 3.2: Impact of snow precipitation on blowing/drifting snow: Did you access the regular weather reports from the ships meteorological office in this case?*

**Reply:** The available 3-hourly weather reports from the ship provided only limited information, but confirmed for the case of precipitation shown in Fig.10 (3-4 July 2013) overcast skies, variable visibility from 0.5 (fog) to 10 km, and occasional ice needles. As stated in the text we inferred occurrence of precipitation qualitatively from direct observation supported by webcam images, if usable, and presence of clouds (p9-l14). The still images of a webcam installed at the crows nest (p8-l16) allowed to see at times, including 4 July, large airborne snow crystals during night time in the beam of the ships search lights. We added a sentence in section 3.3.2 to further clarify.

For the early morning of 4 July 2013 webcam images from the crow's nest confirmed the presence of large airborne snow crystals visible during darkness in the beam of the ship's search lights, whereas the ship's 3-hourly weather report noted the presence of airborne ice needles.

**RC1:** *4. Pages 12/13 and Fig.7: Regarding the salinity ($S_p$) of blowing snow, corresponding $S_p$-values of the uppermost surface snow layer are decisive. Did you take samples from surface snow; say <1 cm deepness below surface? Figure 7: The reader cannot get an idea about the salinity of the surface snow layer from this graph. It would be informative as well to specify the total depth of the snow layer shown here, not just the snow height above sea ice.*

**Reply:** Typically snow pit profiles were measured at 2 cm depth resolution (see methods section 2.4); except at ice station S6 some profiles include a surface snow sample from a layer of ~0.5-1.0 cm thickness. As discussed, $S_p$ in blowing snow is consistent with the local surface snow measurements (section 3.2.2). In Figure 7 depth information is readily available since data points at the top of each profile, i.e. with the largest snow height above ice, represent the surface snow layer. We updated Figure 7 and caption accordingly (Fig. 1).

**RC1:** *5. Chapter 3.3.2, Snow particle size distribution: Is it possible to rate the impact of the ships profile on the local wind field and eventually on the measured snow particle size distribution?*

**Reply:** We did not attempt to quantify the distortion of the local wind field by the ship and its impact on measured snow particles, and had therefore included a respective caveat in the method description (p4 line 30): "It should be borne in mind that the distortion of flow caused by the ship may mean both that speed at 39 m is not representative of flow in the far field at that height, and further, the turbulent field strength, which governs the gradient of the logarithmic profile, may be a residual from a different, likely lower, height. Thus, we suggest care when interpreting the data, and estimate that the conversion from particle counts to number density be seen as an estimate suitable for comparison, rather than quantitative with a well behaved uncertainty."

**RC1:** *6. Chapter 3.4.1, page 19, lines 28-30: As for Antarctic winter, acid induced Cl$^-$ loss is rather extraordinary because production of acidic sulphur compounds usually cease at the end of summer / fall. Are there any indications for alternative HNO$_3$ induced Cl$^-$ loss in your data?*

**Reply:** We agree. Sea salt reaction with atmospheric HNO$_3$ is a plausible alternative chloride loss process in winter; e.g. at Halley on the nearby Antarctic coast observations in winter show low but non-zero levels of atmospheric HNO$_3$ of 1-2 pptv (Jones et al., 2011). Unfortunately, no usable filter data of aerosol nitrate are available from this study to test the suggested process due to a very high lab procedure blanc. However, we include the point in section 3.4.1 as follows.

Snow on sea ice follows closely the theoretical mirabilite fractionation line, whereas aerosol shows large scatter and a tendency to apparent Na$^+$ enrichment with respect to Cl$^-$ of up to 20 %, equivalent to Cl⁻ depletion with respect to Na⁺ of 17 % (Fig. 14). Dechlorination of sea salt aerosol observed in Antarctica has a maximum in spring/summer, when gaseous acidic species (nitric, sulfuric and methanesulfonic acid) are available to replace chloride on sea-salt aerosol (Wagenbach et al., 1998; Rankin and Wolff, 2003; Legrand et al., 2017). Acidic sulphur species are close to zero during winter in coastal Antarctica e.g. at Neumayer (Weller et al., 2011), whereas nitric acid is low but non-zero, e.g. 1-2 pptv at Halley (Jones et al., 2011). Thus nitric acid induced Cl⁻ loss from sea salt is a plausible explanation for the observed Cl⁻ depletion either in airborne SSA or as a sampling artefact from sea salt already accumulated on the filter surface as suggested previously (Wagenbach et al., 1998; Legrand et al., 2017). Unfortunately no usable filter data of aerosol nitrate are available from this study to further test the association between nitrate and sea salt due to a very high lab procedure blank.

**RC1:** *7. Chapter 3.4.2, Chemical fractionation of Br⁻, lines 28-33 and Fig. 15: There is strong bromine depletion during polar night in July when global radiation was about zero (Fig. 15b). This peculiarity deserves some discussion.*

**Reply:** We did mention (p21-line16) that bromide escape from aerosol was detected previously year-round at DDU in coastal Antarctica, including during winter months, except in June (Legrand et al., 2016). As suggested, we expand the discussion in section 3.4.2:

Contrary to expectation bromide depletion of aerosol was significant even during winter darkness from mid June to mid July (Fig. 15b), whereas previous observations at DDU showed a similar trend but less bromide depletion and none in June (Legrand et al., 2016). At DDU $DF_{Br^-}$ in bulk aerosol increased gradually from a minimum in June (0.04), intermediate values in July to Sep (0.22-0.39) to a maximum in October (0.42) (Legrand et al., 2016). Light conditions are unlikely a cause of differences in bromide depletion, since DDU is located at a similar latitude (66° 40'S) as the area covered by this study. However, one of a number of processes identified leading to bromide loss from snow or aerosol involves HOBr oxidation of bromide, which leads to its autocatalytic release (Abbatt et al., 2012). The early laboratory study by Oum et al. (1998) has shown that the required HOBr can be chemically produced in darkness through the reaction of ozone with bromide. Another study during the ANT-XXIX/6 expedition reports significant bromoform ($CH_3Br$) production in sea ice during winter darkness (Abrahamsson et al., 2018), which requires HOBr (and organic matter) as precursors, and therefore indicates that bromine loss processes were active in the sea ice in the absence of sunlight. It therefore appears plausible that the same reactions may have caused significant bromide depletion observed here in sea salt aerosol, provided the aerosol pH was low enough.

**RC1:** *8. Figure 5 and page 12, lines 8-9: By the way: During late afternoon of the 11 July, there is an outstanding Na⁺ peak associated with corresponding sulphate depletion, while the wind speed seemed just close to the threshold value (well below 10 m/s throughout the whole day). Any ideas?*

**Reply:** From midnight to the early morning of 11 July 2013 wind speed was indeed at or slightly above the snow drift threshold (= 7.1 m/s) (Fig.5a) suggesting that drifting snow near the surface was present and after sublimation contributed to the observed sodium peak. The increase in SSA number densities and atmospheric sodium occurred in the afternoon a few hours after wind speed had dropped again below the threshold, consistent with a similar phasing observed during the blowing snow event on 14-16 July and discussed in section 3.2.2. To better illustrate episodes of snow drift we include in Figures 4-6 of the revised manuscript a horizontal line marking the estimated threshold wind speed $U_t$.

**RC2:** *Frey et al present an observational study of sea salt aerosol (SSA) production from blowing snow above sea ice, through measurements during winter 2013 in the Weddell Sea, Antarctica. Since the modelling hypothesis presented by Yang et al (2008, GRL), the mechanism of SSA production from blowing snow has been implemented in numerous modelling studies, unfortunately without observational evidence of the mechanism itself. This work provides a detailed study of the proposed mechanism through measurements of size distributions and inorganic chemical composition of aerosols and blowing snow, and comparisons to modelled parameters of blowing snow SSA production. Given the prevalence of the use of the blowing snow SSA production parameterisation, this is a very valuable study.*

**Reply:** We thank the reviewer for the positive assessment.

**RC2:** *My comments mainly focus on clarification of the manuscript and assessment of statistical significance throughout. Given the significant length and many figures and tables, the authors are encouraged to consider moving some material to a supplementary information file if appropriate.*

**Reply:** We consider tables and figures all essential, in agreement with reviewer 1. However, additional material as suggested is now included and presented in a supplement (see below).

**RC2:** *One overarching and major comment that needs to be addressed throughout the manuscript is for uncertainties (or standard deviations) to be listed with average values. This is important for assessing data variability, as well as for assessment of statistical significance. Indeed, statistical tests of significance should be applied to inform whether 'trends' and 'differences' are indeed statistically significant, which would greatly strengthen the findings presented in the manuscript. This is important because trends sometimes seem to be overstated in the text when compared to large scatter shown in the figures. Routine statements of statistical significance would significantly strengthen the conclusions throughout.*

**Reply:** We added standard deviations for all averages reported in the tables (Tables 1, 2) and include in the final manuscript significance of trends and differences, where appropriate. As an example below updated text in section 3.4.4.

During storms median atmospheric sea salt concentrations from both estimates showed increases above background values (Fig. 17a) that were statistically significant based on the Wilcoxon rank-sum test ($p<0.01$).

**RC2:** *I highly recommend reorganising the manuscript to improve readability. Section 3.2 relies significantly on depletion factors. Therefore, I recommend reorganising to move Sections 3.4.1-3.4.3 to be before Section 3.2. Also, the current Section 3.4.3 would be best after Section 3.3.*

**Reply:** We are considering this suggestion for the final manuscript.

**RC2:** *Major Comments: Page 1, Line 21 & Page 25, Lines 14-15: These sentences state generally that 'similar processes take place in the Arctic', yet no supporting discussion is provided. Since the current work focuses on the specific conditions of the Antarctic work and no data are provided to evaluate this statement, these sentences should be removed.*

**Reply:** Agreed. We removed the sentence referring to the Arctic from abstract and conclusions, and added text in the conclusions as follows:

Similar in situ measurements are needed to corroborate the importance of sea salt aerosol production from blowing snow also in the Arctic to validate atmospheric and ice core models (e.g. Rhodes et al., 2017; Huang and Jaeglé, 2017).

**RC2:** *Page 1, Lines 2-3 and Page 3, Lines 5-7: The statement 'validating a model hypothesis to account for winter time SSA maxima in polar regions not explained otherwise' generalises beyond the Antarctic, which is not appropriate, and it also not consider other factors, such as lower boundary layer height and lead-based SSA production. This statement should be rephrased to focus on validating wintertime SSA production from blowing snow (which is excellent), as a comprehensive discussion of wintertime SSA maxima causes in both the Arctic and Antarctic is not presented in*

*this work. Further, the work of Huang and Jaegle (2017) did not consider the observed influence of lead-based SSA production in the Arctic (May et al. 2016, JGR). I suggest focusing on the Antarctic, as this is the strength of this work.*

**Reply:** We agree and rephrased in abstract (1.) and introduction (2.) accordingly.

1. Two consecutive cruises in the Weddell Sea, Antarctica, in winter 2013 provided the first direct observations of sea salt aerosol (SSA) production from blowing snow above sea ice, thereby validating a model hypothesis to account for winter time SSA maxima in the Antarctic. 2. Indeed, model agreement with SSA winter maxima observed at a number of locations in the polar regions is much improved when a SSA source from blowing snow based on the parameterisation of (Yang et al., 2008) is included in the model (Huang and Jaeglé, 2017; Yang et al., 2019).

**RC2:** *Figure 1; Page 3, Lines 30-33; Page 13, Lines 22-23: Please provide a legend for sea ice concentration. It appears that stations S2, S3, and S9 were in areas of reduced sea ice concentration. While there is significant evidence for blowing snow SSA production based on chemical analyses, a discussion of the distance to open leads, in addition to open water (Page 3, Line 32), needs to be included, since there is measurement evidence of wind-dependent lead-based SSA production (e.g., Nilsson et al. 2001, JGR).*

**Reply:** A legend is now included in Fig. 1 and we added the following text in section 2 (1.) and in the discussion p13 - after line26 (2.).

1. Sea ice concentrations in mid July 2013 derived from Nimbus-7 satellite microwave radiometer measurements (Comiso, 2018) show areas with 85-95 % ice cover near ice stations S2-3 and S7-9 indicating that open leads may be present (Figure 1). 2. Open leads, which may have been present in areas of reduced sea ice concentration e.g. near ice stations S2-3 and S7-9 (Figure 1) are another potential wind-dependent source of SSA from open water as observed in the Arctic (Nilsson et al., 2001; May et al., 2016), albeit with a much smaller flux contribution per surface area compared to the open ocean due to reduced fetch and low fraction of surface coverage (<15 %).

**RC2:** *Page 7, Lines 3-5: Please clarify whether these time periods of ship exhaust influence were also removed from the aerosol size distribution data, as they should be.*

**Reply:** Ship exhaust influence on measurements of aerosol size and concentration was removed by using a wind-sector filter (section 2.2). We clarified text in section 2.2 including also the fraction of CLASP data filtered out as follows (see corresponding reply to RC1):

Raw aerosol number concentrations at the crow's nest showed significant spikes, when air came from the direction of the ship's engine stack, whereas no evidence of pollution was detected in the observations on the sea ice. Pollution spikes were effectively filtered out prior to averaging by excluding all data when relative wind direction was in the 135–225° sector encompassing the ship's engine stack. A total of 21% of the available 1-second data was removed from the crow's nest data.

**RC2:** *Page 7, Line 9 and Table 3: LODs are normally defined as 3*sigma, rather than 2*sigma. What is the authors justification here? Also, LODs should be reported with one significant figure (too many shown in Table 3, which can be misleading).*

**Reply:** Here we follow Wagenbach et al. (1998) who employed a similar aerosol filter method and defined the mean detection limits as 2*sigma. Two figures for LOD were reported in Table 3 to account for increased LOD at the shorter run times of filters deployed on the sea ice. However, we removed that line to report only one figure for LOD and clarified the footnote of Table 3 as follows.

[c]based on crow's nest mean air sample STP-volume ($6.4\,\text{m}^3$); mean air sample STP-volume for filters deployed on the sea ice was $3.3\,\text{m}^3$ increasing respective LODs by a factor 1.6

**RC2:** *Tables 4-5: Data below the LOD should be labeled as such, as exact values below LOQs are not meaningful.*

**Reply:** Agreed. Snow concentrations (Tab. 2) were typically 2 orders of magnitude above the LOD of $\sim 2\,\text{ng}\,\text{g}^{-1}$, whereas some aerosol concentrations (Tab. 1) where below the estimated LOD. In the final manuscript a corresponding footnote is added to those values in Table 4:

[c]below the estimated LOD (see Table 3)

**RC2:** *Page 8, Lines 3-5: Instead of reporting depletion factors, I highly encourage the authors to*
*consider reporting 'enrichment factors' (e.g. Krvanek et al. 2012, Atmos. Environ.), which are more*
*intuitive to understand in my opinion (i.e. enrichments are >1, depletion corresponds to <1).*
**Reply:** Deviations from bulk sea water ion rations are reported in the literature in both ways, either
as enrichment or as depletion factors (e.g. Sander et al., 2003). Since the focus here is on depletion
processes we choose to report depletion rather than enrichment factors, also to be consistent with
some of the previous related work (Yang et al., 2008). To help interpretation we added a sentence to
section 2.4:
For example, $DF_x$ =-1.5 or 150 % enrichment means the respective ion concentration is 2.5 times
that in reference sea water.

**RC2:** *Page 8, Lines 8-11: I am quite concerned that data were selectively removed from the datasets*
*presented. I can understand if certain samples are not used for externally identified reasons, but if,*
*for example, sulfate concentration is removed for a given sample, I'm concerned about continuing*
*to use other ions from that sample, as appears to have been done based on the numbers shown in*
*Tables 4 and 5. I worry that the presented datasets are skewed based on the removal of these data*
*points. What fraction of the time did ship emissions impact the dataset? It needs to be clarified what*
*fraction of the data were removed. This data treatment is very important for later statements about*
*the distribution of depletion factors (e.g., statements on Page 10, Lines 7-9).*
**Reply:** No snow data were removed whereas the fraction of aerosol filter data removed was relatively
small, and is now mentioned in the revised text. Filter samples suspected of contamination based
on anomalous sulfate enrichment (total of 6 samples) are not anymore used in the statistics. The
pollution impact on filter chemistry is now discussed (see reply to RC1 above). Bromide depletion
factors below a threshold of -7 are considered outliers and removed. The corresponding statistics in
Table 4 are updated. Follow up statements are not affected by any of these changes.
A total of 6 (= 6% of all crow's nest samples) $DF_{SO_4^{2-}}$ values were below that of pure mirabilite (=
-7.3) and are attributed either to sulfate contamination from the ship's engine emissions discussed
below or measurement error. We therefore removed all ion concentrations of the corresponding filter
samples from the dataset. $DF_{Br^-}$ only below -7 were considered outliers due to measurement error
and removed: a total of 4 (= 3% of all samples) from the crow's nest data, and a total of 6 (= 14% of
all samples) from the sea ice data.

**RC2:** *Page 9, Lines 28-30; Page 10, Lines 1-3: Please reference where these data are presented,*
*or please add them as supplementary information.*
**Reply:** Agreed. In a supplement we include now a Figure S1 (Fig. 2) with an overview of the available
observations during ANT-XXIX/7, and Table S1 and Table S2 (Table 3, 4) with the statistics of particle
concentration and size. The text has been amended as follows:
1. At 29 m mean total number densities $N_{46-478}$ were $8.7 \times 10^3$ m$^{-3}$ during ANT-XXIX/6 and very similar
$7.2 \times 10^3$ m$^{-3}$ during ANT-XXIX/7 (Table S1, Figure S1c).
2. At 29 m mean total number densities $N_{0.4-12}$ were $2.1 \times 10^6$ m$^{-3}$ during ANT-XXIX/6 (Table S2, Fig-
ure 2d). $N_{0.4-12}$ mean values at 2.0 and 0.2 m during ice stations were $1.4 \times 10^6$ and $1.7 \times 10^6$ m$^{-3}$,
respectively, about the same as the number densities observed during the same time at 29 m (Ta-
ble S2). The median aerosol particle diameters $\overline{d_p}$ at the measurement heights 0.2, 2.0m and 29 m
ranged between 0.60 and 0.66 μm (Table S2) showing dominance of sub-micron sized particles in
atmospheric aerosol below the instrument particle size cut-off (>11 μm).
3. Median $DF_{SO_4^{2-}}$ values at 29 m were very similar during ANT-XXIX/6 (=0.34) and ANT-XXIX/7
(=0.30), but larger near the sea ice surface (=0.49), suggesting throughout a significant contribution
to the total SSA burden from a fractionated sea ice source (Table 4, Figure S1e).

**RC2:** *Section 3.4.2 and associated text in Conclusions: The authors should be mindful that only*
*aerosol and snow bromine were measured and that no measurements of reactive bromine are pre-*

*sented. Therefore, the strength of the implications for reactive bromine production should be weakened to account for this uncertainty and other factors that contribution to reactive bromine production and abundance.*

**Reply:** Indeed, we do not infer any details on speciation of reactive bromine chemistry. We added a sentence in section 3.4.2 (1.) and amended a sentence in conclusions (2.):

1. Detailed measurements of participating bromine species in air, snow and aerosol are needed to further understand relevant processes and constrain the mass budget.

2. It is found that SSA produced by blowing snow is depleted in bromide suggesting it is a source of reactive bromine to the atmosphere, which then can contribute to ozone depletion events.

**RC2:** *Page 21, Lines 9-10: Depletion factors examine the degree of depletion, but they do not provide information on the mass present. Therefore, the data here cannot assess contribution to the fraction of net bromine release, as currently presented, especially without reactive bromine measurements.*

**Reply:** Agreed, we don't discuss detail of the bromine mass budget. We amended the corresponding sentence in section 3.4.2, conclusions and abstract (1.), as well added a note (2.) (see reply to previous comment):

1. On average snow on sea ice and blowing snow showed no or small depletion of bromide relative to sodium with respect to sea water, whereas aerosol at 29 m was depleted suggesting that significant bromine loss takes place in the aerosol phase between 2 and 29 m above the sea ice surface.

2. Detailed measurements of participating bromine species in air, snow and aerosol are needed to further understand relevant processes and constrain the mass budget.

**RC2:** *Page 19, Lines 22-25: This analysis is only valid if you assume there is no precipitation of NaCl.2H2O. Please verify that based on temperature, and perhaps take out the very low temperature points.*

**Reply:** A complete model of freezing seawater is beyond the scope of this study. Thus we acknowledge that precipitation of NaCl.2H2O introduces some uncertainty to this analysis by adding the sentence below.

Further $Na^+$ depletion may arise from the precipitation of hydrohalite ($NaCl \cdot 2\,H_2O$) once ambient temperature drops below the threshold of -22.9 °C (e.g. Butler et al., 2016), which occurred here during some periods of time (Fig.2b). In the analysis below however we consider only the precipitation of mirabilite.

**RC2:** *Page 19, Lines 127-28: Does this also mean that the aerosols collected were a mixture of sea salt emitted from the ocean and sublimation of blowing snow?*

**Reply:** Mixing with a pool of non-fractionated sea salt aerosol from the open ocean ($DF_{Na^+}$, $DF_{SO_4^{2-}}$=0) would move data points towards the origin in Figure 14, but would not explain apparent $Na^+$ enrichment or $Cl^-$ loss in aerosol at a given $SO_4^{2-}$ depletion. We believe a plausible explanation for the deviation of aerosol observations from the mirabilite precipitation model is $HNO_3$ induced $Cl^-$ loss from sea salt either in airborne SSA or as an artefact on filters, as stated in the reply to reviewer 1. Below we repeat the amended text.

Snow on sea ice follows closely the theoretical mirabilite fractionation line, whereas aerosol shows large scatter and a tendency to apparent $Na^+$ enrichment with respect to $Cl^-$ of up to 20 %, equivalent to $Cl^-$ depletion with respect to $Na^+$ of 17 % (Fig. 14). Dechlorination of sea salt aerosol observed in Antarctica has a maximum in spring/summer, when gaseous acidic species (nitric, sulfuric and methanesulfonic acid) are available to replace chloride on sea-salt aerosol (Wagenbach et al., 1998; Rankin and Wolff, 2003; Legrand et al., 2017). Acidic sulphur species are close to zero during winter in coastal Antarctica e.g. at Neumayer (Weller et al., 2011), whereas nitric acid is low but non-zero, e.g. 1-2 pptv at Halley (Jones et al., 2011). Thus nitric acid induced $Cl^-$ loss from sea salt is a plausible explanation for the observed $Cl^-$ depletion either in airborne SSA or as a sampling artefact from sea salt already accumulated on the filter surface as suggested previously (Wagenbach et al., 1998; Legrand et al., 2017). Unfortunately no usable filter data of aerosol nitrate are available from this study to further test the association between nitrate and sea salt due to a very high lab procedure blank.

**RC2:** *Page 22, Lines 32-33: A conversion factor is used to calculate [SSA] based on Na+ and using*
*seawater composition, but this seems to undermine and not take into account the sulfate-depletion*
*observed.*

**Reply:** It does not. The impact of the depletion due to mirabilite precipitation on our calculation is
indeed very small, and is therefore neglected. We added the text below to clarify:

As shown in section 3.4.1 depletion of $SO_4^{2-}$ due to the precipitation of mirabilite decreases $Na^+$ by up
to 12 %. Reduction in both ions decreases the mass fraction of $Na^+$ in the depleted sea salt aerosol
by a maximum of ~0.7 % compared to reference seawater. Thus, by not considering the depletion
effect conversion factor and calculated SSA mass are underestimated by up to ~0.7 %, which is neg-
ligible given all other uncertainties.

**RC2:** *Page 21, Lines 30-31 and elsewhere: Is this U10m and the associated data in Fig 16 an aver-*
*age, or threshold? It isn't clear how the data were binned. Please clarify calm and stormy conditions.*
*Does calm represents U10m<5 m/s? How about stormy?*

**Reply:** We used a relatively narrow wind speed range for calm and windy conditions. We amended
this to include more data, particularly for the open ocean case when only a few days of measurements
were available. Aerosol data are now selected based on a wind speed threshold: calm conditions
when $U_{10m}$ <4 m s$^{-1}$ and windy conditions when $U_{10m}$ >9 m s$^{-1}$. We updated Figure 16, including also
the standard deviation of the mean, to show statistical significance of differences in size distributions,
as suggested further below (Fig. 3). And the text in section 3.4.3 is clarified as follows:

Average aerosol number density and volume distributions observed in the Weddell sea show that
during calm conditions ($U_{10m}$<4 m s$^{-1}$) concentrations across most of the size spectrum were smaller
above sea ice than above the open ocean (Fig. 16a). Depending on particle size the variability was
relatively large as illustrated by the standard deviation of the mean values (Fig. 16a). Thus differ-
ences in mean size distributions were statistically significant only for $d_p$ <2 μm in the case of aerosol
number density, and $d_p$ 1-8 μm in the case of aerosol volume distributions (Fig. 16b). The wind speed
threshold chosen for calm conditions is well below the mean snowdrift threshold wind speed $U_t$ of
7.1 m s$^{-1}$ observed during this study and within the range when breaking of waves commences (3-
4 m s$^{-1}$; O'Dowd et al., 1997). ... During stormy conditions ($U_{10m}$>9 m s$^{-1}$) average aerosol number
densities above sea ice increased significantly for particle diameters $d_p$<2 μm, reaching at the lower
end of the size spectrum levels similar to those observed above the open ocean (Fig. 16a). Average
aerosol volume concentrations above sea ice also showed an increase during storms, significant for
particle sizes $d_p$ 0.8 to 9 μm (Fig. 16b).

**RC2:** *Page 22, Lines 16-19: It seems "not all water is lost" could represent a large uncertainty of blow-*
*ing snow sublimation. This is important for reactions that depend on the surface area of aerosols.*
*It could be highlighted in the abstract or conclusion. Also, please justify how to get 10$^{-3}$ μm. Using*
*snow salinity of 0.06 psu from Table 5, median snow particle of 100 um from Table 6, yields d(dry) of*
*1 um.*

**Reply:** We agree the degree to which water ice is lost on particles during sublimation has implica-
tions for heterogeneous chemistry, something future experiments will need to address; text below has
been added to the conclusions.

The degree of water ice loss from particles has implications for particle surface area and heteroge-
neous chemistry, which future experiments will need to address.

**Reply:** We disagree regarding the calculation of $d_{dry}$: to convert $S_p$ from psu (equivalent to g of
dissolved salt per kg of sea water as defined in Section 2.4) into units of kg per kg in order to be
consistent with units of density (kg of salt per m$^3$ of salt) requires division by one thousand as the
equation states (Page 22, Line 16), correctly yielding $d_{dry}$ of ~1 nm.

**RC2:** *Page 23, Lines 1-3: Please show this comparison and data in a supplementary file.*

**Reply:** We included Figure S3 in the supplement to show the comparison (Fig. 4), and amended the sentence as follows:

The sea salt mass estimates show that most filter-based values have a low bias compared to median sea salt concentrations derived from $N_{0.4-12}$ during filter sampling intervals (Fig. S3), on average of ~26 %. The bias shows also a weak but significant positive correlation wind speed (R=0.4, p<0.01) (Fig. S3). A low bias of the filter samples especially during high wind speeds is expected because the smaller cut-off diameter (<6 $\mu$m) compared to the optical particle counter (>11 $\mu$m) limits capture of coarse sea salt aerosol, where much of the particle mass is located (Fig. 16b).

**RC2:** *Page 25, Lines 5-10: This is not a new finding and has been presented in other work. Therefore, either these sentences should be removed here or other work should be referenced to further support these findings.*

**Reply:** Presenting the links between snow salinity, differences in sea ice age and SSA source strength of blowing snow together with direct observations is of course new. However, we reference relevant work on sea ice and snow on sea ice as follows:

- at a given salt migration distance from the sea ice surface it is total snowpack depth, that determines the salinity probability distribution of snow on sea ice consistent with previous studies (Domine et al., 2004; Massom et al., 2001). FYI can therefore be distinguished from MYI based on snow salinity, because snow on FYI is in general more shallow than on MYI. Secondary factors potentially increasing the difference in salinity between FYI and MYI and identified previously (e.g. Massom et al., 2001) are more frequent flooding of FYI with seawater due to negative freeboard and MYI desalination due to brine drainage.

**RC2:** *Data Availability: Since the current work is expect to be very valuable for informing future modelling work and other studies, I highly encourage the authors to put these data in a public archive.*

**Reply:** All data from this study used are stored in the UK Polar Data Centre. The DOI is provided in the final manuscript.

All data are stored in the UK Polar Data Centre, Natural Environment Research Council, UK Research and Innovation (https://doi.org/10.5285/853dd176-bc7a-48d4-a6be-33bcc0f17eeb, Frey et al., 2019).

**RC2:** *Figure 7: Please add a legend to give meaning to the colors presented. Also, it is stated throughout the manuscript that the surface snow is typically significantly sulfate depleted (justifying the sea ice source for sulfate-depleted aerosol), but here the surface is more often near 0. Please clarify.*

**Reply:** Figure 7 now includes a legend (Fig. 1). There is significant spatial heterogeneity in the sampled local snowpack profiles, whereas blowing snow integrates over a wider area of sea ice. We clarified the discussion of the snow pit observations (Page 13 - Lines 9-14) as follows:

$DF_{SO_4^{2-}}$ profiles exhibited large scatter: except at one location surface-near snow showed no or small depletion, whereas most profiles showed significant depletion in deeper layers within 5-10 cm of the sea ice surface (Fig. 7c). ... However, the $DF_{SO_4^{2-}}$ values of blowing snow were at the top end of the range observed only in the deeper and more saline local snowpack (Fig. 6c). A plausible explanation for this observation during the storm on 15 July is that blowing snow integrates snow contributions from a wider area. And given the spatial heterogeneity of local snowpack thickness and composition blowing snow contributions must have dominated from areas where fractionated snow was at or near the surface such as seen in one of the profiles sampled on 12 July (Fig. 7c).

**RC2:** *The highly relevant work of Giordano et al. (2018, ACP) 'The importance of blowing snow to halogen-containing aerosol in coastal Antarctica: influence of source region versus wind speed' should be considered in this manuscript.*

**Reply:** We agree and correct the oversight by referring to this work in the introduction:

A recent observational study in the Ross Sea sector of coastal Antarctica also shows a significant association between increased SSA and high wind speed suggesting a link to blowing snow above sea ice as a source (Giordano et al., 2018).

**RC2:** *Minor/Technical Comments: Throughout the manuscript, watch for 'paragraphs' that are only 1-2 sentences, as this disrupts the flow and limits discussion. Consider reorganization to prevent this.*

**Reply:** We reorganised, where appropriate.

**RC2:** *Page 1, Line 9: Please state the size of the sulphate-depleted aerosol.*

**Reply:** Done.

Similar depletion in bulk aerosol observed in the 1-6 $\mu$m range suggests that most sea salt originated from snow on sea ice and not the open ocean or leads, e.g. on average $\sim$93% during the 8 June and 12 August 2013 period.

**RC2:** *Page 1, Line 13: Based on the data presented later, 'enriched' is likely a typo and should be 'depleted' here with respect to aerosol at 29 m.*

**Reply:** This is now corrected (see reply to reviewer 1 above).

**RC2:** *Page 2, Line 20: Provide a reference to a SSA review here.*

**Reply:** The reference below is now included.

de Leeuw et al. (2011)

**RC2:** *Page 4, Lines 27-28: I think it is dividing kappa instead of multiplying. Please check. Also, please provide the value for the von Karman constant in parentheses.*

**Reply:** Corrected as follows:

To do this a logarithmic wind profile $U(z)$ is assumed given by $U(z) = u_*/\kappa \ln(z/z_0)$ (e.g. Li and Pomeroy, 1997), with measurement height $z$, the von Karman constant $\kappa$ (= 0.4), friction velocity $u_*$ and the surface roughness length of momentum $z_0$ set to $5.6 \times 10^{-5}$ m as measured very consistently above snow at Halley (King and Anderson, 1994).

**RC2:** *Page 5, Lines 12-14: Please provide a greater description of the inlet. Also, please clarify whether the data presented where corrected for these particle loss estimates ('we adopt' is confusing phrasing).*

**Reply:** Clarified as follows.

Particle losses to inlet walls are minimised by using a short and straight inlet tube of 0.3 m length similar to the original configuration (Hill et al., 2008, Figure 9). We assume as an upper limit of particle losses those estimated previously for a similar inlet configuration (Norris et al., 2012), which amount to 43% at $d_p$ = 11.32 $\mu$m, 19% at $d_p$ = 6.06 $\mu$m and 0.1% at $d_p$ = 0.44 $\mu$m, respectively.

**RC2:** *Page 5, Lines 22 and 27: Please clarify the size range of aerosol collected.*

**Reply:** Clarified as follows.

Filters were estimated to collect aerosol in the diameter range $\sim$0.3 $\mu$m to less than 6 $\mu$m. The lower end of the range is based on previous measurements of collection efficiencies of PTFE filters as a function of particle size (Soo et al., 2016), whereas the upper end is based on the estimated cut-off diameter described below.

**RC2:** *Page 9, Line 8: I assume the authors are discussion temperature in degrees Celsius, but this needs to be stated.*

**Reply:** Added.

Near-zero or positive ambient temperatures $T_a$ in degrees Celsius ...

**RC2:** *Page 9, Line 14: Where is the timing of the snowfall presented/shown?*

**Reply:** Only the timing of airborne snow particles is shown. Occurrence of precipitation is based on 3-hourly ship's weather reports and occasional webcam images. We rephrased accordingly.

Winter storms occurred frequently with wind speeds ranging between 10 and 20 m/s, occasionally exceeding 20 m/s, and coincided often with snowfall based on the ship's 3-hourly weather report, occasional webcam images and presence of clouds (data not shown).

**RC2:** *Page 9, Line 22-23: Please provide a reference that connects the friction velocity with the boundary layer conditions. Also, reference where these data are shown, or add to a SI.*

**Reply:** We included Jacobson (2005) and Nishimura and Nemoto (2005) as references, as well as a figure (Fig. 5) in the supplement showing the correlation between friction velocity $u_*$ and horizontal wind speed $U$.

**RC2:** *Page 10, Line 16: Please clarify 'two 7-10 day long periods'. I'd suggest wording such as 'two periods, one lasting 7 days and another 10 days', or similar.*

**Reply:** Added as follows.

Two periods, one lasting 7 days and another 10 days, were chosen based on data coverage to discuss key features of observed blowing snow and associated SSA increases.

**RC2:** *Page 11, Lines 3-4: Please provide concentrations in parentheses for context.*

**Reply:** Clarified as follows.

Near the surface spectral number densities $N_{0.4-12}$ for particles with $d_p<2\,\mu$m during the storm on 24 June remained with $10^5$ m$^{-3}$ below those seen at 29 m ($10^6$ m$^{-3}$) likely due to scavenging of aerosol by snow particles (Fig. 4d-e).

**RC2:** *Page 13, Lines 16-17: The direct comparison of $N_{0.4-12}$ to $d_pp<2\mu m$ here is confusing since these are different size ranges.*

**Reply:** Clarified as follows.

Aerosol size spectra show that number densities of particles with size $d_p<2\,\mu$m increased during individual storms by 2-3 orders of magnitude above background levels.

**RC2:** *Page 14, Line 15: Please define SWE (snow water equivalent?) and the 'saltation layer' (what height?).*

**Reply:** Amended as follows.

1. (mm day$^{-1}$ snow water equivalent) 2. The saltation layer is a layer just above the snow surface usually several centimetres thick (e.g. Déry and Yau, 1999).

**RC2:** *Page 15, Line 3: What does '(0.001)' correspond to here? Please clarify.*

**Reply:** Amended as follows.

... when snow drift density $\mu$ right above the snow surface exceeds a critical value $\mu_c$ (= 0.005 kg m$^{-3}$). For comparison a lower value of $\mu_c$ (= 0.001 kg m$^{-3}$) is also considered.

**RC2:** *Page 15, Lines 6 and 11: Please clarify that Ut and u*t are calculated, not observed.*

**Reply:** $U_t$ and u$^*$ are not calculated. Windspeed and snow particle number densities are both measured quantities; thus drift threshold wind speed is an observed quantity based on the combination of two measurements (symbols in Figure 8) as opposed to modelled values (Eq 4). Similar for friction velocity u$^*$. We added a sentence to clarify.

The observed threshold wind speed $U_t$ and friction velocity $u_t^*$ are the respective measurements at the onset of drifting or blowing snow.

**RC2:** *Page 15, Line 15: Please show how u*t values were calculated.*

**Reply:** u$^*$ is not calculated. See reply above.

**RC2:** *Page 15, Line 32: Please define what you mean by 'minor' here. Please quantify.*

**Reply:** We did not run the model but the model bias in absolute values will cancel out because ratios are used (see Eq.2). Clarified as follows.

The model bias in q$_{bsalt}$ is expected to cancel out in estimates of bulk sublimation rate Q$_s$ (Eq. 2) and therefore also of SSA production Q$_{SSA}$ (Eq. 1) because the calculation uses not absolute values but ratios of actual q$_{bsalt}$ and its maximum q$_{b0}$.

**RC2:** *Page 17, Line 13: Please delete 'have' typo.*

**Reply:** done

**RC2:** *Page 17, Line 3: Didn't mean $d_p$ increase?*

**Reply:** Decrease is correct. Expected is a decrease of $d_p$ with height above the surface snow particle source in the absence of snowfall due to gravitational settling.

**RC2:** *Page 19, Line 32: Do you mean 0.1204 here?*

**Reply:** This has been corrected.

**RC2:** *Page 20, Lines 8-10: The wording 'well established' should be removed, as the Yang et al papers are models based on a hypothesis rather than measurement based and this associated uncertainty should be noted.*

**Reply:** Agreed and clarified as follows.

Modelling studies suggest that sea salt may be an important source of atmospheric bromine species in the mid to high southern latitudes, and that SSA from blowing snow releases bromine (Yang et al., 2008, 2010) driving ozone depletion events observed during or after snow storms (Jones et al., 2009).

**RC2:** *Page 20, Line 27: Data in Table 5 are presented in $\mu g\,g^{-1}$. Please fix or clarify.*

**Reply:** Corrected as follows.

Median bromide concentrations in snow ranged between 0.07 and 0.18 $\mu g\,g^{-1}$ (Table 5).

**RC2:** *Page 21, Line 11: Change "due a" to "due to a".*

**Reply:** Corrected.

**RC2:** *Page 21, Line 14: No data are presented examining the acidity of the surface snowpack.*

**Reply:** Agreed, pH of aerosol and snow was not measured. We therefore removed reference to acidity.

The bromine release from SSA produced by blowing snow may be more efficient because it has a large fraction of sub-micron sized particles (see section 3.4.3), and resides at the well ventilated top of the blowing snow layer.

**RC2:** *Page 23, Line 22: Delete extra "the".*

**Reply:** Corrected.

**RC2:** *Page 23, Lines 29-30: Remove "always" and replace with "often" to more appropriately reflect the data shown.*

**Reply:** Agreed and amended.

**RC2:** *Page 25, Line 27: "LL & MM"?*

**Reply:** Mentors who prefer to remain anonymous

**RC2:** *Figure 16: The variations in these distributions (e.g. standard deviations) should be shown.*

**Reply:** Agreed, we updated Figure 16 including the standard deviation of the mean values, and corrected the caption (Fig. 3).

**RC2:** *Figure 17: This figure is difficult to understand currently.*

**Reply:** We updated Figure 17 and clarified the caption (Fig. 6).

**SC1:** *This manuscript describes an interesting set of measurements and detailed analysis confirming the blowing snow as a significant source for sea salt aerosol in the vicinity of sea ice in coastal Antarctica. We agree that this is an important result with significant implications for polar tropospheric aerosol loadings and heterogeneous halogen chemistry. However, it would be helpful to both the authors and readers of this article to refer to prior work also published in ACP showing similar results from measurements taken on sea ice in the Ross Sea. Giordano et al., 2018 also clearly identifies blowing snow on sea ice as a significant source of chlorine rich sea salt aerosol from online Aerosol Mass Spectrometer measurements of aerosol composition, optical measurements of blowing snow and interstitial aerosol concentrations and offline measurements of surface and blowing snow composition. The consistency between the results from observations using different techniques and on opposite sides of the Antarctic continent further indicates the importance of this mechanism to the overall Antarctic aerosol budget.*

*Lars Kalnajs and Peter DeCarlo*

*Reference: Giordano, M. R., Kalnajs, L. E., Goetz, J. D., Avery, A. M., Katz, E., May, N. W., Leemon, A., Mattson, C., Pratt, K. A., and DeCarlo, P. F.: The importance of blowing snow to halogen-containing aerosol in coastal Antarctica: influence of source region versus wind speed, Atmos. Chem. Phys., 18, 16689-16711, https://doi.org/10.5194/acp-18- 16689-2018, 201*

**Reply:** We agree and apologise for the oversight of this interesting study (see also reply to RC2 above). We now refer to this work in the introduction:

[revised manuscript text omitted]

[a] for direct comparison of vertical differences statistics of the 29 m measurements only for times when sea ice observations at 0.2 m were available [b]sample size [c]total aggregated time during which airborne snow particles were detected

Table 4: manuscript Table S2 - Descriptive statistics of aerosol observed during 8 June - 26 July 2013 (ANT6): total number densities $N_{0.4-12}$ and particle diameter $d_p$.

| Parameter | at 0.2 m | at 2 m | at 29 m[a] | at 29 m |
|---|---|---|---|---|
| $N_{0.4-12}$ (m$^{-3}$) | | | | |
| mean | $1.7{\times}10^6$ | $1.4{\times}10^6$ | $1.4{\times}10^6$ | $2.1{\times}10^6$ |
| $\sigma$ | $2.5{\times}10^6$ | $1.9{\times}10^6$ | $1.6{\times}10^6$ | $6.4{\times}10^6$ |
| median | $8.6{\times}10^5$ | $5.6{\times}10^5$ | $8.0{\times}10^5$ | $1.1{\times}10^6$ |
| $d_p$ (µm) | | | | |
| mean | 0.67 | 0.60 | 0.67 | 0.69 |
| $\sigma$ | 0.11 | 0.06 | 0.11 | 0.14 |
| median | 0.66 | 0.60 | 0.65 | 0.66 |
| $N$[b] | 13077 | 14907 | 9963 | 48892 |
| sampling time (days)[c] | 9 | 10 | 7 | 34 |

[a] for direct comparison of vertical differences statistics of the 29 m measurements only for times when sea ice observations at 2 m were available [b]sample size [c]total aggregated sampling time

---

## Author Response (AR2)

**Reply to Referee Comment (RC2)**

We thank referee No.2 for the time taken and additional comments which helped to further improve the manuscript. Below we address all comments (in italics), with the revised text shown in blue and line numbers referring to the new non-tracked changes manuscript submitted to ACP.

**RC2:** *The authors revised the manuscript to clarify many points. Unfortunately, many references in the response seem to refer to the wrong (or missing) sections, figures, and/or tables (eg., reference to section 3.4.4 at top of page 2 of response; reference to aerosol data in Fig 16 in RC2 reply on page 8 of response, etc). This made my assessment difficult and means that I may have accidentally missed certain changes when trying to go between the response and manuscript. My questions and comments are detailed below, with line numbers referring to the non-tracked changes manuscript.*

**Reply:** As explained, our reply in the discussion forum refers to the ACPD online version, including all references to sections, tables and figures. Some section and figure numbers changed in the revised manuscript submitted to ACP as we rearranged the section order following the reviewer's suggestion; yet all changes were marked up in the tracked changes manuscript. We apologise if this made it more difficult to review.

**RC2:** *Page 5 Line 4, Page 9, Line 20, and Figure S3: This methods section describes the aerosol size distribution measurements, and therefore the header should state such. Sea salt aerosol was not chemically identified by the methods described, and using such a header is misleading. Similarly, the response Figure 4 (referred to as manuscript Figure S3) states "Sea Salt CLASP", but again, the CLASP measures all particles, not just sea salt, and therefore, it should be labeled simply as "CLASP". Further, since the CLASP measures number concentration, it is critical that the assumptions used to convert from number concentration to mass concentration are included in the figure caption, in addition to the text. Data are not available to verify that all of the particles were indeed sea salt, and this needs to be clear. In the results, "the size spectra of sea salt aerosol" are referred to, which again is misleading, as this was not measured.*

**Reply:** The method section clearly describes, what the CLASP is measuring. However, we agree that CLASP output is not equivalent to SSA and therefore amended the text by replacing SSA with aerosol, when referring to CLASP data. The underlying assumptions of the estimate of sea salt mass concentration from the CLASP are clarified in the text and now also included in the caption of Figure S3:

Page 5, Line 8: Atmospheric aerosol

Page 9, Lines 22-23: ... followed by the discussion of chemical fractionation of aerosol and snow and size spectra of aerosol above sea ice (section 3.2) ...

Page 11, Line 7: The sea ice source of aerosol

Page 14, Line 27: Aerosol size distributions

**Reply:** The underlying assumptions of the estimate of sea salt mass concentration from the CLASP are clarified in the text and now also included in the caption of Figure S3:

Page 25, Lines 3–9: To do this atmospheric sea salt mass concentrations were estimated in two ways for the time period when $RV\,Polarstern$ was well within the FYI zone (18 June to 21 July 2013; Fig. 1). One by converting spectral particle number densities $N_{0.4-12}$ at 29 m into sea salt mass concentration assuming that all particles are spherical and consist of sea salt with the density of $NaCl$ (= $2160\,kg\,m^{-3}$). Contributions from non-sea salt (nss) aerosol to CLASP $N_{0.4-12}$ can lead to an overestimate of the calculated sea salt mass but are considered to be small as nss-aerosol has a winter minimum in coastal Antarctica (e.g. Weller et al., 2011; Abram et al., 2013).

Fig S3 caption: Comparison of atmospheric sea salt mass concentrations during the 8 June to 26 July 2013 period estimated from filter (ss–filter) and CLASP (ss–CLASP) measurements at 29 m above the sea surface (R= 0.32, p<0.01). ss–filter was derived by multiplying the $Na^+$ concentration measured on aerosol filters with a conversion factor of 3.262 based on the $Na^+$ mass fraction in reference seawater. ss–CLASP was estimated based on median number densities $N_{0.4-12}$ during filter sampling intervals assuming that all particles are spherical and consist of sea salt with the density of $NaCl$ (=

$2160\,\mathrm{kg\,m^{-3}}$) (see details in main text). Symbols are color coded based on wind speed $U_{10m}$.

**RC2:** *Page 7, Lines 20-22: What is the justification for removing this bromide-enriched data? The*
*measurement uncertainty reported just not justify these samples as outliers. Surface snow is often*
*significantly enriched in bromide (e.g., see Maffezzoli et al, 2017, The Cryosphere, for East Antarctic*
*snow; see Peterson et al, 2019, Elementa, for Arctic snow), and blowing snow in Antarctica has been*
*shown to be bromide-enriched (Hara et al 2018, Sci Rep). Therefore, removing such samples from*
*the study biases the results. Further, bromide enrichment supports gas-phase bromine chemistry.*
**Reply:** Agreed. We now include all available DF-Br data in aerosol and snow, and updated accord-
ingly Tables 4 and 5, Figures 4 and 6, and text. Note all median DF-Br values changed only slightly
or remained the same, and therefore our conclusions also remain the same.

**RC2:** *Page 11 Line 2: Please clarify "blowing sea ice source".*
**Reply:** Clarified as follows:
Page 11, Lines 8–9: Below we discuss aerosol properties characteristic for a sea ice source from
blowing snow, including the chemical fractionation of $SO_4^{2-}$ and $Br^-$ as well as aerosol size distribu-
tions observed above sea ice.

**RC2:** *Page 14, Line 12: The recent work of Artiglia et al. (2017, Nature Commun.) also shows dark*
*ozone oxidation of bromide and doesn't require low pH.*
**Reply:** We added the reference to Artiglia et al. (2017) and clarified as follows:
Page 14, Lines 18–22: However, the early laboratory study by Oum et al. (1998) has shown that $HOBr$
required for $Br^-$ oxidation can be chemically produced in darkness through the reaction of ozone with
bromide, a reaction which takes place most rapidly at the air-substrate interface, where acidity plays
a minor role (Abbatt et al., 2012, and references therein), and where it was recently observed to occur
at neutral pH for the case of a aqueous phase-vapour interface (Artiglia et al., 2017).
Page 14, Lines 25–27: It therefore appears plausible that the same reactions may have caused sig-
nificant bromide depletion observed here in aerosol above sea ice.

**RC2:** *Page 14, Lines 26-27: Table 1 does not show aerosol data, so it is not clear how it supports the*
*sentence here. Further, it is not clear how lower concentrations at wind speeds below the threshold*
*for sea salt support "the absence of any local sources and the long distance to the nearest open*
*ocean". Sea salt aerosol emissions from leads increase with wind speed (Nilsson et al, 2001, JGR),*
*so one would expect increased concentrations at higher wind speeds. Further, May et al (2016,*
*JGR) show contributions from sea salt aerosol from open ocean and leads in the Arctic only for wind*
*speeds above the 4 m/s threshold, so looking at data here at only ¡4 m/s (below the threshold for sea*
*salt aerosol production) to rule out leads and open ocean influence is confusing.*
**Reply:** We clarified as follows:
Page 15, Lines 2–5: Thus a lower aerosol background above sea ice compared to the open ocean
during calm conditions is consistent with the inactivity of local sources from blowing snow or poten-
tially open leads combined with the long distance between the ship's position and the nearest open
water during 18 June to 21 July 2013 (Fig. 1, Table 1).

**RC2:** *Page 15 Lines 1-2 and Page 26 Lines 13-16: This statement of higher fluxes above sea ice than*
*open ocean needs to be supported by data, as fluxes depend on wind speed, and while the relative*
*increase (%) appears to be higher in Figure 7 for the sea ice data, the open ocean concentrations*
*are higher. Without calculated absolute values calculated for similar wind conditions, this statement*
*is not clear to me.*
**Reply:** We agree that relative and absolute increases in aerosol number and volume concentrations
need to be considered separately. The statement about expected fluxes above sea ice and open
ocean is reworded accordingly in abstract and main text:
Page 1, Lines 14-18: The relative increase of aerosol concentrations with wind speed was much
larger above sea ice than above the open ocean highlighting the importance of a sea ice source in winter and early spring for the aerosol burden above sea ice. Comparison of absolute increases of aerosol concentrations during storms suggests that to a first order corresponding aerosol fluxes above sea ice can rival those above the open ocean depending on particle size.

Page 15, Lines 18-28: It is striking that the relative increase of aerosol number densities ($d_p$<2 µm) during storms was larger above sea ice compared to the open ocean with mean enhancements by a factor of ~3.2 and ~1.8, respectively (Fig. 7a). Similarly, the relative increase of aerosol volume concentrations ($d_p$ 1-8 µm) during storms was larger above sea ice compared to the open ocean with mean increases by a factor of ~3.6 and ~1.2, respectively (Fig. 7b). The above highlights the importance of a winter time sea ice source for the aerosol burden above sea ice. For comparison, the absolute increase of aerosol number densities ($d_p$<2 µm) during storms above sea ice was larger compared to the open ocean only for the two smallest size bins ($d_p$ 0.36-0.48 µm) (Fig. 7a). And the absolute increase of aerosol volume concentrations ($d_p$ 1-8 µm) during storms above sea ice was larger compared to the open ocean only for the larger particle sizes ($d_p$ 3.5-6.9 µm) (Fig. 7b). This suggests that to a first order corresponding aerosol number and mass fluxes during storms above sea ice can rival those above the open ocean depending on particle size. Direct flux measurements are needed to further corroborate these findings under varying environmental conditions.

Page 27, Lines 6-10: The relative increase of aerosol concentrations with wind speed was much larger above sea ice than above the open ocean highlighting the importance of a sea ice source in winter and early spring for the aerosol burden above sea ice. Comparison of absolute increases of aerosol concentrations during storms suggests that to a first order corresponding aerosol number and mass fluxes above sea ice can rival those above the open ocean depending on particle size.

**RC2:** *Page 15 Line 10: This should be 1 um. Please check the mass balance of NaCl. See also Fig. 3 of Yang et al. (2019). The factor of 1000 in the above equation (Page 15 Line 8) is so that you can use psu (g/kg) directly. For example, if your salinity is 0.06 psu, i.e. 0.06 g/kg, then you just use 0.06, not 0.06E-3, in the above ddry equation. Please re-check.*

**Reply:** We corrected the computation error and adjusted the text as follows:

Page 15, Lines 11-17: A significant increase of small particle number densities ($d_p$<2 µm) during storms above sea ice (Fig. 7a; section 3.3) is consistent with a blowing snow SSA source based on physical arguments: snow particles of low salinity as observed during this study can generate a significant proportion of small sea salt particles after complete loss of water ice by sublimation. Assuming one snow particle produces one aerosol the dry particle diameter $d_{dry}$ is given by $d_{dry} = d_p(S_p/1000\, \rho_{ice}/\rho_{\mathrm{NaCl}})^{1/3}$, where $\rho_{ice}$ and $\rho_{\mathrm{NaCl}}$ are densities of ice (= 917 kg m$^{-3}$) and NaCl (= 2160 kg m$^{-3}$), respectively. Taking the medians of salinity observed in surface snow (=0.06 psu) and of mean snow particle size $\overline{d_p}$ (=115-129 µm) (Tables 5, 6) we would then expect a potential median $d_{dry}$ of 3-4 µm.

**RC2:** *Page 15 Line 14: How do you come to the following conclusion? "Observations from this study support the former requirement, but not the latter". This discussion is not clear. Also, please include references for the previous work, as ice core concentrations also depend on boundary layer heights, cloud formation, dry and wet deposition, etc., and this seems to be a simplified argument that sea salt emissions from sea ice are higher than for open water. In my opinion, this paragraph could be deleted, or at least moved out of the results section, because ice cores were not examined in this study.*

**Reply:** Agreed. We removed the paragraph from the results section and made a note in the conclusions regarding ice core interpretation as follows:

Page 27, Lines 10-15: Lower SSA concentrations above sea ice relative to the open ocean may have been due to a combination of low storm frequency and low snow salinities in the area sampled. Model calculations suggest that size distribution and associated flux of SSA originating from blowing snow are very sensitive to snow salinity $S_p$, e.g. decreasing $S_p$ from 0.92 to 0.06 psu causes SSA spectral number densities to decrease by about one order of magnitude (Yang et al., 2019). Upscaling of the SSA source flux of Weddell sea ice and atmospheric modelling similar to the study of Levine et al. (2014) are needed to address this issue as well as implications for the interpretation of sea salt in ice cores as a proxy for sea ice.

**RC2:** *Page 26 Lines 2-3: This is related to an above comment. Please explain how you get submicron particle produced from blowing snow sublimation at the salinity and blowing snow particle size reported in Tables 5 and 6. This discussion does not agree with the results of inputting these data into Fig. 3 of the related manuscript by Yang et al. (2019, ACP).*

**Reply:** Agreed. In line with the above reply we also changed the text in the conclusions:

Page 26, Lines 12-14: Significant enhancement of small aerosol particles ($d_p$<2 µm) during storms is consistent with sublimation of low-salinity (<0.1 psu) snow particles assuming complete loss of water ice.

**RC2:** *Figure 3: Please increase the font sizes in this figure to make them readable.*

**Reply:** The font size in Fig. 3 has been increased in the revised manuscript.

[revised manuscript text omitted]
 legends shows respective correlation coefficients. Note that $U_{2m}$ has been derived from the 3-D wind measurements of the sonic anemometer.

[Figure]

Figure S 2. Overview of atmospheric observations in the Weddell Sea from 14 August to 17 October 2013 (ANT-XXIX/7): (a) horizontal wind speed $U$ at 39 m. (b) ambient temperature $T_a$ and relative humidity with respect to ice $RH_{ice}$ at 29 m. (c) total number densities $N_{46-478}$ of airborne snow particles at 29 m. (d) aerosol $Na^+$ concentrations and (e) sulphate depletion factor $DF_{SO_4^{2-}}$, both at 29 m.

[Figure]

Figure S 3. Comparison of atmospheric sea salt  concentrations during the 8 June to 26 July 2013 period  estimated from filter  (ss–filter) and  CLASP (ss–CLASP) measurements at 29 m above the sea surface (R= 0.32, p<0.01). ss–filter was derived by multiplying the $Na^+$ concentration measured on aerosol filters with a conversion factor of 3.262 based on the $Na^+$ mass fraction in reference seawater. ss–CLASP was estimated based on median number densities $N_{0.4-12}$  during filter sampling intervals  assuming that all particles  spherical and consist of sea  salt with the density of NaCl $(= 2160 \, kg \, m^{-3})$ (see details in main text). Symbols are color coded based on wind speed $U_{10m}$.

Table S 1. Descriptive statistics of airborne snow particles observed for 8 June to 12 August 2013 (ANT6) and for 14 August to 16 October 2013 (ANT7): total number densities $N_{46-478}$ and particle diameter $d_p$. Statistics refer to periods when airborne snow particles were present, i.e. times with no snow particles observed were removed prior to averaging.

| Parameter | ANT6 | | | ANT7 |
| --- | --- | --- | --- | --- |
| | at 0.2 m | at 29 m[a] | at 29 m | at 29 m |
| $N_{46-478}$ (m$^{-3}$) | | | | |
| mean | $2.6{\times}10^5$ | $4.0{\times}10^3$ | $8.7{\times}10^3$ | $7.2{\times}10^3$ |
| $\sigma$ | $7.4{\times}10^3$ | $9.5{\times}10^3$ | $2.7{\times}10^4$ | $2.2{\times}10^4$ |
| median | $4.7{\times}10^3$ | $7.7{\times}10^2$ | $9.9{\times}10^2$ | $1.3{\times}10^3$ |
| $d_p$ (µm) | | | | |
| mean | 138 | 132 | 133 | 143 |
| $\sigma$ | 59 | 59 | 53 | 53 |
| median | 132 | 117 | 124 | 136 |
| $N$[b] | 8608 | 11766 | 42959 | 37123 |
| sampling time (days)[c] | 6 | 8 | 30 | 26 |

[a] for direct comparison of vertical differences statistics of the 29 m measurements only for times when sea ice observations at 0.2 m were available [b]sample size [c]total aggregated time during which airborne snow particles were detected

Table S 2. Descriptive statistics of aerosol observed during 8 June - 26 July 2013 (ANT6): total number densities $N_{0.4-12}$ and particle diameter $d_p$.

| Parameter | at 0.2 m | at 2 m | at 29 m[a] | at 29 m |
|---|---|---|---|---|
| $N_{0.4-12}$ (m$^{-3}$) | | | | |
| mean | $1.7\times10^6$ | $1.4\times10^6$ | $1.4\times10^6$ | $2.1\times10^6$ |
| $\sigma$ | $2.5\times10^6$ | $1.9\times10^6$ | $1.6\times10^6$ | $6.4\times10^6$ |
| median | $8.6\times10^5$ | $5.6\times10^5$ | $8.0\times10^5$ | $1.1\times10^6$ |
| $d_p$ (µm) | | | | |
| mean | 0.67 | 0.60 | 0.67 | 0.69 |
| $\sigma$ | 0.11 | 0.06 | 0.11 | 0.14 |
| median | 0.66 | 0.60 | 0.65 | 0.66 |
| $N$[b] | 13077 | 14907 | 9963 | 48892 |
| sampling time (days)[c] | 9 | 10 | 7 | 34 |

[a] for direct comparison of vertical differences statistics of the 29 m measurements only for times when sea ice observations at 2 m were available [b]sample size [c]total aggregated sampling time